# FTO deficiency in older livers exacerbates ferroptosis during ischaemia/reperfusion injury by upregulating ACSL4 and TFRC

Rong Li [1,7], Xijing Yan[1,2,7], Cuicui Xiao[3,7], Tingting Wang[1,2,7], Xuejiao Li [1,7], Zhongying Hu[1], Jinliang Liang [1], Jiebin Zhang[1,2], Jianye Cai[1,2], Xin Sui[4], Qiuli Liu[5], Manli Wu[6], Jiaqi Xiao [1,2], Haitian Chen [1,2], Yasong Liu [1,2], Chenhao Jiang[1,2], Guo Lv[1], Guihua Chen[1,2], Yingcai Zhang[1,2] ✉, Jia Yao [1,2] ✉, Jun Zheng [1,2] ✉ & Yang Yang[1,2] ✉

Older livers are more prone to hepatic ischaemia/reperfusion injury (HIRI), which severely limits their utilization in liver transplantation. The potential mechanism remains unclear. Here, we demonstrate older livers exhibit increased ferroptosis during HIRI. Inhibiting ferroptosis significantly attenuates older HIRI phenotypes. Mass spectrometry reveals that fat mass and obesity-associated gene (FTO) expression is downregulated in older livers, especially during HIRI. Overexpressing FTO improves older HIRI phenotypes by inhibiting ferroptosis. Mechanistically, acyl-CoA synthetase long chain family 4 (ACSL4) and transferrin receptor protein 1 (TFRC), two key positive contributors to ferroptosis, are FTO targets. For ameliorative effect, FTO requires the inhibition of *Acsl4* and *Tfrc* mRNA stability in a m6A-dependent manner. Furthermore, we demonstrate nicotinamide mononucleotide can upregulate FTO demethylase activity, suppressing ferroptosis and decreasing older HIRI. Collectively, these findings reveal an FTO-ACSL4/TFRC regulatory pathway that contributes to the pathogenesis of older HIRI, providing insight into the clinical translation of strategies related to the demethylase activity of FTO to improve graft function after older donor liver transplantation.

Hepatic ischaemia/reperfusion injury (HIRI) inevitably occurs during liver transplantation (LT), HIRI predisposes patients to early allograft dysfunction (EAD) and rejection reactions after LT. Importantly, as both the severe shortage of grafts and the rapidly ageing populations become challenging problems worldwide, organs from older donors, which were not used for transplantation in the past, are increasingly being used in the clinic; this approach is the most common strategy that is used to close the gap between supply and demand[1]. The impact of ageing on liver function is less pronounced than that on kidney or heart function. However, the ability of older livers to cope with external stress is significantly less than that of young livers. For instance, older livers are relatively more susceptible to IRI, which

[1]Guangdong Provincial Key Laboratory of Liver Disease Research, Third Affiliated Hospital of Sun Yat-sen University, Guangzhou 510630, China. [2]Department of Hepatic Surgery and Liver Transplantation Center of the Third Affiliated Hospital of Sun Yat-sen University; Organ Transplantation Research Center of Guangdong Province, Guangdong Province Engineering Laboratory for Transplantation Medicine, Guangzhou 510630, China. [3]Department of Anesthesiology, the Third Affiliated Hospital of Sun Yat-sen University, Guangzhou 510630, China. [4]Surgical ICU, the Third Affiliated Hospital of Sun Yat-sen University, Guangzhou 510630, China. [5]The Biotherapy Center, the Third Affiliated Hospital of Sun Yat-Sen University, Guangzhou 510630, China. [6]Department of ultrasound, the Third Affiliated Hospital of Sun Yat-Sen University, Guangzhou 510630, China. [7]These authors contributed equally: Rong Li, Xijing Yan, Cuicui Xiao, Tingting Wang, Xuejiao Li. ✉e-mail: zhangyc3@mail.sysu.edu.cn; yaojia6@mail.sysu.edu.cn; zhengj67@mail2.sysu.edu.cn; yysysu@163.com

severely limits the utilization of older livers in LT[2]. Therefore, revealing the underlying molecular mechanisms of IRI to facilitate the development of novel therapeutic approaches that effectively mitigate IRI in older livers is urgently needed.

Ferroptosis is a recently identified form of cell death that is mediated by iron-dependent phospholipid peroxidation, and it has been reported to participate in the occurrence and development of multiple diseases[3,4]. The liver is one of the most important organs for iron storage. Therefore, ferroptosis has been extensively implicated in various forms of liver diseases, including HIRI[5,6]. For the first time, Yamada et al. demonstrated a contribution by ferroptosis in the pathogenesis of HIRI and that inhibition of ferroptosis by ferrostatin-1 (Fer-1) can effectively alleviate HIRI[7]. Wu found that the ubiquitin ligase E3 HUWE1/MULE targeted the degradation of transferrin receptor protein 1 (TFRC) to suppress ferroptosis during HIRI[8]. The transmembrane member 16A (TMEM16A) promotes the ubiquitination and degradation of glutathione peroxidase 4 (GPX4), thereby aggravating HIRI[9]. Due to older livers are more susceptible to lipid disorders and mitochondrial failure as well as iron ions further accumulate in the liver with age, we speculated that older livers are more prone to ferroptosis[10]. However, the role of ferroptosis in older HIRI is still unclear, and revealing the potential regulatory mechanism is critical for developing effective treatments to mitigate HIRI in the older.

N6-methyladenosine (m6A), a widely distributed and highly conserved RNA modification, plays critical roles in posttranscriptional gene regulation, including RNA splicing, translation, and degradation in eukaryotes. m6A is a dynamic and reversible modification that has been linked to the stress response, and its functions are mediated by m6A-related writers, erasers, and readers[11,12]. Recent studies have reported the role of m6A modification in the regulation of IRI in myocardial, kidney, cerebral, and other tissues. Du et al. first showed the functional importance of hepatic m6A methylation during HIRI[13]. However, the potential role and related functional mechanisms of m6A modification in older HIRI are largely unknown.

In our study, we evaluated the pivotal role of ferroptosis in older HIRI using clinical specimens and both in vivo and in vitro experiments. We found that low levels of fat mass and obesity-associated gene (FTO) in older livers promoted the posttranscriptional m6A modification of acyl-CoA synthetase long chain family 4 (*Acsl4*) and *Tfrc*, which are two general contributors to ferroptosis sensitivity, and induced their expression during IRI. Furthermore, we demonstrated that nicotinamide mononucleotide (NMN), which is an essential precursor of NAD+, increased FTO demethylase activity to suppress ferroptosis and ameliorate HIRI in the older, suggesting that NMN may be used in the clinic to improve graft function after LT from older donors.

## Results

### Older livers are more prone to IRI and exhibit more ferroptosis than young livers

IR is an inevitable external stress that donor livers undergo during transplantation. Previous studies have shown that the ability of older livers to cope with external stress is weaker than that of young livers, suggesting that older livers are more prone to IRI[2]. To further verify this hypothesis, we evaluated donor age and its correlation with liver damage in a human orthotopic liver transplantation (OLT) cohort (*n* = 20). All the patients were divided into young and old groups according to the corresponding donor age (young, age < 50; old, age >= 50). Compared with the young group, the old group exhibited higher serum aspartate aminotransferase (AST) and alanine aminotransferase (ALT) levels at postoperative day 1 (POD1) (Fig. 1a), as well as slower recovery of the serum AST and ALT levels from POD1 to POD7 (Fig. 1b). Moreover, liver biopsies from the old group showed more obvious changes in liver histology and increased hepatocellular death, as shown by hematoxylin-eosin (HE) staining and TUNEL staining (Fig. 1c-top, middle; Fig. 1d-left). These results proved that older livers

were indeed more prone to IRI, while the potential regulatory mechanism involved remains unclear.

Considering both the physiological changes in older livers and the characteristics of ferroptosis[14], we speculated that older livers are more susceptible to ferroptosis and that ferroptosis may play an important role in mediating HIRI. As shown in Fig. 1c-bottom and Fig. 1d-right, in older livers after reperfusion, the accumulation of ROS, which is an important factor that triggers ferroptosis, was significantly higher than that in young livers. Ferroptosis is mainly characterized by an increase in lipid peroxidation, C11 BODIPY staining was performed, and the results showed more lipid peroxidation in older livers than in young livers (Fig. 1e). In addition, multiple key proteins related to ferroptosis were analysed, and the expression levels of nuclear factor erythroid-derived 2-like 2 (NRF2), GPX4 and ferritin heavy chain 1 (FTH1), which can inhibit ferroptosis, were significantly lower in older livers after reperfusion, while the ferroptosis-promoting molecules ACSL4 and TFRC were conversely overexpressed (Fig. 1f). Previous studies suggest that multiple cell death pathways can be involved in the process of IRI, including apoptosis, necroptosis, and pyroptosis, which are closely related to the specific context of organs or tissues. Therefore, we, herein, also evaluated the expression of apoptosis-, necroptosis-, and pyroptosis-related proteins in young/older HIRI via western blotting assays. As shown in Supplementary Fig. 1a–f, the results suggested that apoptosis, necroptosis, and pyroptosis all participated in the process of young and older HIRI to varying degrees. All these results indicate that these four cell death pathways involved in both young and older HIRI, and older liver tissue is more prone to ferroptosis during IR compared to the young.

The in vivo results confirmed that older livers were more prone to IRI, as evidenced by the increased in serum AST and ALT levels, poorly preserved hepatic architecture, and enhanced cell death (Fig. 1g–i). We also detected the levels of ferroptosis in mouse liver tissues in each group. In addition to Dihydroethidium (DHE) assays and C11 BODIPY staining, transmission electron microscopy (TEM) assays are frequently used to detect the mitochondrial morphological aberrancy associated with ferroptosis. When cells undergo ferroptosis, we can observe that the mitochondrial become smaller and the mitochondrial cristae decrease or disappear with TEM[15]. We performed DHE assays, C11 BODIPY staining, TEM assays and western blotting assays (Fig. 1j–m) further demonstrated that there was more ferroptosis in older livers than in young liver. And among multiple cell death patterns, the alteration of ferroptosis in older HIRI was relatively more significant, consistent with the data from clinical specimens (Supplementary Fig. 1d–f).

In addition, primary hepatocytes were isolated from young and old mice for in vitro validation. Supplementary Fig. 2a–d showed the method used to obtain primary hepatocytes and the morphological and general physiological characteristics of the primary hepatocytes. Then, we used primary hepatocytes to establish hypoxia-reoxygenation (H/R) injury models to simulate IRI in vitro. Calcein-acetoxymethyl (Calcein-AM)/propidium iodide (PI) double-staining assays were performed to evaluate the cell death. Primary hepatocytes derived from old mice exhibited increased cell death during IR (Supplementary Fig. 2e). The results of DHE assays, C11 BODIPY staining and western blotting assays further proved that primary hepatocytes derived from old mice showed more ferroptosis during IR (Supplementary Fig. 2f–h).

Moreover, according to a method for inducing senescence in the mouse liver cell line AML12[16], we established senescent human THLE2 cells Supplementary Fig. 2i-upper). Through cell morphology analyses, β-gal staining assays, and western blotting assays, we confirmed that $H_2O_2$ effectively induced senescence in THLE2 cells (Supplementary Fig. 2i, j). We used normal THLE2 and senescent THLE2 cells to establish H/R models and obtained results (Supplementary Fig. 2k–m) that were consistent with those in primary hepatocytes. Collectively,

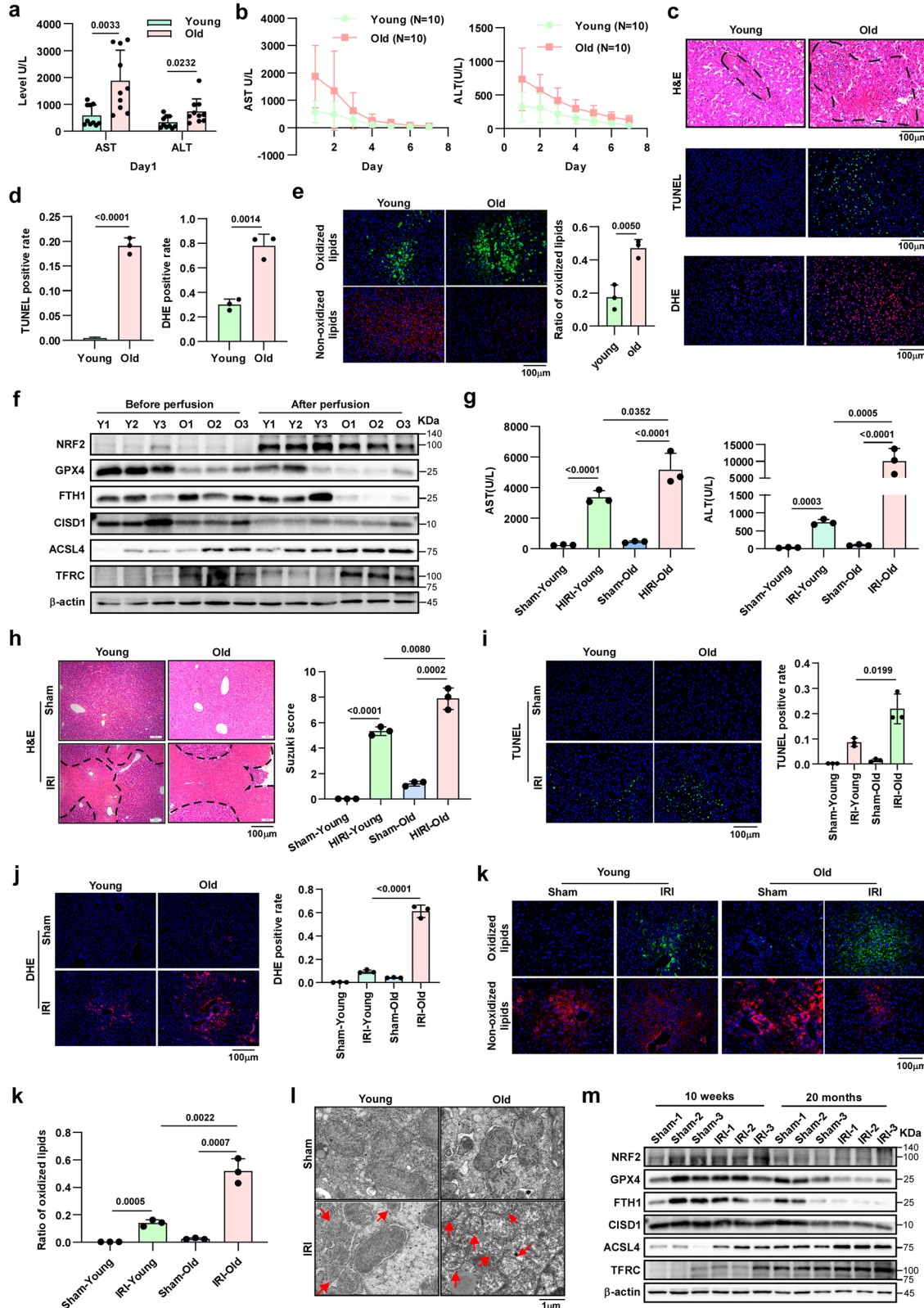

these data demonstrated that older livers were more susceptible to IRI and characterized by a higher intensity of ferroptosis than young livers.

### Ferroptosis inhibition significantly alleviates older HIRI

Based on these findings, we sought to further explore the role of ferroptosis in older HIRI. Ferrostatin-1 (Fer-1), a small molecule inhibitor that can specifically inhibit ferroptosis[17], was used to treat older HIRI. The results showed that Fer-1 significantly mitigated older HIRI, as shown by the reduced serum AST and ALT levels, relatively well-preserved hepatic architecture and decreased cell death, and Fer-1 strongly attenuated the accumulation of ROS and lipid peroxidation in older livers during IR (Fig. 2a–c). Moreover, results from in vitro characterizations verified the role of ferroptosis in older HIRI. Calcein-

**Fig. 1 | Older livers are susceptible to IRI and characterized by increased ferroptosis. a** Serum AST and ALT levels in OLT recipients at POD1 (young group, $n = 10$; old group, $n = 10$). AST and ALT, two-tailed $t$-test. **b** Curve of decreasing serum ALT and AST levels in OLT recipients from POD1 to POD7 (young group, $n = 10$; old group, $n = 10$). **c** Representative images of HE staining, TUNEL staining, and DHE staining in OLT biopsies (magnification, ×200). **d** Relative quantification of TUNEL staining and DHE staining, two-tailed $t$-test. **e** Representative images and relative quantification of C11 BODIPY staining in OLT biopsies (magnification, ×200), two-tailed $t$-test. **c**–**e** Three independent biological human samples. **f** Western blotting showed the expression of ferroptosis-associated proteins in OLT biopsies before and after perfusion. **g** Serum ALT and AST levels in mice suffering

from HIRI ($n = 3$, per group). AST and ALT, two-tailed $t$-test. **h**–**k** Representative images and relative quantification of HE staining (magnification, ×100), TUNEL staining (magnification, ×200), DHE staining (magnification, ×100) and C11 BODIPY staining (magnification, ×200) in liver tissues, two-tailed $t$-test (young mice, $n = 3$; old mice, $n = 3$). **l** Representative TEM images. The red arrows indicate the reduction or disappearance of mitochondrial cristae, which indicate ferroptosis. **m** Western blotting showed the expression of ferroptosis-associated proteins in liver tissues. **l**, **m** Three independent biological mice samples. $P < 0.05$ was considered statistically significant. Statistic data are presented as the mean ± SD, source data are provided as a Source Data file.

AM/ PI double staining, DHE staining, and C11 BODIPY staining showed that treating cells with Fer-1 to inhibit ferroptosis significantly ameliorated the damage to older hepatocytes caused by H/R (Fig. 2d–g). Taken together, these data demonstrated that inhibition of ferroptosis can strongly mitigate older HIRI.

### The m6A demethylase FTO is down-regulated in older HIRI
To further identify the factors that affect older HIRI, we performed Liquid Chromatography Mass Spectrometry (LC-MS) to screen differentially expressed proteins between young HIRI and older HIRI (Fig. 3a). The results of principal component analysis (PCA) showed that liver samples derived from young and old mice could basically be grouped independently, suggesting that there were indeed differences between the groups (Fig. 3b). As shown in the heatmaps and histograms, the expression of multiple proteins in the liver changed with age (Fig. 3c, d). All the differentially expressed proteins shown in Fig. 3d had been listed in Supplementary Table 1. Based on the differentially expressed proteins, we performed functional enrichment analysis. As shown in Supplementary Fig. 3a–d, compared with the sham groups, both young and older HIRI enriched with ferroptosis-related pathway. And the metabolic changes were more active in older groups compared to the young groups, especially the metabolic pathways associated with unsaturated fatty acids, such as "Biosynthesis of unsaturated fatty acids" and "Linoleic acid metabolism". The differentially expressed proteins in young HIRI were mainly involved in inflammation processes, such as "Neutrophil extracellular trap formation". Due to the metabolic pathways associated with unsaturated fatty acids, the main substrates for lipid peroxidation and ferroptosis, were enriched in older HIRI, we further carried out relative quantitative lipidomics to compare lipid composition between young and older livers[18,19]. After quality control of the specimens (Supplementary Fig. 4a), a total of 42 lipid classes (including 2486 lipid species) in young and older liver tissues were identified (Supplementary Fig. 4b); the levels of multiple lipid species significantly differed between these two groups, and lots of lipid species were higher expressed in the older (Supplementary Fig. 4c, d). Among them, the lipid unsaturation of several critical phospholipids for inducing ferroptosis, including membrane-associated glycerophospholipids [phosphatidylethanolamine (PE), phosphatidylserine (PS), and phosphatidylcholine (PC)] and membrane-associated phospholipids [phosphatidylglycerol (PG), phosphatidylinositol (PI), phosphatidic acid (PA) and sphingomyelin (SM)], increased in the older liver tissues (Supplementary Fig. 5a–g). In addition, we also evaluated the lipid unsaturation of fatty acids (FA), and the results suggested that there was a slight upward trend in the unsaturation of FA in the older compared to the young (Supplementary Fig. 5h). Therefore, these data further suggest that older livers are susceptible to ferroptosis.

Next, the key molecules involved in regulating older HIRI from the protein profiling were screened out according to the following criteria: (1) Differentially expressed proteins between young and older liver tissues may be related to the increased susceptibility of older livers to IRI; (2) The proteins whose expression was altered during IR may also participate in the process; and (3) The proteins whose expression

changed only in older livers during IR. Based on these, 3 molecules, namely, FTO, N-acetyltransferase 8 (NAT8), and Actin-binding LIM protein 1 (ABLIM1), were ultimately identified (Fig. 3e). FTO, which is an m6A demethylase, affects multiple physiological and pathological processes by regulating intracellular m6A methylation levels[20–22]. NAT8 can acetylate the free alpha-amino group of cysteine S-conjugates to form mercapturic acids, which have been reported to regulate apoptosis[23]. ABLIM1 is reported to act mainly as a scaffold protein. Based on their basic functions, FTO and NAT8 were selected for further study.

The expression of FTO and NAT8 in human and mouse liver tissues was measured by western blotting (Fig. 3f, g) and IHC assays (Fig. 3h, i), and the results showed that the level of FTO in older livers and senescent hepatocytes was significantly lower than that in young and it further reduced after IR. Unlike that of FTO, the expression of NAT8 did not significantly change with age or IR (Supplementary Fig. 3e, f). Therefore, we ultimately focused on FTO, which may be a key factor in regulating older HIRI. FTO, which is an m6A demethylase, mainly functions by decreasing the levels of intracellular RNA methylation. Dot blotting assays showed that the m6A methylation levels, in contrast to the FTO expression levels, were upregulated during HIRI and more pronounced with age (Fig. 3j, k). Collectively, these data suggested that the downregulation of FTO may participate in the regulation of older HIRI through the modulation of m6A methylation.

### FTO ameliorates older HIRI by inhibiting ferroptosis
To elucidate the effect of FTO on older HIRI, we first clinically measured the correlation between FTO levels in donor livers and serum liver function after OLT. As shown in Fig. 4a, b, OLT recipients with high FTO expression exhibited lower serum AST and ALT levels at POD1, as well as the rapid recovery of AST and ALT levels from POD1-POD7, suggesting that FTO may attenuate IRI in older livers. The in vitro assays in primary hepatocytes further demonstrated that overexpressing FTO significantly mitigated hepatocytes damage during H/R, while silencing FTO aggravated liver injury (Supplementary Fig. 6a, b). FB23-2, a small molecule that specially inhibits the demethylase activity of FTO, was validated in hepatocytes via dot blotting assays (Supplementary Fig. 6c). FB23-2 was also used to show that inhibiting the demethylase activity of FTO could aggravate hepatocytes damage (Supplementary Fig. 6d). Moreover, DHE staining, C11 BODIPY staining and western blotting assays showed that overexpressing FTO in primary hepatocytes significantly reduced the accumulation of intracellular ROS and lipid peroxidation (Supplementary Fig. 6e, g), which was accompanied by the upregulation of ferroptosis-inhibiting molecules (Supplementary Fig. 6k upper) during IR. While silencing FTO or inhibiting the demethylase activity of FTO presented reversed phenomena (Supplementary Fig. 6f, h–k), which also occurred in senescent THLE2 cells (Supplementary Fig. 7a–d). These results suggest that FTO mitigates older HIRI by inhibiting ferroptosis.

To further validate the role of ferroptosis in FTO alleviation of HIRI, we combined Fer-1 and FB23-2 in the treatment of the older HIRI model and found that the conditions of the combined treatment group

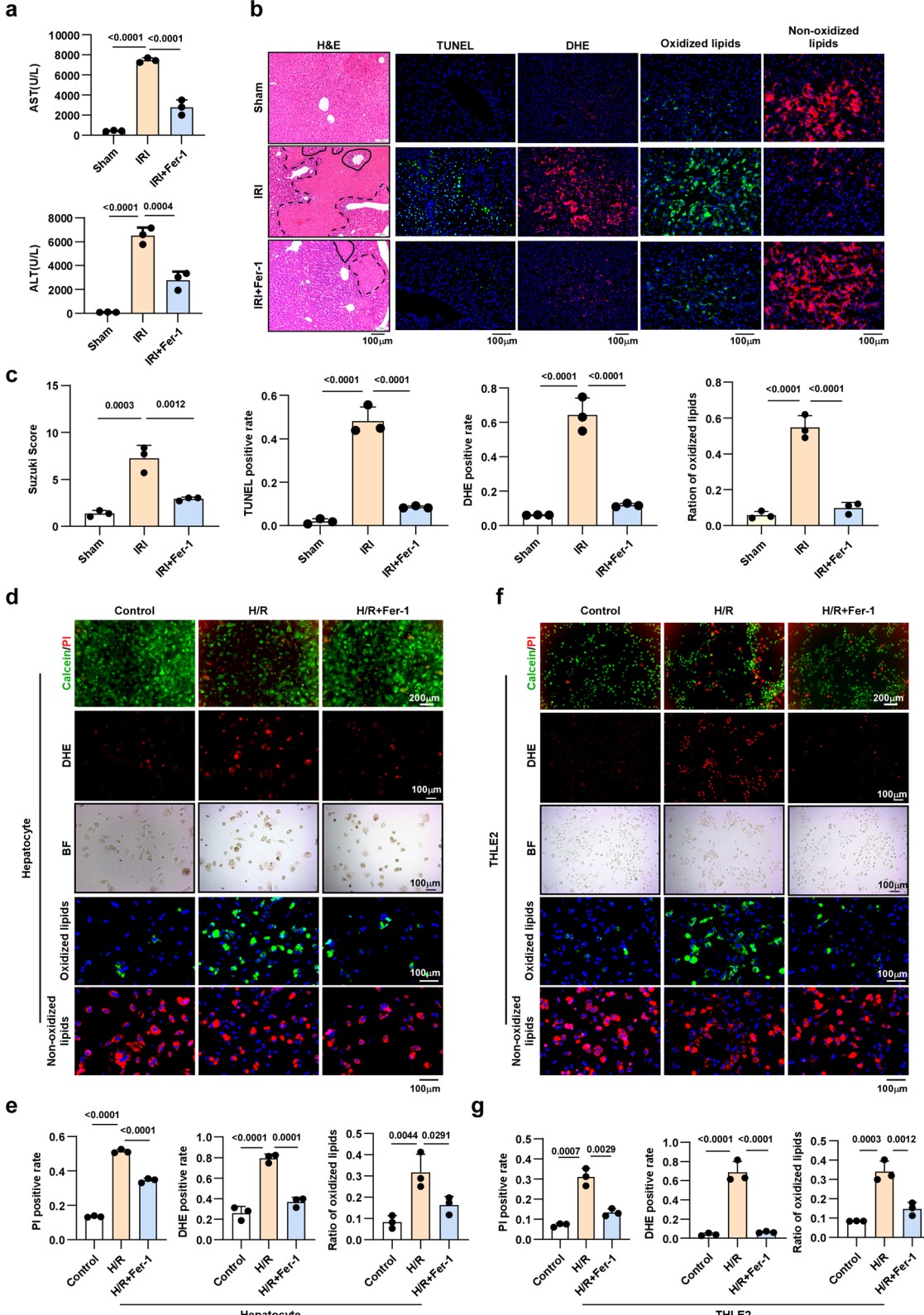

**Fig. 2 | Inhibition of ferroptosis mitigates older HIRI. a** Serum ALT and AST levels in older mice after different treatments (*n* = 3, per group). AST and ALT, one-way ANOVA followed by multiple comparisons. **b, c** Representative images and relative quantification of HE staining (magnification, ×100), TUNEL staining (magnification, ×200), DHE staining (magnification, ×100) and C11 BODIPY staining (magnification, ×200) in liver tissues with different treatments, one-way ANOVA followed by multiple comparisons. **d**–**g** Representative images and relative quantification of Calcein-AM/PI double staining, DHE staining and C11 BODIPY staining in primary hepatocytes and senescent THLE2 cells (Calcein-AM/PI, magnification, ×100; DHE, magnification, ×100; C11 BODIPI, magnification, ×200; one-way ANOVA followed by multiple comparisons). Statistic data are presented as the mean±SD, error bars represent the means of three independent experiments. *P* < 0.05 was considered statistically significant, source data are provided as a Source Data file.

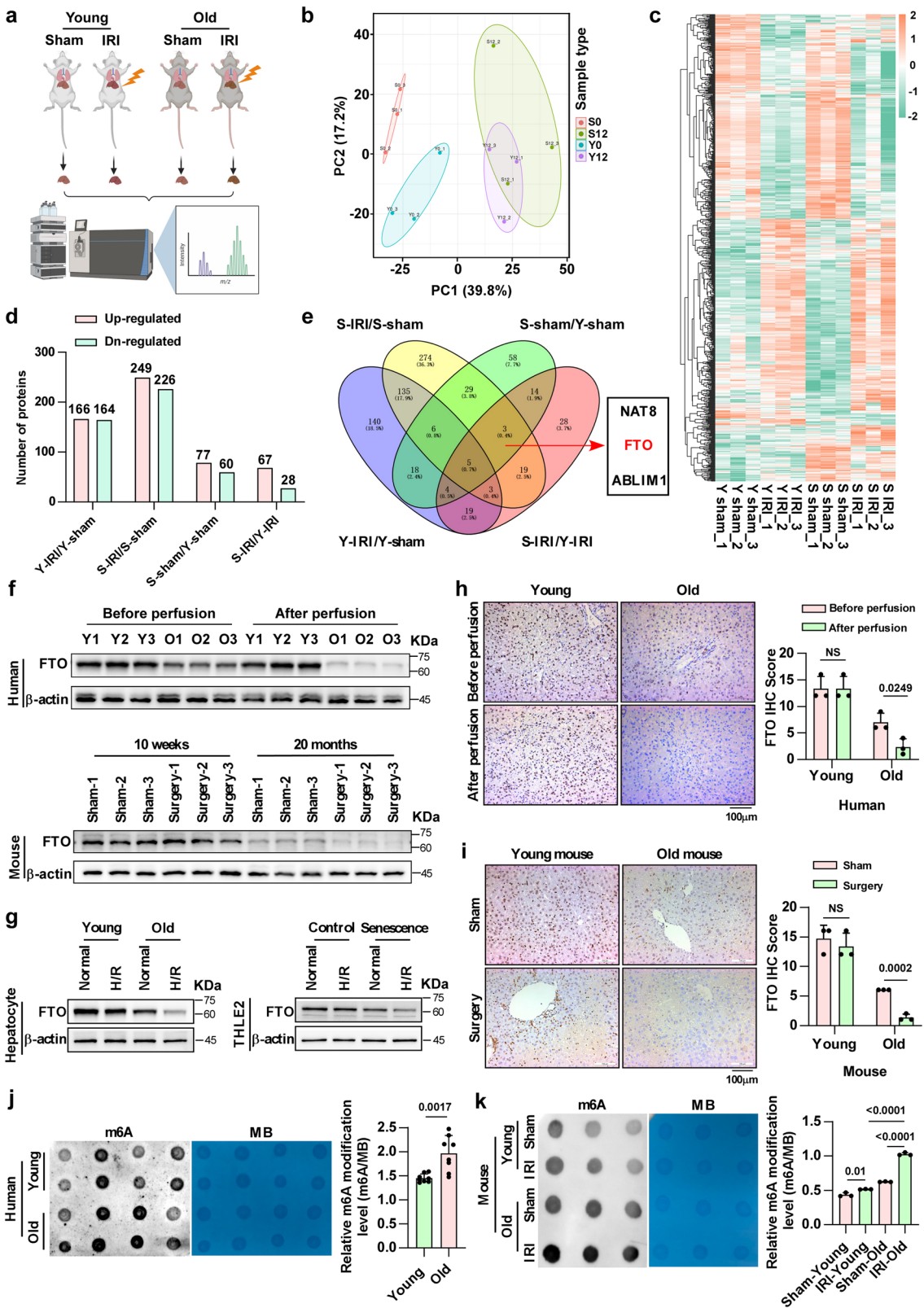

were slightly inferior to the Fer-1 group and significantly better than the FB23-2 group, as evidenced by the changes in serum AST and ALT levels, liver histology, cell death ratio, and intracellular ROS accumulation (Fig. 4c–e); these results suggested that inhibition of ferroptosis can significantly reverse the effects of FTO inactivation in older HIRI. In addition, the results of C11 BODIPY staining, TEM assays and western blotting proved that Fer-1 significantly abolished the FTO-mediated

regulation of ferroptosis during IR (Fig. 4e–g), which also indicated that FTO mitigation of older HIRI was mainly achieved through the regulation of ferroptosis.

In in vitro experiments, erastin, which is a potent ferroptosis activator, was used to confirm the role of ferroptosis in FTO-mediated alleviation of older HIRI. As shown in Fig. 4h and Supplementary Fig. 8a, b, inhibition of ferroptosis by Fer-1 significantly reversed the

**Fig. 3 | FTO is expressed at low levels in older livers and further reduced in HIRI. a** Schematic illustration of LC-MS on liver samples obtained from the HIRI model in young and older mice (*n* = 3, pre group). **b** PCA showed both intragroup repeatability and intergroup variability. S0, old-sham group; S12, old-IRI group; Y0, young-sham group; Y12, young-IRI group. **c** Heatmap showing differentially expressed proteins in the different groups. **d** Relative quantification of up/down-regulated proteins in 4 comparison groups. **e** Venn diagram presenting the number and distribution of differentially expressed proteins in the 4 comparison groups. NAT8, FTO, and ABLIM1 were selected for further investigation. Western blotting showed the expression of FTO in liver tissues (**f** human, upper; mouse, lower) and cells (**g** primary hepatocytes, left; THLE2 cells, right) of different ages during IR. *n* = 3 independent biological mice samples or three independent cell experiments.

Representative images and relative quantification of IHC staining of FTO in young and older liver tissues during IR (**h** human; **i** mouse, two-tailed *t*-test. magnification, × 200). Determination of m6A abundance in mRNA of different groups via dot blotting assay (**j** human samples, *n* = 8 per group; **k** mouse samples. *n* = 3 per group). MB (methylene blue) represents the loading control of RNA samples. The relative m6A quantifications were displayed with bar charts, one-way ANOVA followed by multiple comparisons. Statistic data are presented as the mean±SD, error bars represent the means of three independent experiments. *P* < 0.05 was considered statistically significant. NS no significance, source data are provided as a Source Data file. Figure 3a created with BioRender.com released under a Creative Commons Attribution-NonCommercial-NoDerivs 4.0 International license https://creativecommons.org/licenses/by-nc-nd/4.0/deed.en.

effects of FTO inactivation on cell death and the accumulation of intracellular ROS and lipid peroxidation during H/R, which was consistent with that of the in vivo experiments, while treatment with erastin yielded the opposite results (Fig. 4i and Supplementary Fig. 8c, d). Collectively, these data further suggested that FTO-mediated mitigation of older HIRI was mainly achieved through the regulation of ferroptosis.

### FTO inhibits the stability of ACSL4 and TFRC mRNA via m6A demethylation

To explore the underlying mechanism by which FTO regulates ferroptosis, methylated RNA immunoprecipitation sequencing (MeRIP-seq) was performed in senescent THLE2 cells transfected with control or FTO overexpression plasmids and subjected to H/R. MeRIP-seq revealed a total of 1689 different peaks between the FTO-overexpressing group and the control group, and these peaks were mainly distributed in exons (41.5%), 3′UTRs (38.66%) and 5′UTRs (13.56%) (Fig. 5a). Among them, the m6A modification levels of 49.38% of the differentially abundant peaks were downregulated, and the other 50.62% of the differentially abundant peaks were upregulated (Fig. 5b). Two representative FTO potential binding motifs were shown in Fig. 5c. Considering that when FTO is overexpressed, the m6A levels of its downstream targets should decrease, we performed functional enrichment analysis with the down-regulated peaks and found that these peaks were enriched in ferroptosis-related pathways (Fig. 5d). ACSL4 and TFRC, which can induce ferroptosis, were predicted to be potential targets of FTO. Overexpression of FTO significantly decreased the m6A modification levels of the 5′UTR of *Acsl4* and the 3′ UTR of *Tfrc* (Fig. 5e). Then, MeRIP-qPCR was performed to examine the changes in the m6A modifications of *Acsl4* and *Tfrc*. The results showed that FTO overexpression could decrease the m6A levels of *Acsl4* and *Tfrc* (Fig. 5f). To further validate that FTO can target *Acsl4* and *Tfrc*, we also performed FTO-CLIP-seq. As shown in Fig. 5g, the regions enriched by FTO in *Acsl4* and *Tfrc* overlapped with the regions where the m6A modification levels significantly decreased under FTO overexpression (Fig. 5e), which further suggested that FTO could bind *Acsl4* and *Tfrc* to decrease the m6A modification of both.

Next, RT-qPCR and western blotting assays were performed to determine whether FTO-mediated m6A modification affected the expression of ACSL4 and TFRC, and the results showed that overexpression of FTO inhibited the expression of ACSL4 and TFRC, whereas silencing FTO or inhibiting its demethylase activity significantly reversed this effect (Fig. 5h, i and Supplementary Fig. 9a, b). In human and mouse liver specimens, the expression of FTO was also negatively correlated with the levels of ACSL4 and TFRC (Fig. 5j and Supplementary Fig. 9c). In addition, inhibiting FTO demethylase activity upregulated the expression of ACSL4 and TFRC in the older HIRI model (Supplementary Fig. 9d).

Based on the effect of m6A modification on regulating the stability and decay of mRNA, we examined the half-life of the *Acsl4 and Tfrc* mRNAs after coculture with actinomycin D. We observed that overexpression of FTO resulted in a reduction in the half-life of the *Acsl4*

*and Tfrc* mRNAs (Fig. 5k, l), potentially attributed to the interference of m6A reader-mediated regulation on *Acsl4 and Tfrc*. To figure out this, we identified the m6A readers of *Acsl4 and Tfrc*. YTHDF1, YTHDF2, YTHDF3, EIF3A, IGF2BPs, YTHDC1, and YTHDC2 are the most dominant m6A readers in cells. Based on previous reports[24–27] on the features of these m6A readers, we selected YTHDF1, YTHDF2, EIF3A, IGF2BP1, and YTHDC2 for further investigation. We first detected the effects of different readers on the expression levels of ACSL4 and TFRC. As shown in Supplementary Fig. 9e, YTHDC2 could increase the level of ACSL4 and YTHDF2 could increase the level of TFRC, which suggested that YTHDC2 may be the m6A reader of *Acsl4* and YTHDF2 may be the m6A reader of *Tfrc*. These finding were further validated with RNA Immunoprecipitation (RIP) assays (Supplementary Fig. 9f). Previous studies have shown that YTHDF2 and YTHDC2 could increase the stability of mRNA stability[28,29]. We speculated that the erasure of m6A modification on *Acsl4* and *Tfrc* by FTO affected YTHDC2 regulation of ACSL4 and YTHDF2 regulation of TFRC; these hypotheses were further investigated via western blotting and RIP assays. As shown in Fig. 5m–p, overexpressing FTO almost abolished the regulation of ACSL4 and TFRC mediated by YTHDC2 and YTHDF2, while silencing FTO could enhance YTHDC2 regulation of ACSL4 and YTHDF2 regulation of TFRC. These data collectively suggested that FTO could shorten the half-life of *Acsl4* and *Tfrc* mRNAs via the removal of post-transcriptional m6A modification, which ultimately reduced the expression of ACSL4 and TFRC.

### FTO-mediated mitigation of older HIRI via ferroptosis occurs in an ACSL4- and TFRC-dependent manner

To explore the roles of ACSL4 and TFRC in FTO-mediated mitigation of older HIRI, we induced short-term and transient downregulation of ACSL4 or TFRC expression via hepatotropic adeno-associated virus (AAV8) injection into older mice via the tail vein. The AAV8 we used simultaneously expressed GFP; the in vivo imaging results showed that the AAV8 was mainly concentrated in the liver (Supplementary Fig. 10a, b). Immunofluorescence (IF) assays further demonstrated that hepatocytes could effectively internalize AAV8 (Supplementary Fig. 10c), and western blotting assays confirmed the knockdown efficiency of AAV8-shACSL4 and AAV8-shTFRC in liver tissues (Supplementary Fig. 10d–f).

Furthermore, FB23-2 was used to treat these two AAV8-pretreated models, and the results showed that knockdown of ACSL4 or TFRC not only effectively mitigated older HIRI but also significantly reversed the effects of FB23-2 in these models, as shown by the decrease in serum hepatic enzyme levels and the alleviation of hepatic pathological changes. These results suggested that the hepatoprotective potential of FTO depended on its regulation of ACSL4 and TFRC (Fig. 6a, b and Supplementary Fig. 11a). In addition, knockdown of ACSL4 or TFRC markedly reversed the FB23-2 mediated accumulation of intracellular ROS (Supplementary Fig. 11b), the widen of lipid peroxidation (Supplementary Fig. 11c), the increase in the number of shrunken mitochondria (Fig. 6c), and the expression of key ferroptosis indicator proteins (Fig. 6d) in older HIRI.

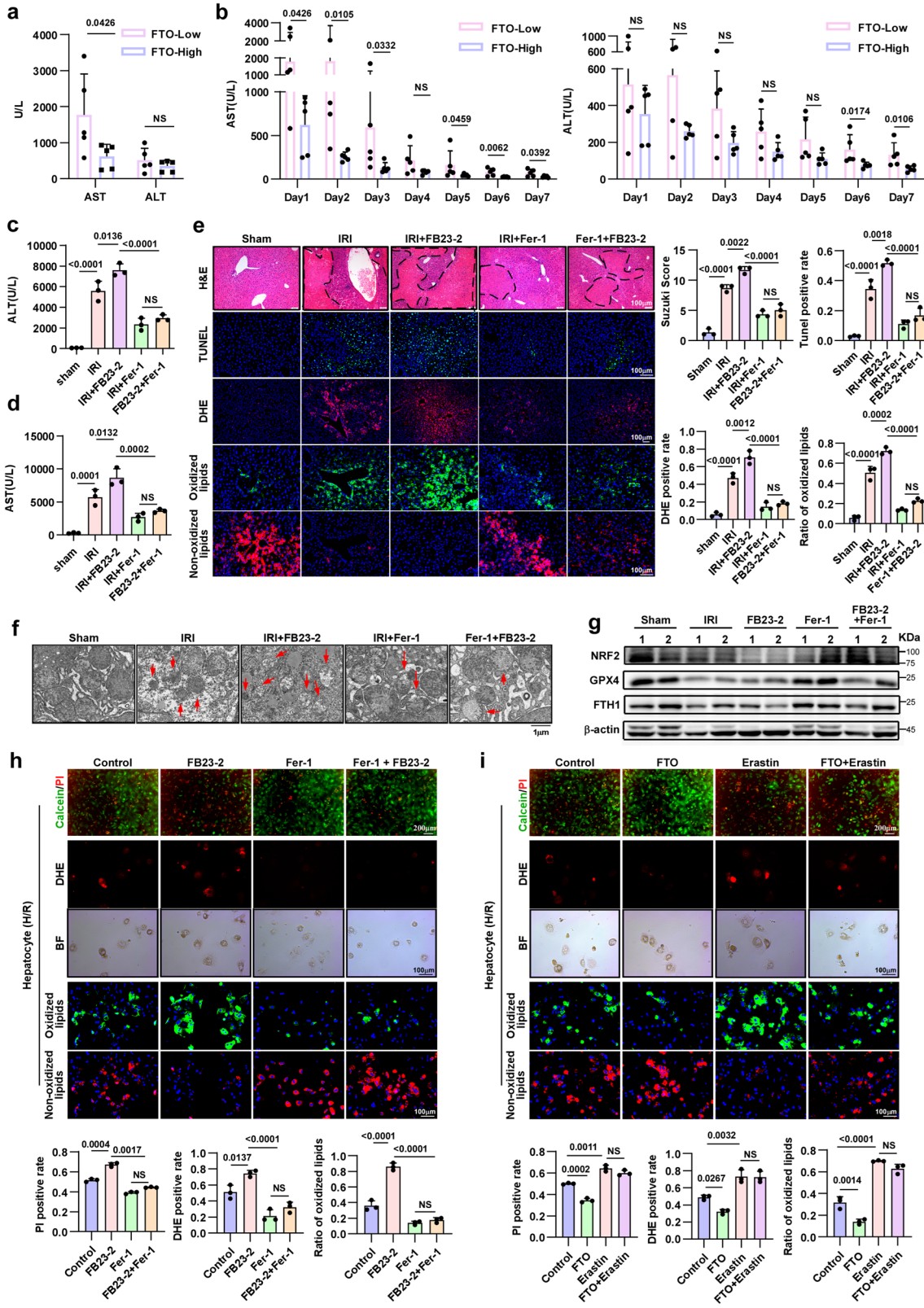

In vitro, the overexpression of ACSL4 or TFRC in senescent hepatocytes significantly reversed the FTO-mediated decrease in cell death (Fig. 6e and Supplementary Fig. 12a), intracellular ROS accumulation (Fig. 6g, h and Supplementary Fig. 12d), and lipid peroxidation during IR (Fig. 6k, l). Conversely, knockdown of ACSL4 or TFRC significantly weakened the effects of FTO silencing or FB23-2 treatment (Fig. 6f, i–j, m and n and Supplementary Figs. 11d–f, 12b, c, e, and f). Taken together,

these in vivo and in vitro data demonstrated that FTO alleviated older HIRI in a manner dependent on its regulation of ACSL4 and TFRC.

## NMN alleviates older HIRI by enhancing the demethylase activity of FTO

At present, drugs that inhibit ferroptosis are still in the research stage, limiting their clinical translation application. Based on the data in this

**Fig. 4 | FTO mitigates older HIRI by decreasing ferroptosis. a** Serum AST and ALT levels in OLT recipients at POD1 (FTO-Low group, $n = 5$; FTO-High group, $n = 5$). All donors were over 50 years old. Recipients were grouped according to the IHC score of FTO in liver tissues. AST and ALT, two-tailed $t$-test. **b** Changes in serum AST and AST levels in OLT recipients with different FTO expression levels from POD1 to POD7. AST and ALT, two-tailed $t$-test. **c, d** Serum ALT and AST levels of older mice in different groups ($n = 3$, per group). ALT and AST, one-way ANOVA followed by multiple comparisons. **e** Representative images and relative quantification/Suzuki score of HE staining (magnification, ×100), TUNEL staining (magnification, ×200), DHE staining (magnification, ×100) and C11 BODIPY staining (magnification, ×200) in older liver tissues with different treatments, one-way ANOVA followed by multiple comparisons. **f** Representative TEM images. The red arrows indicate the reduction or disappearance of mitochondrial cristae to reflect ferroptosis.

**g** Western blotting showed changes in the expression of key factors related to ferroptosis. **f, g** Three independent biological mice samples. **h** Representative images and relative quantification of Calcein-AM/PI double staining (magnification, ×100), DHE staining (magnification, ×200) and C11 BODIPY staining (magnification, ×200) to evaluate the effect of FB23-2 on primary hepatocytes during H/R, one-way ANOVA followed by multiple comparisons. **i** Effects of erastin on cell death (magnification, ×100), ROS accumulation (magnification, ×200) and lipid peroxidation (magnification, ×200) in primary hepatocytes during H/R, one-way ANOVA followed by multiple comparisons. Statistic data are presented as the mean±SD, error bars represent the means of three independent experiments. $P < 0.05$ was considered statistically significant. NS no significance, source data are provided as a Source Data file.

study, drugs that increase FTO demethylase activity may be useful for inhibiting ferroptosis to ameliorate older HIRI. Recently, Lina Wang reported that NADP+ and NAD+ could enhance FTO demethylase activity[30]. However, direct supplementation with NAD+ cannot effectively increase the level of NAD+ in the body due to its molecular weight and stability. At present, the use of NAD+ precursors have been recognized as the main approach for increasing the content of NAD+. NMN is one of the important precursors of NAD+ and has been approved for antiaging treatments[31]. Herein, we wondered whether NMN could be an effective therapeutic approach for alleviating older HIRI in FTO-related m6A manner.

As shown in Fig. 7a–c, NMN treatment substantially ameliorated older HIRI, as shown by the changes in the serum hepatic enzyme levels, H&E staining, and TUNEL staining. Moreover, m6A dot blotting assays showed that NMN significantly downregulated the level of m6A modification in older livers during IR, confirming that NMN could enhance the demethylase activity of FTO (Fig. 7d). Finally, DHE staining, C11 BODIPY staining, western blotting, and TEM assays further demonstrated the potential of NMN inhibition of ferroptosis (Fig. 7e–h). Collectively, our data revealed that an FTO-ACSL4/TFRC regulatory pathway contributes to the pathogenesis of older HIRI and that NMN can increase FTO demethylase activity to ameliorate older HIRI (Fig. 8). These results may be useful for clinical translation to alleviate IRI LT patients with older donor.

## Discussion

With the ageing of the population, the proportion of older livers in the donor pool increases yearly. However, older livers are more prone to IRI than young livers, and the underlying mechanism is still not fully elucidated, limiting the utilization of older livers in LT. In this study, we provide the first evidence that ferroptosis plays a pivotal role in the pathogenesis of older HIRI and that its inhibition can effectively mitigate HIRI. Furthermore, the decline of the demethylase FTO in older livers, which had not been previously reported, acts as a critical inducer of ferroptosis by increasing the m6A modification of *Acsl4* and *Tfrc* (two key ferroptosis inducers), subsequently increasing their levels. In addition, enhancing the demethylase activity of FTO with NMN (an antiaging drug) could effectively reduce the expression of ACSL4 and TFRC, eventually inhibiting ferroptosis, which provides an effective avenue for alleviating older HIRI in the clinic.

Although various patterns of cell death, including necrosis, apoptosis, and pyroptosis, have been reported to participate in the pathogenesis of HIRI in older mice, treatments targeting these processes can only partially attenuate liver injury, indicating that another pathway of cell death may be involved in older HIRI[32,33]. Ferroptosis is a recently discovered form of cell death, and it has gradually become a hot research topic in the field of age-related diseases because senescence likely causes iron overload and ROS accumulation and disrupts lipid metabolism and mitochondrial function. Recent studies have revealed that the lens epithelium, kidney tubular epithelium, and skeletal muscle in older organisms are more susceptible to ferroptosis

under external stresses[34–36]. Bao also reported that ferroptosis occurs in neuronal cells during intracerebral haemorrhage in older organisms[37]. However, the association between ferroptosis and older HIRI has never been investigated. Combining the factors that trigger ferroptosis and the physiological characteristics of older livers, we speculated that older livers are more prone to ferroptosis during IRI. In this study, we comprehensively evaluated the importance of ferroptosis in older HIRI. Through harvesting paired liver grafts during the pre-reperfusion and postreperfusion periods at our centre, establishing HIRI murine models, and performing a series of in vitro experiments, we revealed that older livers were indeed more susceptible to IRI and exhibited a higher intensity of ferroptosis than young livers. Treatment with Fer-1 to specifically inhibit ferroptosis could effectively reduce older HIRI. Indeed, we herein selected 50 years of age as the cut-off value to distinguish the clinical cases into the young and old groups due to only a small number of clinical liver specimens collected at >60 years of age that completely covered the pre-reperfusion and postreperfusion periods, which was different from the previous studies. However, we provided sufficient evidence to show that ferroptosis was aggravated in the liver tissues after IRI with age. Therefore, thoroughly exploring the underlying mechanisms that regulate ferroptosis is pivotal to developing effective therapies to alleviate older HIRI.

As the most widespread posttranscriptional mRNA modification, m6A has been shown to be crucial for affecting the progression of multiple liver diseases[38]. Recent studies have also shown that m6A can regulate ferroptosis in various ways. The m6A methyltransferase METTL3 positively regulates ferroptosis by suppressing the expression of SLC7A11 and FSP1[39]. The reader YTHDF1 promotes ferroptosis by maintaining *Becn1* mRNA stability and increasing autophagy activation[40]. In contrast, another reader, YTHDF2, has been reported to stabilize ferroptosis-associated lncRNA (*lncFal*) to inhibit susceptibility to ferroptosis[41]. Huang et al. recently revealed the ability of the m6A demethylase FTO to suppress ferroptosis by erasing the m6A modification of the *Otub1* transcript[42]. However, the association between m6A methylation and ferroptosis in older HIRI is still unclear. Herein, we first found that FTO was downregulated in older livers and that its expression was further reduced during IR, eventually triggering changes in intracellular m6A methylation levels. Therefore, does FTO affect ferroptosis in older HIRI? A series of gain- and loss-of-function experiments proved that FTO could effectively inhibit ferroptosis and alleviate older HIRI.

FTO was the first identified m6A demethylase, which paved the way for research on m6A methylation and the epitranscriptome. To our knowledge, the exact biological role of FTO during ferroptosis in older HIRI remains unclear. Previous studies have demonstrated that FTO could inhibit ovarian ageing by slowing the degradation of *Fos* mRNA in an m6A-dependent manner[43]. In addition, FTO can stabilize MIS12 to counteract the senescence of human embryonic stem cells (hESCs)[44]. In this study, we revealed that the decline of FTO in senescent hepatocytes increased the abundance of *Acsl4* and *Tfrc* by

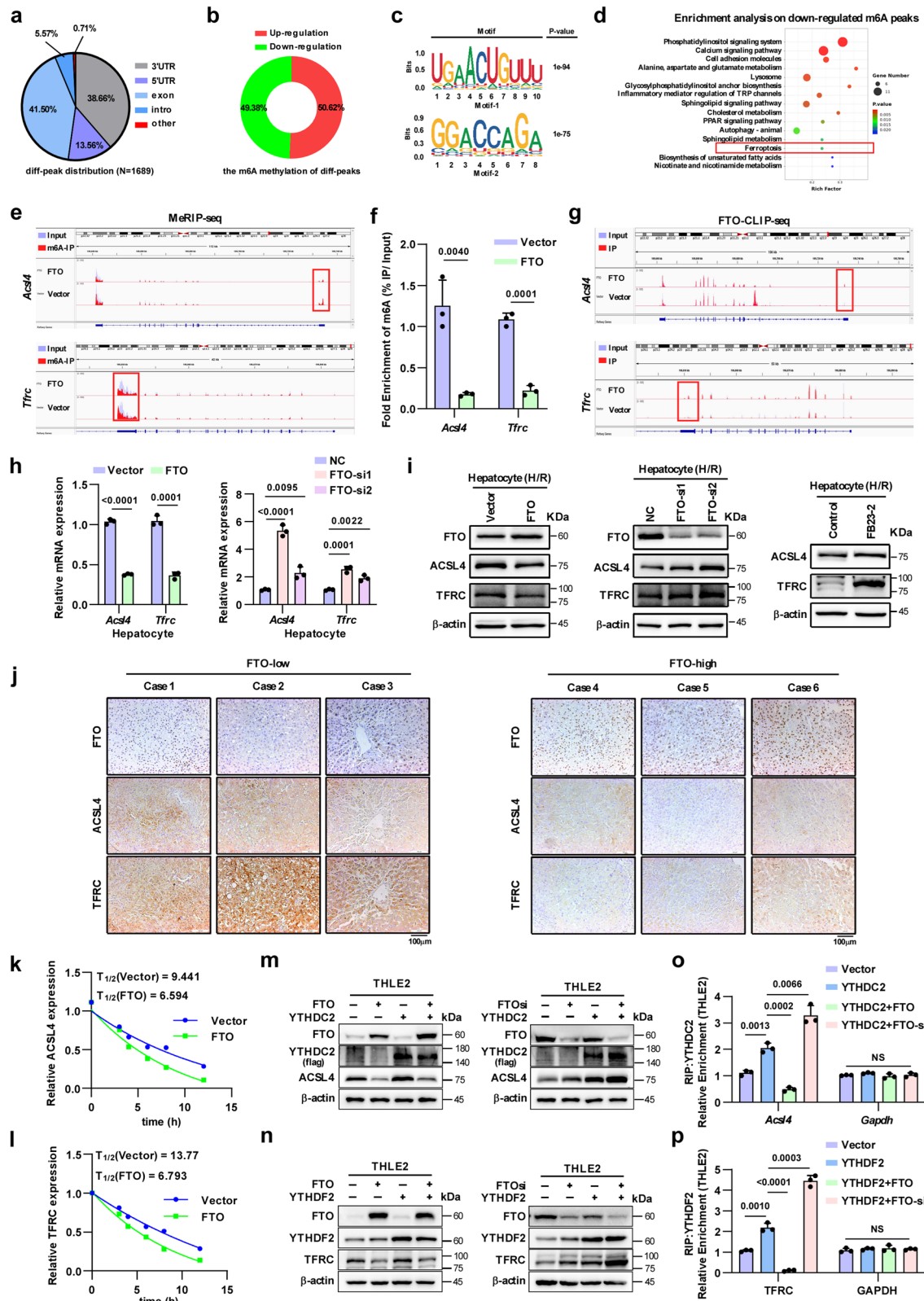

reversing the m6A modification of these mRNAs, eventually triggering ferroptosis and aggravating older HIRI. ACSL4 and TFRC are two of the most extensively recognized ferroptosis markers; ACSL4 is required for polyunsaturated fatty acid (PUFA)-containing phospholipid bio-synthesis, and TFRC is the primary iron transporter that promotes the accumulation of the labile iron pool. To date, the mechanisms underlying ACSL4 and TFRC dysregulation in diseases have not been

fully elucidated, implying that exploring strategies that target them is crucial for disease treatment. Previously, SPI1[45], Yap1[46], and HSP90[47] were shown to promote the expression of ACSL4, while STAT3 decreased ACSL4[48]. Yap1[46], YTHDF1[49], and MYCN[50] were reported to upregulate TFRC, while β-Trcp suppressed its expression. Most of these studies were performed in the context of tumours, and the regulatory mechanisms of ACSL4 and TFRC in other benign diseases

**Fig. 5 | FTO inhibits the expression of ACSL4 and TFRC by decreasing m6A modification. a** Pie chart showing the distribution and relative proportion of differential peaks between the control and FTO groups according to m6A RNA sequencing. **b** Pie chart showing the percentages of up/downregulated differential peaks. **c** Representative FTO potential binding motifs. **d** Bubble chart showing the enriched signaling pathways with down-regulated m6A peaks. Binomial distribution test (two-sided), $P < 0.05$ was considered statistically significant. **e** Integrated genome browser views show the m6A modifications of *Acsl4* and *Tfrc* in the control and FTO groups. **f** MeRIP-qPCR quantitative analysis of the fold enrichment of *Acsl4* and *Tfrc* m6A levels by immunoprecipitation with a specific m6A antibody in senescent THLE2 cells transfected with control and FTO (two-tailed *t*-test). **g** Integrated genome browser views show the binding peaks by FTO in *Acsl4* and *Tfrc* via FTO-CLIP-seq. **h** RT–qPCR validation of target gene expression after overexpression or silencing of FTO in primary hepatocytes (two-tailed t-*test*). **i** Effects of FTO on the expression of ACSL4 and TFRC in primary hepatocytes via western blotting, three independent cell experiments. **j** Representative images of IHC staining (magnification, × 200) for FTO, ACSL4 and TFRC in clinical liver specimens, three independent clinical human samples per group. **k, l** Effects of FTO on the half-life of *Acsl4* and *Tfrc* mRNA in senescent THLE2 cells via RT–qPCR. **m** Effects of FTO on the regulation of YTHDC2 on ACSL4 via western blotting. **n** Effects of FTO on the regulation of YTHDF2 on TFRC via western blotting. **m, n** Three independent cell experiments. **o** Effects of FTO on the enrichment of *Acsl4* mRNA by YTHDC2 via RIP assays, one-way ANOVA followed by multiple comparisons. **p** Effects of FTO on the enrichment of *Tfrc* mRNA by YTHDF2 via RIP assays, one-way ANOVA followed by multiple comparisons. Statistic data are presented as the mean±SD, error bars represent the means of three independent experiments. $P < 0.05$ was considered statistically significant, source data are provided as a Source Data file.

still need to be investigated. In this study, we demonstrated for the first time that FTO could erase the m6A modification of *Acsl4* and *Tfrc* mRNAs, and influence the regulation of *Acsl4 and Tfrc* by m6A readers (YTHDC2/YTHDF2) to decrease mRNA stability and inhibit the expression of ACSL4 and TFRC, which may effectively inhibit ferroptosis in older HIRI. Interestingly, silencing ACSL4 or TFRC did not completely improve the affection of inhibiting FTO demethylase activity, indicating that FTO may also have other methods to alleviate older HIRI in addition to regulating ferroptosis, deserving further investigation.

Currently, treatments for inhibiting ferroptosis are, still in the research stage and are not available for clinical application. Therefore, drugs that increase FTO demethylase activity may be useful for ameliorating older HIRI. Recently, Wang demonstrated the potential of NAD+ in enhancing FTO demethylase activity[30]. However, NAD+ cannot be administered directly due to its high molecular weight and instability. The administration of NMN, which is an intermediate in NAD+ biosynthesis, via the diet can increase the level of NAD+ in the body, and such supplements have received increasing attention as promising antiaging health products[51]. In addition, a wide range of pharmacological activities of NMN have been explored in various diseases, including obesity, type 2 diabetes, NAFLD, and Alzheimer's disease[52]. Here, we attempted to treat older HIRI with NMN and found that accompanied by the upregulation of FTO demethylase activity, NMN effectively suppressed the expression of ACSL4 and TFRC, reduced ferroptosis and significantly alleviated older HIRI.

In summary, we have provided sufficient data to demonstrate that enhancing FTO demethylase activity via NMN alleviates older HIRI by decreasing the expression of ACSL4 and TFRC, both of which are key molecules that contribute to ferroptosis. Further clinical trials need to be carried out to evaluate the effect of NMN on improving liver function in patients who receive older livers during OLT.

## Methods

### Ethics approval and consent to participate
All procedures with human samples in this study approved by the Ethics Committee of the Third Affiliated Hospital of Sun Yat-sen University (Guangzhou, China) (Approval No. [2022]02-322-01). Informed written consent was obtained from all participants as well. All experimental procedures involving animals were carried out complying with the Chinese legislation regarding experimental animals and approved by the Institutional Animal Care and Use Committee (IACUC), Jennio Biotech Co., Ltd. (Approval No. JENNIO-IACUC-2022-A003).

### Clinical data and human liver sample collection
This study was approved by the Ethics Committee of the Third Affiliated Hospital of Sun Yat-sen University (Guangzhou, China). We harvested liver specimens from donors of different ages before and after reperfusion during OLT at the Department of Hepatic Surgery and Liver Transplantation Center of the Third Affiliated Hospital of Sun Yat-

sen University between May 2020 and November 2021. A portion of each liver sample was embedded in paraffin, and another portion was directly frozen in liquid nitrogen. Prereperfusion specimens were collected from donor livers within 3 h after cold perfusion, and reperfusion specimens were collected within 2 h after hepatic artery anastomosis. In addition, the serum ALT and AST levels of these patients were measured from POD1 to POD7. Organ donations were contributed voluntarily, and informed consent was obtained from all subjects or their relatives.

### Animals
The animals used in this experiment were all mice on the C57BL/6 background. Young mice (8–10 weeks old, male) and old mice (20–24 months old, male) were purchased from the Model Animal Research Center of Nanjing University (Nanjing, China). All the experimental mice had free access to water and standard rodent chow and were housed in a specific pathogen-free (SPF) environment with a temperature of 25 °C, a humidity of 40–70%, and a 12 h light-dark cycle. The animal procedures were carried out in compliance with Chinese legislation regarding the use of experimental animals and approved by the Institutional Animal Care and Use Committee (IACUC), Jennio Biotech Co., Ltd. The grouping of each part of the experiment followed the principle of randomization.

### Establishment of the mouse HIRI model
The standard procedure for establishing a mouse HIRI model was conducted according to previously described methods. In brief, after anaesthetization with 1% pentobarbital sodium (100 μl/10 g), an atraumatic artery clamp was used to interrupt blood supply to the left and middle liver lobes for 90 min to induce ischaemia. Then, the clamp was removed to initiate the reperfusion stage. The sham group underwent laparotomy alone. All the mice were sacrificed 12 h after the operation, and serum and liver specimens were harvested for further analysis.

To investigate the role of ferroptosis and m6A in older HIRI, we treated older mice with Fer-1 (20 mg/kg, S7243, Selleck, Houston, TX) or FB23-2 (20 mg/kg, HY-127103, MCE, NJ, USA) dissolved in 2% dimethyl sulfoxide (DMSO, D5879-500ML, Sigma–Aldrich), 5% Tween-80, or Nicotinamide Mononucleotide (NMN, 500 mg/kg, S5259, Selleck, Houston, TX) in normal saline. The control group was treated with the same volume of solvent. According to the manufacturers' instructions, Fer-1 or FB23-2 was intraperitoneally administered 2 h prior to ischaemia, while NMN was administered 4 h prior to ischaemia. The number and age of mice in each experiment are stated in figure legends.

### Animal adeno-associated virus-8 (AAV8) injection and processing
To evaluate the effects of ACSL4 and TFRC on the FTO-mediated inhibition of ferroptosis during HIRI in older livers, we established a

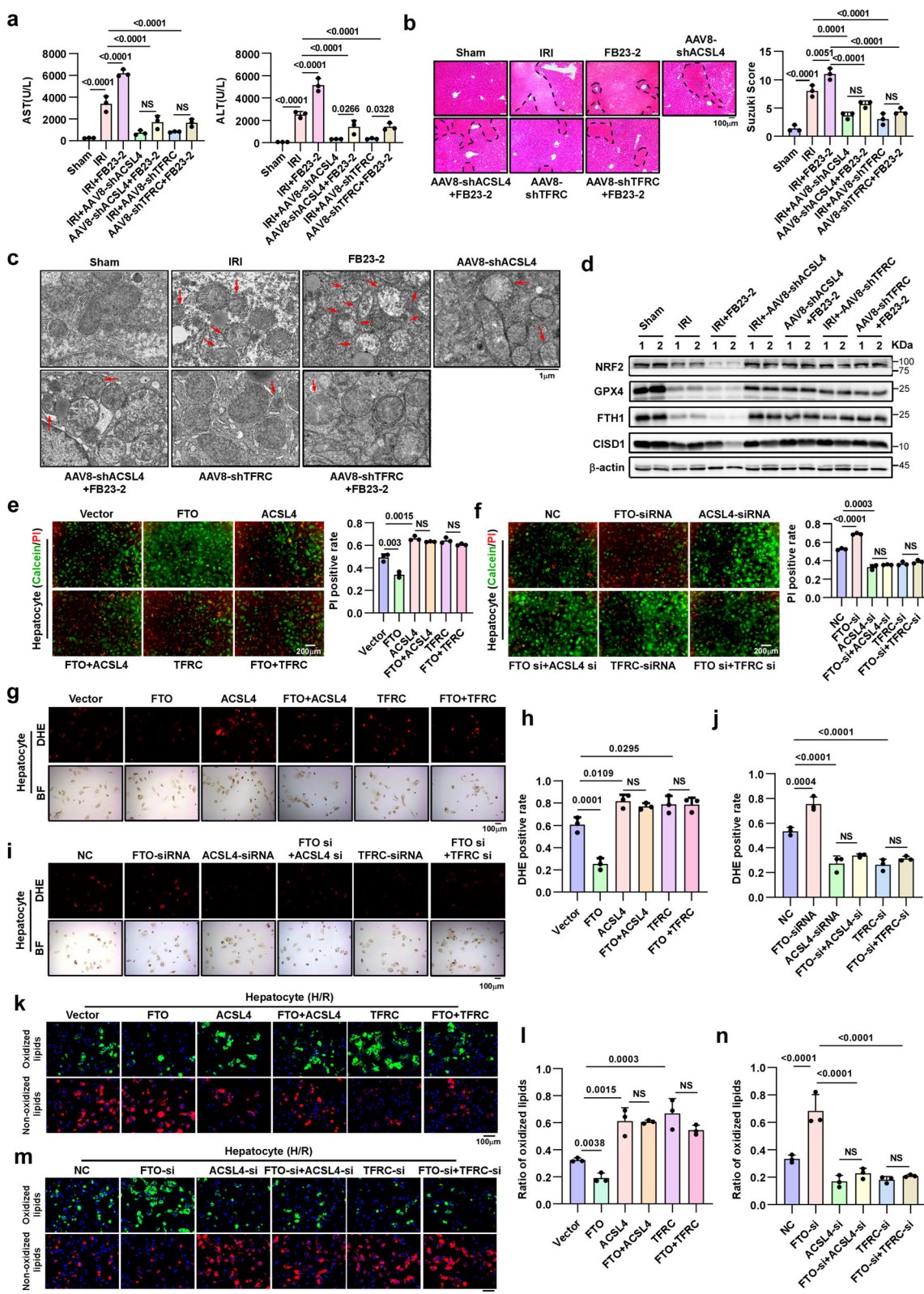

model of hepatocyte-specific ACSL4 and TFRC knockdown in older C57BL/6 mice (three mice per group unless otherwise stated, 20 months, male) by administering shRNA-AAV8-ACSL4 ($1 \times 10^{12}$ viral genomes·mL$^{-1}$, sequence: GCTGCCTGTCCACTTGTTA) or shRNA-AAV8-TFRC ($1 \times 10^{12}$ viral genomes·mL$^{-1}$, sequence: CCA-GATCAGCATTCTCTAA) via tail vein injection; these constructs were purchased from Hanbio Biotechnology, Shanghai, China. The mice in

the control group were administered shRNA-AAV8-NC ($1 \times 10^{12}$ viral genomes·mL$^{-1}$). Similarly, serum and liver specimens were collected for subsequent experiments. Live animal imaging, qRT–PCR, and Western blotting were conducted to verify knockdown efficiency.

After verifying the AAV efficiency, we randomly divided the older C57BL/6 mice into 7 groups. Three groups were injected with AAV8-control via the tail vein, 2 groups were injected with AAV8-ACSL4, and

**Fig. 6 | FTO-mediated mitigation of older HIRI is ACSL4 and TFRC dependent.**
**a** Serum ALT and AST levels of older mice with different treatments ($n = 3$, per group). AST and ALT, one-way ANOVA followed by multiple comparisons.
**b** Representative images of HE staining (magnification, ×100) and relative Suzuki score in older liver tissues in different groups, one-way ANOVA followed by multiple comparisons. **c** Representative TEM images. The red arrows indicate the reduction or disappearance of mitochondrial cristae, which indicates ferroptosis. Scale bars, 1 μm. **d** Western blotting showed the expression of key proteins related to ferroptosis in older liver tissues with different treatments. **c, d** Three

independent biological mice samples. **e–n** Representative images and relative quantification of Calcein-AM/PI double staining (magnification, ×100), DHE staining (magnification, ×100) and C11 BODIPY staining (magnification, ×200) of primary hepatocytes during H/R after different treatments, one-way ANOVA followed by multiple comparisons. Statistic data are presented as the mean±SD, error bars represent the means of three independent experiments. $P < 0.05$ was considered statistically significant. NS no significance, source data are provided as a Source Data file.

the other 2 groups were injected with AAV8- TFRC. After 3 weeks, the HIRI model was established in all 7 groups of mice. The 3 groups that had been injected with AAV8-control were sequentially divided into the sham, IR and IR + FB23-2 groups. The 2 groups that had been injected with AAV8-ACSL4 were sequentially divided into the IR + AAV8-ACSL4 and AAV8-ACSL4 + FB23-2 groups. The other 2 groups that had been injected with AAV8-TFRC were sequentially divided into the IR + AAV8-TFRC and AAV8-TFRC + FB23-2 groups.

## Live animal imaging
In vivo noninvasive multispectral fluorescence imaging was performed using the Bruker Small Animal Optical Imaging System (In-Vivo Xtreme II; Billerica, MA). Mice were anaesthetized by the inhalation of 2.5% isoflurane (Attane, MINRAD Inc.) in $O_2$ at a flow rate of 3–5 μl/min. The filter channels that were used to analyse the fluorescence intensity (eGFP) in livers in each group were Ex.488/Em.509. To observe the accumulation of AAV8 in the liver, the regions of interest of displayed images were quantified using LivingImage software 4.2 (Xenogen, Alameda, CA), and the data are presented as average radiance (photons/s/cm2/steradian). Ex situ measurements were conducted on the heart, lungs, liver, spleen, and kidneys using the same imaging system.

## Liver injury assessment
The serum ALT and AST levels in each group were measured with an A7180 Biochemical Analyzer (Hitachi, Japan). The liver specimens were fixed in 4% paraformaldehyde (PFA), embedded in paraffin, and prepared in 4 μm sections followed by staining with haematoxylin and eosin (H&E). The severity of HIRI was evaluated in a blinded manner by two pathologists under a light microscope (Leica, Germany). Hepatic injury was scored based on Suzuki's criteria, which include three indicators: sinusoidal congestion, necrosis of hepatocytes, and ballooning degeneration. Each indicator is scored on a 5-point scale from 0 to 4. Each sample is randomly photographed with five fields of view, and the Suzuki's score of this sample is the average of the five fields of view.

## Immunohistochemical (IHC) staining
Paraffin-embedded liver sections (4 μm) were deparaffinized and then rehydrated with an ethanol gradient, and then, antigen retrieval was performed with sodium citrate buffer or ethylene diamine tetraacetic acid (EDTA) according to the primary antibody. The liver sections were blocked with QuickBlock Blocking Buffer (P0260, Beyotime, Shanghai, China) for 30 min at room temperature and then incubated with primary antibodies at 4 °C overnight. Then, secondary antibody incubation and DAB colouration were performed using the Dako REAL EnVision Detection System (K5007, Copenhagen, Denmark). The primary antibodies used for IHC targeted FTO (1:400, Santa Cruz, USA), ACSL4 (1:500, Abcam, Cambridge, MA), and TFRC (1:1000, Abcam). The IHC staining intensity was scored as 1 (negative), 2 (weak), 3(moderate), and 4 (strong). Staining extent was scored as 1 (0–25%), 2 (25–50%), 3 (50–75%), and 4 (75–100%) according to the percentage of positively stained cells. The final IHC score = staining intensity × Staining extent. Five random fields per sample were evaluated by two investigators blinded to grouping. The catalog numbers of the corresponding primary antibodies are listed in the Reporting Summary.

## Immunofluorescence (IF) staining
After primary hepatocytes adhered to the culture dish, IF staining was performed to analyse their cellular properties. The cells were fixed with 4% PFA for 20 min at room temperature followed by incubation with 0.5% Triton (T8787-50ML, Sigma–Aldrich, USA) at room temperature for 10 min to increase cell membrane permeability. Then, the cells were blocked with 10% bull serum albumin (BSA, V900933-100G, Sigma–Aldrich, USA) for 1 h and treated with primary antibodies diluted in 1% BSA overnight at 4 °C. Finally, the cells were washed with PBS containing 0.1% Triton and incubated with secondary antibody [Cy3-labelled Goat Anti-Mouse IgG (H + L) (Beyotime)] for 1 h at 37 °C in the dark, and nuclear staining was performed with DAPI (KGA1523, Key-Gen, Jiangsu, China). For liver cryosections from AAV8-injected mice, only the nuclei needed to be stained. Images were acquired with a fluorescence microscope (Leica). The catalog numbers of the antibodies are listed in the Reporting Summary.

## Isolation and identification of primary hepatocytes
Primary hepatocytes were isolated from C57BL/6 mice older 10 weeks and 20–24 months using the portal vein perfusion method as previously described. Briefly, the liver was digested by administrating preheated Hanks Balanced Salt Solution (HBSS, C14175500BT, ThermoFisher) containing 1 mg/ml collagenase type H (11087789001, Sigma–Aldrich, St Louis, MO, USA) through the portal vein. The liver was then excised, minced, and teased through a 70 μm cell strainer (352350, Falcon, BD Biosciences). After centrifugation at $50 \times g$ for 5 min at 4 °C 3 times and removal of the supernatant, the cell pellet was resuspended in 20 ml complete medium and gently overlaid onto 20 ml 90% Percoll (17089109, GE Healthcare, USA). The system was then mixed uniformly and centrifuged at $200 \times g$ for 10 min at 4 °C. After discarding the supernatant, the cell pellet was resuspended in complete medium for counting and plating.

Primary hepatocytes were cultured in DMEM/Ham's F-12 (C11330500BT, Thermo Fischer) supplemented with 1% penicillin-streptomycin (15140122, Gibco, Carlsbad, CA, USA) and 100 nM dexamethasone (D1756-25MG, Sigma–Aldrich) at 37 °C in an incubator with 5% $CO_2$. We performed IF staining with a CK18 surface antibody to verify the purity of the extracted primary hepatocytes.

## Cell lines
THLE-2 cells were purchased from the Guang Zhou JENNIO Biotech Co. Ltd (GuangZhou, China) and were maintained in BEGM medium Kit (BEGM Bullet Kit, CC3170; Lonza/Clonetics Corporation, Walkersville). The kit includes 500 mL basal medium, 5 ng/mL EGF, 70 ng/mL Phosphoethanolamine and supplemented with 10% fetal bovine serum (FBS, 10270-106, Gibco, Carlsbad, CA, USA) and 1% penicillin–streptomycin (15140122, Gibco, Carlsbad). The cell line was monitored for mycoplasma contamination every 6 months and was identified by STR (Short Tandem Repeat).

## Induction of THLE2 cell senescence with hydrogen peroxide ($H_2O_2$)
Senescence was induced in THLE2 cells by prolonged treatment with $H_2O_2$ as previously described. After reaching 60–70% confluence, the cells were treated with serum-free culture medium supplemented with

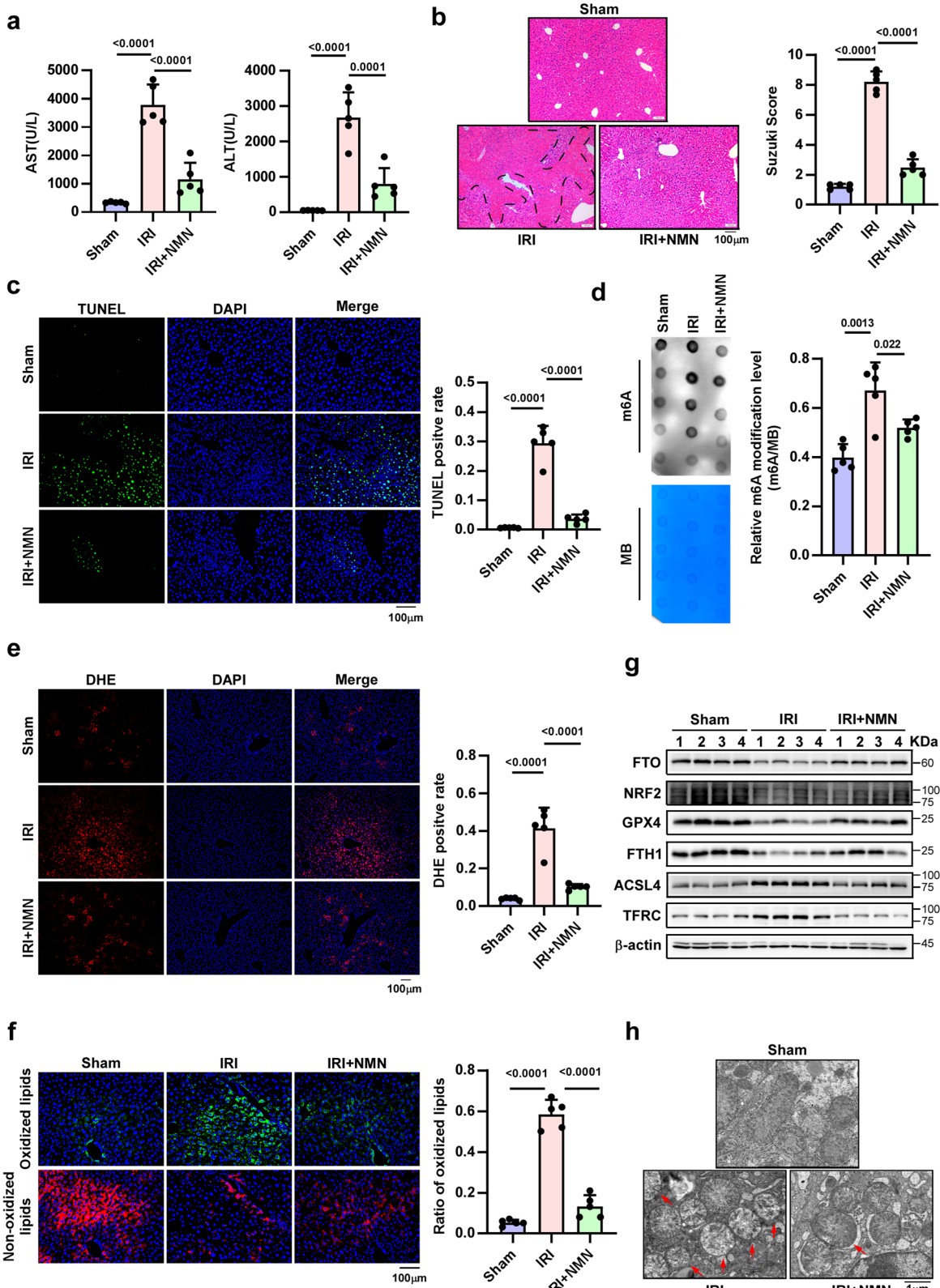

100 µM $H_2O_2$ for 30 min, and then, the medium was replaced with complete culture medium for 23.5 h of incubation for recovery. This cycle was repeated 3 consecutive times.

## β-galactosidase (β-gal) assay

A β-galactosidase assay was performed on primary hepatocytes and cell lines using a Senescence Cells Histochemical Staining Kit (CS0030,

Sigma-Aldrich, St Louis, MO, USA) according to the manufacturer's protocol. The cells in each group were washed twice with PBS and then fixed with fixation buffer for 5 min at room temperature followed by incubation in staining mixture (pH 6.0) without $CO_2$ at 37 °C overnight. Finally, the cells were observed under an inverted light microscope (Leica), and the percentage of cells that were positively stained with β-galactosidase was calculated.

**Fig. 7 | NMN mitigates older HIRI by increasing the demethylase activity of FTO.**
**a** Serum AST and ALT levels in older mice with different treatments, one-way ANOVA followed by multiple comparisons. **b** Representative images for HE staining (magnification, ×100) and relative Suzuki score in older liver tissues with different treatments, one-way ANOVA followed by multiple comparisons. **c** Representative images and relative quantification of TUNEL staining (magnification, ×200) to evaluate the effect of NMN on cell death, one-way ANOVA followed by multiple comparisons. **d** Determination of m6A abundance in mRNA of older liver tissues in different groups via dot blotting, one-way ANOVA followed by multiple comparisons. **e, f** Representative images and relative quantification of DHE staining

(magnification, ×100) and C11 BODIPY staining (magnification, × 200) to evaluate the effect of NMN on ROS accumulation and lipid peroxidation. one-way ANOVA followed by multiple comparisons. **a–f** Five independent biological mice per group. **g** Western blotting showed the expression of key proteins related to ferroptosis in different groups. **h** Representative images of TEM assays. The red arrows indicate the reduction or disappearance of mitochondrial cristae, which indicate ferroptosis. **g, h** Five independent biological mice per group. $P < 0.05$ was considered statistically significant, statistic data are presented as the mean±SD, source data are provided as a Source Data file.

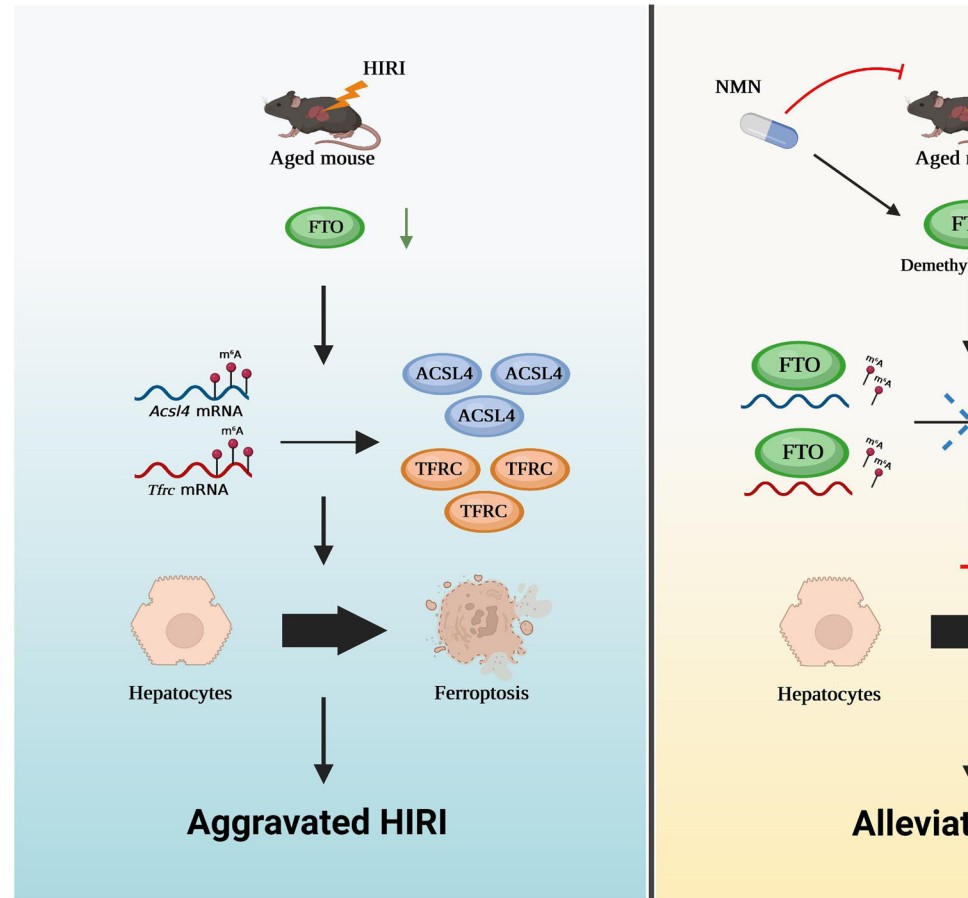

**Fig. 8 | FTO-ACSL4/TFRC signaling-Ferroptosis axis in older HIRI.** Schematic diagram depicting that FTO-deficiency in the older liver triggers a higher degree of ferroptosis during HIRI through increasing the mRNA stability of *Acsl4* and *Tfrc*. NMN could be used to upregulate FTO demethylase activity to suppress ferroptosis

and attenuate older HIRI. Figure 8 created with BioRender.com released under a Creative Commons Attribution-NonCommercial-NoDerivs 4.0 International license https://creativecommons.org/licenses/by-nc-nd/4.0/deed.en.

## In vitro hypoxia reoxygenation (H/R) model
To mimic HIRI, H/R models were established in primary hepatocytes and cell lines as described previously. In brief, the cell medium was replaced with serum-free medium, and the cells were cultured under hypoxic conditions (1% $O_2$, 5% $CO_2$, and 94% $N_2$) for 4 h; then, the medium was replaced with complete medium, and the cells were transferred to a normoxic incubator and incubated for 2 h to allow reoxygenation.

## Terminal deoxynucleotidyl transferase-mediated dUTP nick end labelling (TUNEL) assay
TUNEL assays (One Step TUNEL Apoptosis Assay Kit, C1088, Beyotime) were performed to evaluate the ratio of cell apoptosis in processed cells and frozen tissue sections according to the manufacturer's protocol. The specimens were washed twice with PBS, fixed with 4% PFA for 30 min at room temperature, and permeabilized

with 0.5% Triton for 5 min. Then, the samples were incubated with TUNEL detection solution at 37 °C in the dark for 1 h. After counterstaining with DAPI, the TUNEL-positive cells were detected under a fluorescence microscope and counted in 5 HPF/section. The TUNEL positive rate = the number of TUNEL positive cells / the number of total cells.

## Reactive oxygen species (ROS) detection
The level of ROS generation was measured using dihydroethidium (DHE, HY-D0079, MCE, NJ, USA). DHE dye (final concentration of 2.5 µM) was prewarmed at 37 °C. After washing twice with prewarmed PBS, the samples were incubated with DHE dye in a constant temperature incubator at 37 °C for 30 min. The fluorescence intensity was investigated under a fluorescence microscope (Leica, Germany) at Ex/Em = 518/616 nm. Each sample is randomly photographed with five fields of view, and the quantitative value of this sample is the average of

the five fields of view. The DHE positive rate = the number of DHE positive cells / the number of total cells.

## C11 BODIPY staining

Frozen tissue sections and cells seeded onto the glass coverslips were incubated with 1 µM C11 BODIPY Reagent (GLPBIO, GC40165, Montclair, CA) for 30 min at 37 °C in the dark. Then cells and sections were rinsed with PBS and fixed at room temperature with 4% paraformaldehyde for 20 min. After washing with PBS 3 times, the slides were sealed with DAPI. To detect oxidized (green) and non-oxidized (magenta) signals, 581/591 nm (Texas Red filter) and 488/510 nm (FITC filter) wavelengths were used. All fluorescence images were digitally acquired with Leica Upright Epifluorescence Microscope. Each sample is randomly photographed with five fields of view, and the quantitative value of this sample is determined by calculating the average rate of oxidized lipids across these five fields of view. The ratio of oxidized lipids = the number of cells with oxidized lipids / the number of total cells.

## Transmission electron microscopy (TEM) assays

Mouse liver tissues were collected within 1–3 min after cardiac exsanguination, cut to a size of 2 mm × 2 mm, and quickly fixed in 2.5% glutaraldehyde (Servicebio, G1102, WuHan, China). Tissues were fixed in the dark for 2 h at room temperature and then transferred to 4 °C for further storage. The mitochondrial ultrastructure was observed under a transmission electron microscope to determine the occurrence and severity of ferroptosis.

## Calcein-AM / PI double staining assays

Calcein-AM / PI double staining (Beyotime, C2015, China) was performed to evaluate cell death in vitro. Indicated cells were cultured in a 12-well plate at $5-8 \times 10^4$ per well overnight and then treated with different procedure. Afterwards, the cells were washed with PBS gently and stained with calcein AM and PI for 30 min. The visualization of dead cells (red) and live cells (green) was under Incucyte® Live-Cell Analysis (Sartorius, German). Each sample is randomly photographed with five fields of view, and the quantitative value of this sample is the average of the five fields of view. The ratio of cell death was calculated as the number of PI positive cells / the number of total cells.

## Western blotting analysis

Western blotting was conducted according to previously described methods. The concentrations of proteins extracted from tissues and cells were determined by the BCA protein assay kit (23227, Thermo Fisher Scientific, UK). Then, the proteins in each group were separated by 12% sodium dodecyl sulfate (SDS)-polyacrylamide gel electrophoresis (PAGE) and transferred to polyvinylidene difluoride membranes (PVDF, 0.45 µm, Merck Millipore, Billerica, MA, USA). Subsequently, 5% nonfat milk was used to block the nonspecific antigens in the membranes for 1 h followed by incubating the membranes with the primary antibodies [including antibodies against NRF2 (1:500; Cell Signaling Technology, Danvers, MA, USA), FTH1 (1:2000; Cell Signaling Technology, Danvers, MA, USA), CISD1/mitoNEET (1:2000; Cell Signaling Technology, Danvers, MA, USA), FTO (1:1000; Cell Signaling Technology, Danvers, MA, USA), Rb (1:1000; Cell Signaling Technology, Danvers, MA, USA), Phospho-Rb (Ser807/811) (1:1000; Cell Signaling Technology, Danvers, MA, USA), Phospho-Rb (Ser780) (1:1000; Cell Signaling Technology, Danvers, MA, USA), β-Actin (1:2000; Cell Signaling Technology, Danvers, MA, USA), p53 (1:1000; Abcam, Cambridge, MA), Glutathione Peroxidase 4 (1:5000; Abcam, Cambridge, MA), p27 Kip1 (1:500; Santa Cruz, USA), p21 Waf1/Cip1 (1:500; Santa Cruz, USA), Flag (1:1000; Cell Signaling Technology, Danvers, MA, USA) YTHDF2 (1:500; Cell Signaling Technology, Danvers, MA, USA) and YTHDC2 (1:500; Cell Signaling Technology, Danvers, MA, USA) on a table concentrator overnight at 4 °C. Then, anti-mouse IgG (Cell Signaling Technology) or anti-rabbit IgG (Cell Signaling Technology),

as appropriate, was used as an HRP-conjugated secondary antibody and incubated with the membranes for 1 h at room temperature. After using an enhanced chemiluminescence (ECL) substrate, the blots were investigated with a FluorChem Systems imager (ProteinSimple, CA, USA). The catalog numbers of the antibodies are listed in the Reporting Summary.

## Plasmids, transfection, and RNA knockdown

FTO, ACSL4, TFRC, YTHDF1 and YTHDF2 plasmids were obtained from DHbio (Guangzhou, China). Human FTO, ACSL4, TFRC, YTHDF1 and YTHDF2 plasmids: pcDNA3.1-Flag-FTO (OENM_001080432), pcDNA3.1-Flag-ACSL4 (OENM_022977-3), pcDNA3.1-Flag-TFRC (PR2012281), pcDNA3.1-YTHDF1-3×FLAG (OENM_017798) and pcDNA3.1-YTHDF2-3×FLAG (HY231516). Mouse FTO, ACSL4 and TFRC plasmids: pcDNA3.1-Flag-mFTO (OENM_001080431), pcDNA3.1-Flag-mACSL4 (OENM_207625), and pcDNA3.1-Flag-mTFRC (OENM_011638). Human IGF2BP1, EIF3A and YTHDC2 plasmids were obtained from Miaoling Bio (Wuhan, China). Human IGF2BP1, EIF3A and YTHDC2 plasmids: pCMV-T7-IGF2BP1-3×FLAG-Neo (P41601), pCMV-EIF3A-3×FLAG-Neo (G32852) and pCMV-YTHDC2-3×FLAG (G32851). siRNAs targeting FTO, ACSL4 and TFRC were generated by GenePharma (Shanghai, China). The transfection of plasmids was performed using ViaFect Transfection Reagent (E4982, Promega, Madison, USA) according to the manufacturer's instructions. Transient knockdown of target genes by siRNA was performed with Lipofectamine 3000 reagent (L3000015, Invitrogen, Carlsbad, CA, USA). Plasmids and siRNAs were transfected for 48 h, and then, the samples were subjected to H/R and drug treatment. Overexpression and knockdown efficiencies were verified by WB and qRT–PCR. The sequences of siRNAs are listed in Supplementary Table 2.

## Total RNA extraction and quantitative real-time polymerase chain reaction (RT–qPCR)

Total RNA was extracted from frozen samples and cultured cells using TRIzol (15596018, Invitrogen) according to the manufacturer's protocol. The amount and purity of total RNA were checked by Biophotometer plus (Eppendorf, Germany), and a Transcriptor First Strand cDNA Synthesis Kit (Roche Applied Science, USA) was used to reverse transcribe the RNA into cDNA. cDNA amplification was conducted using a PCR Thermal Cycler (Bio-Rad, USA) by heating for 10 min at 65 °C, incubation for 30 min at 55 °C, deactivation for 5 min at 85 °C, and incubation for 5 min at 4 °C. Finally, RT–qPCR was performed using a reverse transcription system (LC-480, Roche, USA) with Roche Applied Science SYBR Master Mix. The specific primer sequences used in the current study are listed in Supplementary Table 2.

## Mass spectrometry analysis

Mass spectrometry was conducted by PTM BIO (Hangzhou, China) to analyse differentially expressed proteins in young and old mice ($n = 3$ per group, C57BL/6 background. Young, 10 weeks, male; Old, 20 months, male) during HIRI. Protein Extraction: The frozen fresh tissue was ground into a cell powder with liquid nitrogen, and then, four volumes of lysis buffer (8 M urea, 1% protease inhibitor cocktail, 3 µM TSA and 50 mM NAM for acetylation, 1% phosphatase inhibitor for phosphorylation) were added to the cell powder, followed by sonication three times on ice using an ultrasonic processor (Scientz). After centrifugation at 12,000 × g at 4 °C for 10 min, the supernatant was collected, and the protein concentration was determined by a BCA kit according to the manufacturer's instructions. Trypsin Digestion: The protein solution was reduced with 5 mM dithiothreitol for 30 min at 56 °C and alkylated with 11 mM iodoacetamide for 15 min at room temperature in darkness, then diluted by adding 100 mM TEAB to urea concentration less than 2 M. Trypsin was added at 1:50 ratio (trypsin-to-protein mass) for the first digestion overnight and 1:100 ratio (trypsin-to-protein mass) for a second 4 h-digestion. Finally, the

peptides were desalted by C18 SPE column. LC-MS/MS Analysis: Using 4D-Label free Mass Spectrometer, the tryptic peptides were dissolved in solvent A (0.1% formic acid, 2% acetonitrile in water), directly loaded onto a home-made reversed-phase analytical column (25-cm length, 75/100 μm i.d.). Peptides were separated with a gradient from 6% to 24% in solvent B (0.1% formic acid in acetonitrile) over 70 min, 24% to 35% within 14 min, climbing to 80% within 3 min then holding at 80% for the last 3 min, all at a constant flow rate of 450 nL/min on a nanoElute UHPLC system (Bruker Daltonics). An electrospray voltage of 1.60 kV was applied to the peptides before capillary source and TIMsTOF Promass spectrometry (Bruker Daltonics). In parallel accumulation serial fragmentation mode (PASEF), the TOF detector analyzed precursors and fragments in the range of 100 to 1700 m/z. Precursors with charge states 0 to 5 were selected for fragmentation, and 10 PASEF-MS/MS scans were acquired per cycle. The dynamic exclusion was set to 30 s. Database Search: The resulting MS/MS data were processed using MaxQuant search engine (v.1.6.15.0). Tandem mass spectra were searched against the human SwissProt database (20422 entries) concatenated with reverse decoy database. Trypsin/P was specified as cleavage enzyme allowing up to 2 missing cleavages. The mass tolerance for precursor ions was set as 20 ppm in first search and 5 ppm in main search, and the mass tolerance for fragment ions was set as 0.02 Da. Carbamidomethyl on Cys was specified as fixed modification, and acetylation on protein N-terminal and oxidation on Met were specified as variable modifications. FDR was adjusted to <1%. Data analysis: To test the quantitative results of biological or technical replicate samples are statistically consistent, Pearson's Correlation Coefficient (PCC), principal component analysis (PCA) and relative standard deviation (RSD) were used. In order to identify differential proteins, the Fold Change (FC) was calculated as the ratio of the mean relative quantitative values of each protein in repeated samples. A threshold of significance for up-regulation was set at a fold change greater than 1.5 with a $P$ value less than 0.05, while a threshold for down-regulation was set at a fold change less than 1/1.5. The differentially expressed proteins in different comparison groups were enriched by GO classification, KEGG pathway and protein domain, and analysed by Venn Diagram package. According to the Fisher exact test $P$ value obtained by enrichment analysis, the relevant functions in different comparison groups were clustered together by hierarchical clustering method.

## Lipidomics

Liver samples from eight young mice (C57BL/6 background, 2 months, male) and eight older mice (C57BL/6 background, 20 months, male) were removed and immediately frozen in liquid nitrogen until analysis. Lipids were extracted according to MTBE method[53]. Briefly, samples were first spiked with appropriate amount of internal lipid standards and then homogenized with 200 μL water and 240 μL methanol. After that, 800 μL of Tert-butyl Methyl Ether (MTBE, 650560-1 L, Sigma-Aldrich) was added and the mixture was ultrasound 20 min at 4 °C followed by sitting still for 30 min at room temperature. The solution was centrifuged at 14,000 × $g$ for 15 min at 10 °C and the upper organic solvent layer was obtained and dried under nitrogen. Then the solution was proceeded to LC-MS/MS for lipid analysis[54]. Reverse phase chromatography was selected for LC separation using CSH C18 column (1.7 μm, 2.1 mm × 100 mm, Waters). The lipid extracts were re-dissolved in 200 μL 90% isopropanol/ acetonitrile, centrifuged at 14,000 × $g$ for 15 min, finally 3 μL of sample was injected. Solvent A was acetonitrile–water (6:4, v/v) with 0.1% formic acid and 0.1 mM ammonium formate and solvent B was acetonitrile–isopropanol (1:9, v/v) with 0.1% formic acid and 0.1 mM ammonium formate. The initial mobile phase was 40% solvent B at a flow rate of 300 μL/min. It was held for 3.5 min, and then linearly increased to 75% solvent B in 9.5 min, and then linearly increased to 99% solvent B in 6 min, followed by equilibrating at 40% solvent B for 5 min. Mass spectra was acquired by

Q-Exactive Plus in positive and negative mode, respectively[55]. ESI parameters were optimized and preset for all measurements as follows: Source temperature, 300 °C; Capillary Temp, 350 °C, the ion spray voltage was set at 3000 V, S-Lens RF Level was set at 50% and the scan range of the instruments was set at m/z 200–1800. Lipid species based on MS/MS math were identified by "Lipid Search" engine[56]. Both mass tolerance for precursor and fragment were set to 5 ppm. The lipidomics data were evaluated and visualized using ggplot2, scatterplot3d and Venn Diagram package. Mfuzz software fuzzy c-means (FCM) algorithm for analysis, Mfuzz R package. Data from biological samples were normalized to the internal standard and all data are presented as the mean ± SD of multiple samples from different animals. The differential analysis of critical esterified lipids between young and older liver tissue was using two-tailed $t$-test. The complete lipid composition data are provided in the Source data file.

## Methylated RNA immunoprecipitation sequencing (MeRIP-seq)

Total RNA was isolated and purified as previously described. Then, MeRIP-seq was performed by LC-BIO (Hangzhou, China). In brief, the integrity of the RNA (RIN) was over 7.0, which was confirmed with denaturing agarose gel electrophoresis. Poly(a) RNA was purified from 30 μg total RNA using Dynabeads Oligo (dT) 25-61005 (ThermoFisher, USA) and fragmented into small pieces using Magnesium RNA Fragmentation Module (NEB, USA) at 86 °C for 7 min. Then, the cleaved RNA fragments were incubated for 2 h at 4 °C with a m6A-specific antibody (Synaptic Systems, Germany) in IP buffer (50 mM Tris-HCl, 750 mM NaCl and 0.5% Igepal CA-630). The IP RNA was reverse-transcribed to cDNA, which was next used to synthesize U-labelled second-stranded DNA. Each strand was then ligated to fragment DNAs with A-tails by adding an A-base to the blunt ends. A single- or dual-index adapter was ligated to the fragment, and AMPureXP beads were used to select the size. After treating the U-labelled second-stranded DNA with sheat-labile UDG enzyme (NEB, USA), the products were amplified with PCR. Finally, 2 × 150 bp paired-end sequencing (PE150) was performed on an Illumina NovaSeq™ 6000.

## m6A MeRIP-qRT–PCR

The MeRIP assay was performed with a riboMeRIP m6A Transcriptome Profiling Kit (C11051-1, RiboBio, Guangzhou, China) to assess the m6A modification of certain transcripts. After THLE-2 cells were transfected with FTO for 48 h, at least 100 μg of total RNA was extracted using TRIzol reagent. The whole process was RNA/DNase free, and the RNA was quantified with a Nanodrop One Microvolume UV–Vis Spectrophotometer (Thermo Fisher, USA). One hundred micrograms of total RNA were randomly fragmented into products that were 200 or fewer nucleotides in size at 70 °C for 7 min, and then, the fragmented RNA was precipitated with 3 M sodium acetate and absolute ethanol overnight. Magnetic beads A/G were prepared by 30 min of incubation with an m6A-specific antibody in immunoprecipitation buffer at room temperature. After the RNA was purified and recovered, one-tenth of the fragmented RNA was saved as a standardized control of input, and the rest was incubated with the MeRIP reaction mixture containing the prepared beads overnight with rotation at 4 °C. To elute RIP-RNA, the beads were incubated in 100 μl of elution buffer twice at 4 °C in a thermoshaker (1100 rpm) for 1 h. Input and RIP samples were finally purified using the miRNeasy Mini Kit (50) (217004, QIAGEN, Germany). Further m6A enrichment was analysed using qRT–PCR, and the results were normalized by the input. The primers are listed in Supplementary Table 2.

## Crosslinking-immunoprecipitation and high-throughput sequencing (CLIP-Seq)

At least 2 × 10⁷ cells are needed for each repeated experiment. THLE-2 cells were transfected with Vector or FTO plasmid for 48 h. Then the cells were washed with cold phosphate-buffered saline (PBS) and

irradiated twice at UV254 (600 mJ/cm²) in a Stratalinker crosslinker (Stratagene) on ice. Cells were collected with CLIP lysis buffer containing 0.5 mM DTT and EDTA-free protease inhibitor cocktail and incubated on ice for 20 min then centrifuged at 14,000 × *g* for 15 min at 4 °C and transferred supernatant into a new centrifuge tube. For immunoprecipitation procedure, the supernatant was incubated with FTO antibody (Cell Signaling Technology, #31687) or IgG antibody overnight at 4 °C. The next day, the system was further incubated with Protein A/G magnetic beads for 4 h at 4 °C. Afterwards, the beads were washed twice with IP wash buffer (50 mM HEPES-KOH, 300 mM KCl, 0.05% (v/v) NP40, 0.5 mM DTT, complete EDTA-free protease inhibitor cocktail (Roche), pH 7.5) and then incubated at 37 °C for 30 min and at ice for 5 min. To elute protein-RNA adducts, beads were resuspended in NuPAGE SDS-PAGE loading buffer and incubated at 72 °C for 10 min, and then separated on Novex NuPAGE 10% Bis-Tris gels (Invitrogen) and transferred to nitrocellulose membranes (Amersham). For extraction of RNA-protein complexes, the membrane was treated with proteinase K buffer (100 mM Tris pH 7.5, 10 mM EDTA, 50 mM NaCl, 1% SDS, 4 mg/ml proteinase K) at 55 °C for 30 min with constant agitation. An equal volume of phenol: chloroform: isoamylalcohol (125:24:1) to RNA was added and precipitated with 1 μl glycogen at −20 °C for 6 h and then centrifuged at 19,000 × *g* for 30 min at 4 °C then discarded the supernatant. The precipitate was washed twice with 75% ethanol and resuspended with 10 μl RNase-free water. The recovered RNA was used to perform high-throughput sequencing using Illumina P5 adapter and cDNA libraries were then sequenced by using DNBSEQ-T7 sequencer (MGI Tech Co., Ltd. China) under the help of Seqhealth Technology Co., LTD (Wuhan, China).

### mRNA stability assays

The indicated cells were cultured in 12-well plates followed by FTO overexpression for 24 h. Then, actinomycin D (MCE, HY-17559) was added to each well at a final concentration of 5 μg/ml, and the cells were collected after 0, 3, 4, 6, 8, and 12 h of incubation. Total RNA was isolated and subsequently subjected to qRT–PCR to quantify the relative abundance of *Acsl4* and *Tfrc* mRNA (relative to 0 h).

### RNA Immunoprecipitation (RIP)

To demonstrate the binding of m6A reader proteins to *Tfrc/Acsl4* mRNA, THLE-2 cells were transfected with indicated plasmids for 48 h. Then the cells were harvest on ice using IP lysis buffer (P0013, Beyotime, Shanghai, China) supplemented with RNase Inhibitor and Protease Inhibitor Cocktail (HY-K1033, MCE, NJ, USA). Add YTHDF2 or YTHDC2 antibody (CST, Danvers, MA, USA) to pre-washed Protein A/G magnetic beads (Thermo Fisher, USA) and incubate with rotation for 2 h at room temperature. Using 1/10 cell lysates as input, the remaining samples were subsequently incubated with magnetic beads conjugated to YTHDF2 or YTHDC2 antibody at 4 °C overnight. The Co-Immunoprecipitated RNAs were eluted using TRNzol reagent (Thermo Fisher, USA), and the relative enrichment of *Acsl4* and *Tfrc* mRNA were detected by RT-qPCR. GAPDH was used as negative control. The catalog numbers of the antibodies are listed in the Reporting Summary.

### RNA m6A dot blotting assays

Total RNA was extracted from in vitro and in vivo samples for dot blotting assays. RNA samples were diluted with nuclease-free water to 250 ng and subsequently denatured at 95 °C within 5 min. Then, the samples were spotted onto an Amersham Hybond-N+ membrane (GE Healthcare, USA). After two rounds of 2400 UV crosslinking (5 min each time), the membrane was washed with 0.1% TBST (compounded by DEPC water), blocked with 5% nonfat milk in 0.1% TBST, and incubated with an anti-N6-methyladenosine (m6A) antibody (1:1000; Cell Signaling Technology, CA, USA) overnight at 4 °C. After washing with 0.1% TBST, horseradish peroxidase-conjugated anti-rabbit immunoglobulin G (Cell Signaling Technology, CA, USA) was diluted 1:5000

and incubated with the membranes for 1 h at room temperature. Finally, the membranes were visualized by the ECL Western Blotting Detection Kit (201005-79, Advansta, CA, USA). To determine the total amount of input RNA, the membrane was stained with 0.02% methylene blue (M4159, Sigma–Aldrich, USA) in 0.3 M sodium acetate (pH 5.2) before blocking. Densitometric analysis of m6A normalized by MB was conducted by Image J. (ImageJ software, USA). The catalog numbers of the antibodies are listed in the Reporting Summary.

### Statistical analysis

All the statistical analyses and plots were conducted using GraphPad Prism 8 software (GraphPad Software, San Diego, CA). Single comparisons were made using unpaired two-tailed *t*-tests. One-way analysis of variance (ANOVA) was used to analyze the statistical significance between the groups. As appropriate, all the values are expressed as the value directly or as the mean ± standard deviation (SD). Biological replicates times are indicated in the figure legends. *P* values < 0.05 were considered statistically significant. The specific *P* values are labeled above the error bar.

### Reporting summary

Further information on research design is available in the Nature Portfolio Reporting Summary linked to this article.

## Data availability

The MeRIP-seq raw data generated in this study have been deposited in the Genome Sequence Archive (GAS) -Human database under accession code HRA006502. The raw data of FTO-CLIP-seq in this study have been deposited in the Genome Sequence Archive (GAS) -Human database under accession code HRA006446. The mass spectrometry proteomics data have been deposited to the ProteomeXchange Consortium via the iProx partner repository with the dataset identifier PXD051387. The Lipidomics data generated in this study are provided in the Supplementary Information and Source Data file. The remaining data are available within the Article, Supplementary Information or Source Data file. Source data are provided with this paper.

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

## Acknowledgements

This work was supported by The National Key Research and Development Program of China (2017YFA0104304); National Natural Science Foundation of China (81802897, 81900597, 81901943, 81972286, 81970567, 82202191); Natural Science Foundation of Guangdong

Province (2021A1515011156, 2022A1515012331, 2021A1515012136, 2019B020236003, 2021A1515010571, 2018A030313043); the Operating Foundation of Guangdong Provincial Key Laboratory of Liver Disease Research (2020B1212060019); Guangzhou Basic and Applied Basic Research Foundation (202102020237); Guangdong Basic and Applied Basic Research Foundation (2021A1515111058, 2022A1515110316); Science and Technology Program of Guangzhou City (201803040005, 202201020398); Guangzhou Basic and Applied Basic Research Project Co-funded by Municipal Schools (institutes) (2023A03J0727, 2023A1515010346); China Postdoctoral Science Foundation (2022M713621, 2022M713617); National Bioengineering Research Center Cultivation Platform (WW201905) and Major talent project cultivation plan project (P02095, P02093).

## Author contributions

Rong Li, Xijing Yan, Cuicui Xiao, Tingting Wang, Xuejiao Li, Zhongying Hu, Qiuli Liu, Manli Wu: Conceptualization, Methodology, Validation, Formal analysis, Data Curation, Writing - Original Draft. Jiaqi Xiao, Haitian Chen, Yasong Liu, Chenhao Jiang, Guo Lv: Software, Validation, Visualization. Jinliang Liang, Jiebin Zhang, Jianye Cai, Xin Sui: Formal analysis, Investigation, Data Curation. Guihua Chen and Jia Yao: Writing –Review & Editing, Supervision. Rong Li, Yingcai Zhang, Jun Zheng and Yang Yang: Conceptualization, Resources, Writing-Review & Editing, Supervision, Project administration, Funding acquisition.

## Competing interests

The authors declare no competing interests.
