## [Peer Review File · Nature Communications]

FTO deficiency in older livers exacerbates ferroptosis during ischaemia/reperfusion injury by upregulating ACSL4 and TFRCREVIEWER COMMENTS

Reviewer #1 (Remarks to the Author):

In the present study, Li et al attempts to address the mechanisms underlying the increased severity of liver damage in hepatic ischaemia/reperfusion injury during aging. The authors found stronger lipid peroxidation signals using BODIPY-C11 staining in aged livers, and proposed that aged livers are more susceptible to ferroptosis. Using proteomic analysis, the authors found that expression of FTO, an mRNA N(6)-methyladenosine demethylase, is changed by hepatic ischaemia/reperfusion only in aged livers. Dot blot assays confirmed the increased m6A levels during hepatic ischaemia/reperfusion injury, and this difference is more prominent in aged livers. The authors used FB23-2, a small molecule inhibitor of FTO's demethylase activity, to confirm that FTO inhibition further aggravated aged hepatic ischaemia/reperfusion injury. Moreover, using methylated RNA immunoprecipitation sequencing (MeRIP-seq) analysis in senescent THLE2 cells overexpressing FTO and treated with hypoxia/reoxygenation, and focused on two FTO target genes, ACSL4 and TFRC, for characterizing their involvement in liver damage. Finally, the authors used NMN supplementation to boost NAD biosynthesis and enhance the demethylase activity of FTO, revealing that NMN treatment could strongly ameliorate aged hepatic ischaemia/reperfusion injury. Overall, the question this study focuses on addressing has important clinical significance. However, there are some key questions that the authors may consider addressing:

1. Are the BODIPY-C11 positive cells also TUNEL positive cells? Did the authors characterize the involvement of other cell death pathways such as apoptosis (cleaved-Caspase 3) or necroptosis (p-MLKL)? Did the authors attempt to image BODIPY-C11 signal without PFA fixation to reduce alterations in the tissue metabolites and fluorescent intensity?
2. A key question of this study is related to the selection of ACSL4 and TFRC as functional mediators of FTO's effect on liver damage and cell death. The authors used pathway analysis to help focusing on known ferroptosis-related genes, disregarding other genes that could be more prominent target genes of FTO. How do those genes contribute to hepatic ischaemia/reperfusion injury? Could some of these genes even be novel regulators of liver ferroptosis?
3. If the major difference between young and old liver tissues includes ACSL4 expression, are there key differences in the polyunsaturated-lipidome during aging? The authors may consider performing lipidomics/metabolomics experiments to dissect the lipidome compositions of liver tissues.
4. The TEM images acquired to observe the mitochondrial ultrastructures are generally quite unclear, and it is very difficult to draw conclusions based on the presented images.

Suggestions:

1. Consider merging some of the main figures, since in Figure 1,2,4,5, the experiments are quite similar except adding one additional treatment condition each time. It will also help to shorten the text descriptions, which is quite long in the current version.

2. When citing the figures in the text, the authors frequently cites a large collection of figures together without detailed description of the results, e.g. Line 180, Fig. 3f-3i; Line 208, Fig. 4h-4n. The authors should make the text more concise but still describe each individual result in more detail.

Minor comments:

Details on statistical analysis should be more clearly stated in the Methods and Figure legends.

Figure 3b: lack of explanation for the abbreviation

Figure 3d: add table

Figure 3f: very weak control band

Figure 4j-k: uneven staining of reduced BODIPY-C11

For all figures with quantitation, y-axis label is percent. Should this be in different unit?

Line 160: "lipid" should be "liquid"

Lack of explanation on TEM analysis

Line 173, add FTO citations

Reviewer #2 (Remarks to the Author):

In this paper, Li et al. demonstrate that decreased expression of FTO in aged liver causes the exacerbation of ferroptosis at HIRI. Although the involvement of ferroptosis and FTO in the aggravation of HIRI has been already reported by the other groups, the authors found that hepatic FTO expression decreased with age by LC-MS/MS analysis and that deficiency of FTO in aged liver could affect m6A modification of UTR of ACSL4 and TFRC, two key molecules for ferroptosis accompanied with the upregulation of these gene products. These findings are novel and interesting, accounting for the reason why aged liver is more susceptible to HIRI and/or ferroptosis compared to young one. Conversely, overexpression or activation of FTO in hepatocytes are shown to ameliorate signs of ferroptosis and HIRI in vivo and in vitro. These experiments and presented data are generally well done and sound.

Comment to the authors:

1. In this paper, TUNEL assay are used for the detection of cell death in liver at IRI. However, it is not generally used for the detection of ferroptosis but apoptosis. Does this image mean the severity of liver injury or a total number of cell death including apoptosis and ferroptosis? On the other hand, I think that nuclear staining with PI or SYTOX Green etc. will distinguish ferroptosis from apoptosis clearly in the in vitro assay.
2. The value of Y axis in many graphs are represented as “percent”. I think it is a mistake since percent means the value in one hundred. The terms “ratio” or “rate” would be better.
3. In Figure 3j, the levels of m6A methylation seemed not to be upregulated in old samples at a glance. Densitometric analysis of M6A normalized by MB with a software such as Image J may demonstrate a significant difference between old and young samples. Also in Figure 8d, such an analysis is recommended since this data are important to demonstrate that NMN actually regulate m6A demethylase activity in hepatocytes. Even though NMN ameliorated IRI, NAD⁺ could inhibit ferroptosis by affecting mitochondrial function directly.
4. Regarding the processing of photo information, I wonder how the ratio is calculated. How many photos are used for the average that corresponds to n=1? Or is each dot derived from one photo? As the authors know, one photo with high magnification does not always represent the status of whole liver. In addition, I want to know how the authors calculate the percent of oxidized lipid in DODIPY C11 assay. The authors showed both oxidized lipids (green) and non-oxidized lipids (magenta) in the same view. Is the non-oxidized image information utilized for the calculation of ratio (e.g. used for the calculation of total lipid area) or shown as only reference? Otherwise, does the presented graph mean the ratio of green signals per a total view area? Please make it clear.

5. Figure 4n, S3e and Line 206: The authors mention “which was accompanied by the upregulation of ferroptosis-inhibiting molecules” by referring to the data of Figure 4n and S3e. However, the presented bands of WB seem to be comparable between Vector and FTO. Is there any other evidence that FTO is involved in the upregulation of these molecules via m6A demethylation?

6. Figure 5a,b,d,f and j: Although the comparison between IRI+Fer-1 and FB23-2+Fer-1 represents NS, these graphs show a clear tendency of higher value in FB23-2+Fer-1 samples. If Fer-1 worked to inhibit the final production of lipid peroxides, and FTO-mediated ACSL4 and TFRC worked at the upstream side from the accumulation of lipid peroxides, no difference between two values would be expected. In other words, there is a possibility that FTO may play a role in the other ferroptosis pathway as well as lipid peroxidation. How do the authors think about it?

7. Figure 8a: Serum ALT seems to be suppressed almost completely by only shACSL4. However, FB23-2 worsen ALT under shACSL4 condition. There is no discussion about this result. Does this result mean that increased part of ACSL4 by FB23-2 cannot be suppressed by shACSL4? Otherwise, although shACSL4 could suppress ACSL4, is there other ferroptosis promoting molecule by FB23-2 involved in the increase of ALT? WB data about ACSL4 and TFRC in Figure 7d would make it clear which possibility is plausible.

8. If FTO is generally downregulated and thereby ACSL4 is upregulated in aged liver, it follows that the ratio of phospholipids including PUFA is higher in aged livers than young ones. If there is such a report, please add it to references. In addition, what is the mechanism underlying the decline of FTO in liver with age?

Reviewer #3 (Remarks to the Author):

This manuscript discusses how the FTO regulated hepatic ischemia-reperfusion injury (HIRI) in aged liver by targeting ferroptosis inducers ACSL4 and TFRC. In line with higher susceptibility to HIRI in aged liver, they found aged liver were characterized by a higher intensity of ferroptosis than young livers, and inhibiting ferroptosis can significantly alleviates aged HIRI. What's more, they found that aged livers had lower FTO expression and higher m6A methylation levels. FTO-mediated demethylation was found to shorten the half-life of ACSL4 and TFRC, two important inducers for ferroptosis, to regulate ferroptosis in aged HIRI.

The author investigated how FTO influence aged HIRI by regulating ferroptosis. However, the regulation of ferroptosis by FTO(ref 1,2) has been studied, and the link between HIRI and ferroptosis (ref3) has also been established previously. Therefore, it is not surprising that FTO affects HIRI through ferroptosis. A more interesting question is that, why FTO was down-regulated in aged livers. What's more, the authors

show that FTO will be down-regulated after IR both in young and old livers. Therefore, exploring the regulatory mechanisms of FTO down-regulation becomes even more intriguing.

Another major concern is about the validity of their MeRIP-seq analysis. It appears that the analysis lacks replicates and an input control (Figure 6e), and there is also a notable absence of enrichment scores (such as P-value or fold change) and rankings for motifs(Figure 6C).

Overall, the aforementioned concerns limit the impact and validity of the manuscript.

Major point:

1. In the manuscript, the authors did not adequately utilize HPLC to explain why the aging liver is more susceptible to ischemia-reperfusion injury (Figure 3). I would like the authors to employ HPLC for additional analyses, such as enrichment analysis, to elucidate the differential responses observed during IRI in young and aged livers. This would provide a more comprehensive understanding of the mechanisms underlying the increased susceptibility of the aging liver to IRI.
2. The authors focused on the differential expressed protein in comparison groups (Figure 3E). However, it is essential for them to clearly specify whether this analysis is conducted specifically on up-regulated proteins or down-regulated proteins.
3. The western blot in Figure 3f indicates that FTO was down-regulated after IR in both young and old livers. However, the HPLC analysis in Figure 3d demonstrates that FTO was only down-regulated in old livers. These two results appear to be somewhat contradictory. Therefore, it is crucial for the authors to clearly present and discuss the HPLC data regarding FTO expression to reconcile the observed differences.
4. The author presented two presentative motifs in Figure 6C. However, it appears that the motif-1 does not correspond to canonical m6A motif. Furthermore, it is important to include enrichment scores, such as p-values and fold changes, for the identified motifs in the MeRIP-seq analysis. Providing these additional metrics would enhance the interpretation and significance of the identified motifs.
5. Although the author performs function enrichment on differential m6A peaks (Figure 6d), it is not clearly specified whether this analysis is conducted specifically on up-regulated peaks, down-regulated peaks.
6. Given that MeRIP-seq signal is positively correlated with gene expression, it is essential to include replicates samples and input control to ensure the reliability of the finding. If there are replicates and input control, there are recommended to presented in the IGV(Figure 6E).
7. In addition to MeRIP-seq, it is important to obtain additional evidence, such as FTO CLIP-seq, to confirm the direct targeting of ACSL4 and TFRC by FTO.
8. The authors observed that FTO inhibits the stability of ACSL4 and TFRC mRNAs by demethylation. To strengthen this conclusion, it is necessary for the authors to present supplementary evidence that confirms the participation of m6A reader proteins in the regulation of ACSL4 and TFRC.

Minor points:

1. The spelling of "ischaemia" or "ischemia" should be consistent throughout the text
2. The intensity of the β -actin signal appears inconsistent across different groups (Figure 3f, bottom)
3. The author needs to improve the annotation of the Figure, for example, Figure x should be annotated on the page
4. The author should pay attention to the writing standards for protein and mRNA labeling.

Reference

1. Huang, W.-M. et al. m6A demethylase FTO renders radioresistance of nasopharyngeal carcinoma via promoting OTUB1-mediated anti-ferroptosis. *Transl. Oncol.* 27, 101576 (2023).
2. Ji, F.-H., Fu, X.-H., Li, G.-Q., He, Q. & Qiu, X.-G. FTO Prevents Thyroid Cancer Progression by SLC7A11 m6A Methylation in a Ferroptosis-Dependent Manner. *Front. Endocrinol.* 13, 857765 (2022).
3. Zhu, L. et al. The Emerging Role of Ferroptosis in Various Chronic Liver Diseases: Opportunity or Challenge. *J. Inflamm. Res.* Volume 16, 381–389 (2023).

Reviewers 1:

In the present study, Li et al attempts to address the mechanisms underlying the increased severity of liver damage in hepatic ischaemia/reperfusion injury during aging. The authors found stronger lipid peroxidation signals using BODIPY-C11 staining in aged livers, and proposed that aged livers are more susceptible to ferroptosis. Using proteomic analysis, the authors found that expression of FTO, an mRNA N(6)-methyladenosine demethylase, is changed by hepatic ischaemia/reperfusion only in aged livers. Dot blot assays confirmed the increased m6A levels during hepatic ischaemia/reperfusion injury, and this difference is more prominent in aged livers. The authors used FB23-2, a small molecule inhibitor of FTO's demethylase activity, to confirm that FTO inhibition further aggravated aged hepatic ischaemia/reperfusion injury. Moreover, using methylated RNA immunoprecipitation sequencing (MeRIP-seq) analysis in senescent THLE2 cells overexpressing FTO and treated with hypoxia/reoxygenation, and focused on two FTO target genes, ACSL4 and TFRC, for characterizing their involvement in liver damage. Finally, the authors used NMN supplementation to boost NAD biosynthesis and enhance the demethylase activity of FTO, revealing that NMN treatment could strongly ameliorate aged hepatic ischaemia/reperfusion injury. Overall, the question this study focuses on addressing has important clinical significance. However, there are some key questions that the authors may consider addressing:

1. Are the BODIPY-C11 positive cells also TUNEL positive cells? Did the authors

characterize the involvement of other cell death pathways such as apoptosis (cleaved-Caspase 3) or necroptosis (p-MLKL)? Did the authors attempt to image BODIPY-C11 signal without PFA fixation to reduce alterations in the tissue metabolites and fluorescent intensity?

Response: Thanks for your kind suggestion. This is an essential point for our study. Ischaemia/reperfusion injury (IRI) generally refers to the injury of organs or tissues subjected during the process of ischaemia and reperfusion. Previous studies demonstrated that multiple cell death patterns are involved in this process, such as apoptosis, necroptosis, pyroptosis, and ferroptosis, which is closely related to the specific context of the organs or tissues. As you are concerned, it is very important for us to identify that apart from ferroptosis whether other types of cell death (such as apoptosis, necroptosis) also participated in the aged HIRI and why we chose ferroptosis for further investigation. Until now, there are still rare studies on cell death associated with ischaemia / reperfusion injury in aging liver. Ferroptosis is a recently identified form of cell death that is mediated by iron-dependent phospholipid peroxidation, and it has been reported to participate in the occurrence and development of multiple diseases. Due to aged livers are more susceptible to lipid disorders and mitochondrial failure as well as iron ions further accumulate in the liver with age, we speculated that aged livers are more prone to ferroptosis. In light of the increasing proportion of elderly livers in the donor pool and their heightened vulnerability to IRI, it is of great significance to elucidate the cell death pattern involved in IRI of aged livers for the mitigation of HIRI.

To reveal this question, we, as suggested, detected the changes of core proteins

related to ferroptosis (NRF2, GPX4, FTH1, CISD1, ACSL4, and TFRC), apoptosis (BCL-2, BAX, Caspase-3, and PARP), pyroptosis (RIP3 and MLKL), and necroptosis (NLRP3, GSDMD, Caspase-1, IL-1 β , and IL-18) with human liver tissues collected from before and after reperfusion during liver transplantation at different ages via western blotting assays. As shown in revised Fig. 1e and revised Supplementary Fig. 1a-1c, ferroptosis, apoptosis, pyroptosis, and necroptosis all participated in the process of ischaemia/reperfusion to varying degrees. Compared with other types of cell death, the alteration of ferroptosis was more significant in both young and aged livers during the process of ischaemia/reperfusion, and the degree of ferroptosis in aged livers was more serious. In addition, we also examined the changes of different cell death patterns in liver tissues collected from young/aged mouse model of HIRI. As shown in revised Fig. 1l and revised supplementary Fig. 1d-1f, all four types of cell death were involved in mouse HIRI to varying degrees and the liver injury was more severe in aged mice, which were consistent with the results obtained from human liver tissues. Combined with the above results and the physiological characteristics of aged livers, we selected ferroptosis for further investigation.

In this study, we performed TUNEL staining to reflect the injury of cells or tissues following our previous studies (*Biomaterials*. 2022 May;284:121486; *Adv Sci (Weinh)*. 2020 Aug 20;7(18):1903746; *Cell Death Dis*. 2020 Apr 20;11(4):256; *Aging (Albany NY)*. 2018 Aug 8;10(8):1902-1920). And we apologize for this during manuscript preparation, as TUNEL staining was mainly used to represent cell apoptosis in previous studies and ferroptosis is the key point of our research. It is accepted that PI or SYTOX

Green, two common nuclear dyes, can enter the cells and stain the nucleus when the cell membrane is damaged. When cells undergo apoptosis, the cell membrane is relatively intact and PI or SYTOX Green can't enter the cells. However, when cells suffer from ferroptosis, pyroptosis or other types of cell death, the cell membrane is damaged and PI or SYTOX Green can enter the cells, staining the nucleus. Calcin-AM can penetrate the membrane of living cells and is one of the most commonly used fluorescent probes for staining living cells. Calcin-AM/PI double staining can be used to display both living and death cells. Therefore, we performed Calcin-AM/PI double staining to evaluate the cell injury in *in vitro* assays and replaced the related TUNEL staining in the revised manuscript.

For the evaluation of ferroptosis in our study, we mainly used DHE staining, C11 BODIPY staining, transmission electron microscopy (TEM) assays, and western blotting assays to represent the ferroptosis in HIRI. According to the reviewer's suggestion, we co-staining C11 BODIPY and TUNEL with human liver tissues before and after reperfusion as well as mouse tissues collected from HIRI model. As shown in attached Fig. 1A-1B, red represented TUNEL positive cells and green represented cells with lipid oxidation. Relatively, the regions of cells with lipid oxidation were seemed larger than that of TUNEL positive cells and both regions had significant overlaps, especially in the human liver tissues.

In the C11 BODIPY staining assays, we used 4% PFA to fix the slides after incubating with C11 BODIPY Reagent according to the previous studies (*Cell Metab.* 2019 Jan 8;29(1):156-173.e10; *Autophagy.* 2020 Oct;16(10):1889-1904; *Redox Biol.*

2020 May:32:101483). And we also appreciate the reviewer's concern about whether PFA fixation will induce alterations in the tissue metabolites and fluorescent intensity of C11 BODIPY staining. Therefore, we additionally carried out C11 BODIPY staining with/without PFA fixation with continuous sections of liver tissue to evaluate the influence of PFA fixation on the results of C11 BODIPY staining. As shown in attached Fig. 1C, although PFA fixation could reduce the fluorescence intensity of non-oxidized lipids to some extent, the regions of oxidized lipids looked similar in PFA-fixation group and non-fixation group. Therefore, our study suggested that the oxidized lipids in tissues and with/without PFA fixation had relatively little impact on the results of C11 BODIPY assays.

Attached Figure 1

Attached Figure 1. (A-B) Representative images showing the distribution of TUNEL positive cells (red) and cells with oxidized lipids (green) in mouse (A) and human (B) liver tissues. (C) Representative images showing the effects of with/without PFA fixation on C11 BODIPY assays. Scale bar = 100µm.

2. A key question of this study is related to the selection of ACSL4 and TFRC as functional mediators of FTO's effect on liver damage and cell death. The authors used pathway analysis to help focusing on known ferroptosis-related genes, disregarding other genes that could be more prominent target genes of FTO. How do those genes contribute to hepatic ischaemia/reperfusion injury? Could some of these genes even be novel regulators of liver ferroptosis?

Response: Thanks for your kind question. And this is indeed an important point for our study. The reasons why we chose ACSL4 and TFRC as downstream targets of FTO for liver damage and cell death are mainly as follows: (1) Based on the results shown in Fig.1-4 of this study, we proved that aged livers were more prone to ferroptosis during IRI. FTO can effectively mitigate IRI by inhibiting ferroptosis. Therefore, when screening the potential downstream target genes of FTO, we mainly focused on the reported regulatory genes related to ferroptosis. (2) Based on the results of MeRIP for FTO, we identified a total of 5 ferroptosis related genes with significant alteration in m6A levels, including Acsl4, Tfr, Slc39a8, Gss, and Acsl3. Among them, the m6A modification of Acsl3 was significantly upregulated under FTO-overexpressing. Considering that when FTO is overexpressed, the m6A levels of its downstream targets should decrease, we focused our attention on Acsl4, Tfr, Slc39a8, and Gss (Fig. 5e and Attached Fig. 2A). Among them, only the regulation of FTO on Acsl4 and Tfr were successfully validated with subsequent meRIP-qPCR (Fig. 6f and Attached Fig. 2B). Therefore, we chose ACSL4 and TFRC for further investigation in our study.

In addition to the reported ferroptosis-related genes, whether other genes that

could be more prominent target genes of FTO and contribute to HIRI via regulating ferroptosis is a very valuable and meaningful question. According to the reviewer's suggestion, we screened the top 10 upregulated/downregulated genes with the most significant changes in m6A modification level based on the results of MeRIP for FTO, which were displayed in Attached Fig. 2C. Based on the function of FTO, we mainly focused on the top 10 downregulated genes, including FAM156B, SLC6A3, ARHGEF34P, THOC3, AC009962, B3GNT8, CLK3, IL32, MAP2K4, and AC018641. Among them, FAM156B was predicted to enable methylated histone binding activity (*Cancer Med.* 2021 Jul;10(14):4964-4976; *Biol Direct.* 2011 Jun 13:6:30); SLC6A3 was reported to mediate sodium- and chloride-dependent transport of dopamine and norepinephrine (also known as noradrenaline) and regulate the light-dependent retinal hyaloid vessel regression (*Nat Cell Biol.* 2019 Apr;21(4):420-429; *Gene.* 1999 Jun 11;233(1-2):163-70; *Mol Pharmacol.* 2003 Mar;63(3):653-8); THOC3, as the component of the THO subcomplex of the TREX complex, is thought to couple mRNA transcription, processing, and nuclear export, and is essential for the export of Kaposi's sarcoma-associated herpesvirus (KSHV) intron-less mRNAs and infectious virus production. (*Cell Death Dis.* 2023 Jul 27;14(7):475; *Gene.* 2020 Mar 30:732:144350); B3GNT8, Beta-1,3-N-acetylglucosaminyltransferase 8, plays a role in the elongation of specific branch structures of multiantennary N-glycans and has strong activity towards tetraantennary N-glycans and 2,6 triantennary glycans (*FEBS Lett.* 2005 Jan 3;579(1):71-8; *Glycobiology.* 2005 Oct;15(10):943-51); CLK3, a dual specificity kinase acting on both serine/threonine and tyrosine-containing substrates, could

phosphorylate serine- and arginine-rich (SR) proteins of the spliceosomal complex, which may be a constituent of a network of regulatory mechanisms that enable SR proteins to control RNA splicing and can cause redistribution of SR proteins from speckles to a diffuse nucleoplasmic distribution (*J Exp Med.* 2020 Aug 3;217(8):e20191779; *Cell Stem Cell.* 2018 Apr 5;22(4):575-588.e7); IL32, a cytokine that may play a role in innate and adaptive immune responses, induce various cytokines such as TNFA/TNF-alpha and IL8, and activate typical cytokine signal pathways of NF-kappa-B and p38 MAPK, which has been reported to play import roles in the progression of multiple cancers (*Haematologica.* 2022 Dec 1;107(12):2905-2917; *Cancer Immunol Res.* 2014 Sep;2(9):890-900); MAP2K4, a dual specificity protein kinase, acts as an essential component of the MAP kinase signal transduction pathway and has been reported to participate in multiple physiological and pathological process, including tumor (*Cell Res.* 2018 Jul;28(7):719-729); the functions of ARHGEF34P, AC009962 and AC018641 are still rarely reported. In general, whether the above genes are regulated by FTO and their roles in HIRI and ferroptosis are unclear, which are deserved further investigation in the future.

Attached Figure 2. (A) Integrated genome browser views show the m6A modifications of *Slc39a8* and *Gss* in the control and FTO groups. (B) MeRIP-qPCR quantitative analysis of the fold enrichment of *Slc39a8* and *Gss* m6A levels by immunoprecipitation with a specific m6A antibody in senescent THLE2 cells transfected with control and FTO (two-tailed *t*-test). (C) Volcano map (left) and Tables (right) showing the 20 genes with the most significant alterations in m6A modification under FTO overexpressing (10 upregulated and 10 down-regulated).

3. If the major difference between young and old liver tissues includes ACSL4 expression, are there key differences in the polyunsaturated-lipidome during aging? The authors may consider performing lipidomics/metabolomics experiments to dissect the lipidome compositions of liver tissues.

Response: Thank you for your valuable suggestion. This is an essential point for our study. As suggested, we performed targeted metabolomics study via Q300 Metabolomics chip in Metabo-Profile (Shanghai, China). Q300 Metabolomics chipTM technology focus ~300 targeted metabolites across all major functional metabolite classes such as amino acids and amines, organic acids, carbohydrates, fatty acids and lipids, nucleotides, sugars, vitamins, and co-factors, etc., measured by triple quadrupole mass spectrometer with an ultrahigh performance liquid chromatography.

We collected liver samples from 10 young mice (10 weeks) and 10 old mice (20 months) for targeted metabolomics analysis. The results of principal component analysis (PCA) showed that liver samples derived from young and old mice could basically be grouped independently, suggesting that there were indeed differences between the groups (Attached Fig. 3A). Subsequently, we analyzed the metabolic composition of liver tissues in different groups. As shown in Attached Fig. 3B, the proportion of fatty acids in liver samples of the elderly group increased significantly. We also displayed the differential expression of metabolites in young and old liver samples with heatmaps. As shown in Attached Fig. 3C, there were significant differences in the levels of multiple metabolites in young and old liver tissue. Compared with young liver tissues, lots of metabolites were low in old liver tissues, except

multiple fatty acids, which were highlighted in a red box. Based on the differential metabolites, functional enrichment analysis was performed. As shown in Attached Fig. 3D, the pathways with the highest enrichment scores were mainly focus on the fatty acid-related signaling pathways. A total of 25 fatty acids showed significant differences between young and old liver tissues. Among them, 19 were highly expressed in old liver tissues, and the other 6 were low expressed in old liver tissues (Attached Fig. 3E). We further analyzed the types of differential fatty acids and found that about 43% highly expressed in old liver tissues were poly-unsaturated fatty acids (Attached Fig. 3F). A detailed list of differential fatty acids has been shown in Attached Table 1. Taken together, the results of targeted metabolomics analysis suggested that poly-unsaturated fatty acids were significantly higher in old livers than that in young livers. Polyunsaturated fatty acids are the important substrates of ACSL4 (*Cell Res. 2018 Jul;28(7):719-729*), and our study also found that ACSL4 is highly expressed in old liver, further suggesting that aged liver is more prone to ferroptosis.

Attached Figure 3

Attached Figure 3. The Metabolomics profile of liver tissues between young and aged mice. (A) PCA showed both intragroup repeatability and intergroup variability. (B) Bar chart showing the composition of common metabolites in young and aged livers. (C) Heatmap showing differentially expressed metabolites in the different groups. (D) Enrichment analysis of differential metabolic pathways in young and aged livers. (E) Bar chart showing the statistic of differential fatty acids in young and aged livers. (F) Pie charts showing the detail composition of differential fatty acids.

Attached Table 1. Differential Fatty Acids between young and aged liver using Fold Change (FC) ≥ 1.1					
Class	HMDB	Metabolite	P-value	FC(OM/YM)	Types
Fatty Acids	HMDB0000847	Nonanoic acid	0.086449976	2.85	Medium-chain fatty acid
Fatty Acids	HMDB0001873	Isobutyric acid	0.000487129	2.092926491	Short chain fatty acid
Fatty Acids	HMDB0000207	Oleic acid	1.08E-05	1.972348687	Mono-unsaturated fatty acid
Fatty Acids	HMDB0003229	Palmitoleic acid	0.003886207	1.885591792	Mono-unsaturated fatty acid

Fatty Acids	HMDB0062248	Myristelaidic acid	1.08E-05	1.794871795	Mono-unsaturated fatty acid
Fatty Acids	HMDB0013622	Nonadeca_10Z_enoic acid	0.014689645	1.787224942	Mono-unsaturated fatty acid
Fatty Acids	HMDB0000673	Linoleic acid	1.08E-05	1.72604227	Poly-unsaturated fatty acid
Fatty Acids	HMDB0060038	10Z_Heptadecenoic acid	0.000725281	1.681465453	Mono-unsaturated fatty acid
Fatty Acids	HMDB0000529	5_Dodecenoic acid	0.017131827	1.581632653	Mono-unsaturated fatty acid
Fatty Acids	NA	Gamma_Linolenic acid	0.000487129	1.466145151	Poly-unsaturated fatty acid
Fatty Acids	HMDB0002183	Docosahexaenoic acid, DHA	0.014689645	1.43877303	Poly-unsaturated fatty acid
Fatty Acids	HMDB0001043	Arachidonic acid	0.018543376	1.409956754	Poly-unsaturated fatty acid
Fatty Acids	HMDB0001999	Eicosapentaenoic acid, EPA	0.075256013	1.340252238	Poly-unsaturated fatty acid
Fatty Acids	HMDB0002925	8,11,14_Eicosatrienoic acid	0.014689645	1.291791986	Poly-unsaturated fatty acid
Fatty Acids	HMDB0000826	Pentadecanoic acid	0.001050034	1.281845479	Long-chain fatty acid
Fatty Acids	HMDB0006528	Docosapentaenoic acid, DPA	0.005196042	1.252585083	Poly-unsaturated fatty acid
Fatty Acids	HMDB0002259	Heptadecanoic acid	0.023290235	1.249637725	Long-chain fatty acid
Fatty Acids	HMDB0034297	Ricinoleic acid	0.049195354	1.235294118	Mono-unsaturated fatty acid
Fatty Acids	HMDB0002226	Adrenic acid	0.063012839	1.197719053	Poly-unsaturated fatty acid
Fatty Acids	HMDB0031158	2_Methyl_4_pentenoic acid	0.039052775	0.684210526	Methyl-branched fatty acid
Fatty Acids	HMDB0000407	2_Hydroxy_3_methylbutyric acid	0.000328133	0.624209575	Hydroxy fatty acid
Fatty Acids	HMDB0000555	3_Methyladipic acid	0.031082168	0.469314079	Methyl-branched fatty acid
Fatty Acids	HMDB0000237	Propanoic acid	0.000324753	0.386490341	Short chain fatty acid
Fatty Acids	HMDB0000754	3_Hydroxyisovaleric acid	0.001003556	0.371841155	Short-chain hydroxy acid
Fatty Acids	HMDB0001901	Aminocaproic acid	0.000180635	0.122699387	Medium-chain fatty acid

Attached Table 1. Differential Fatty Acids between young and aged liver tissues

4. The TEM images acquired to observe the mitochondrial ultrastructures are generally quite unclear, and it is very difficult to draw conclusions based on the presented images.

Response: Thanks for your kind suggestion. And we apologize for the oversight in the preparation of TEM images. According to your suggestion, we have replaced all blurry TEM images with clearer ones in our revised manuscript.

Suggestions:

1. Consider merging some of the main figures, since in Figure 1,2,4,5, the experiments are quite similar except adding one additional treatment condition each time. It will also help to shorten the text descriptions, which is quite long in the current version.

Response: Thank you for the kind suggestion. This is an essential point for our study. In Section 1 (revised Fig. 1 and Supplementary Fig. 2), we intended to utilize clinical specimens combined with both *in vivo* and *in vitro* experiments to compare the degree of ferroptosis between aged HIRI and young HIRI, which showed that aged HIRI is more serious than the young. Moreover, we, according to the reviewer's suggestion, additionally detected the changes of core proteins related to ferroptosis, apoptosis, pyroptosis, and necroptosis to answer why we selected ferroptosis for further investigation and present that the alteration of ferroptosis was more significant in both young and aged HIRI compared to other types of cell death (revised Supplementary Fig. 1). And we utilized Ferrostatin-1 (Fer-1), a small molecular reagent for specially inhibiting the occurrence of ferroptosis in the *in vivo* and *in vitro* studies, to demonstrate that suppressing ferroptosis could remarkably alleviate aged HIRI in Section 2 (revised

Fig. 2). Therefore, we deemed that the gist of Section 1 is different from Section 2 leading us to retain Section 1 and Section 2 as well as Fig. 1 and Fig. 2 in the revised manuscript.

For Fig. 4 and Fig. 5, we would like to express our gratitude to the reviewer for informing us of the relevant concern. As suggested, we merged Fig. 4 and Fig. 5 and their relevant description in our revised manuscript, which were highlighted with yellow.

2. When citing the figures in the text, the authors frequently cites a large collection of figures together without detailed description of the results, e.g. Line 180, Fig. 3f-3i; Line 208, Fig. 4h-4n. The authors should make the text more concise but still describe each individual result in more detail.

Response: Thank you for the kind suggestion. We apologize for any oversight during manuscript preparation. As suggested, we made more detail description for Figure 3f-3i and Figure 4h-4n (revised Supplementary Fig. 4e-4k) in our revised manuscript, which were highlighted with yellow.

Minor comments:

Details on statistical analysis should be more clearly stated in the Methods and Figure legends.

Response: Thanks for your kind suggestion. We apologize for the oversight during the description of statistical analysis and we have further refined the relevant details in the

Methods and Figure legends in the revised manuscript according to your suggestion.

Figure 3b: lack of explanation for the abbreviation

Response: Thank you for the kind suggestion. And we apologize for any oversight during manuscript preparation. S0 represents the old-sham group, S12 represents the old-IRI group, Y0 represents the young-sham group and Y12 represents the young-IRI group. As suggested, we have added the explanation into the figure legends for Fig. 3b in the revised manuscript.

Figure 3d: add table

Response: Thanks for your kind suggestion. According to your suggestion, we listed the molecules corresponding to the bar charts in Fig. 3d and displayed them in revised supplementary materials (Table. S1). The corresponding description had been added into revised manuscript, which were highlighted in yellow. In addition, to present the overview of differential proteins more intuitively among the four groups, we added the detail numbers in Fig. 3d in the revised manuscript.

Figure 3f: very weak control band

Response: Thank you for the kind suggestion. We apologize for any oversight during manuscript preparation. As suggested, we repeated the western blotting assays for Fig. 3f and replaced low quality bands in the revised manuscript.

Figure 4j-k: uneven staining of reduced BODIPY-C11

Response: Thank you for the kind suggestion. We apologize for any oversight during manuscript preparation. As suggested, we have replaced Fig. 4j-k (revised Supplementary Fig. 4g-4h) with new C11 BODIPY staining images in the revised manuscript.

For all figures with quantitation, y-axis label is percent. Should this be in different unit?

Response: Thanks for your kind reminding. We apologize for the mistake in y-axis label. We have carefully modified the problematic Y-axis label and replaced “percent” with “ratio” or “rate”.

Line 160: “lipid” should be “liquid”

Response: Thanks for your kind reminding. We apologize for the mistake spelling of “liquid” and we have corrected it.

Lack of explanation on TEM analysis

Response: Thanks for your kind reminding. We apologize for the oversight during the description of TEM assays. Observation of the alteration in mitochondrial morphology by TEM assays can relatively intuitively reflect the situation of cells suffering ferroptosis, and is one of the main methods to reflect ferroptosis (*Redox Biol.* 2022 *Dec:58:102541*). When cells undergo ferroptosis, we can observe that the mitochondrial become smaller and the mitochondrial cristae decrease or disappear with

TEM. According to your suggestion, we have supplemented the description of results for TEM assays in the revised manuscript, which were highlighted in yellow. In addition, we marked ferroptosis with red arrows in all TEM images.

Line 173, add FTO citations

Response: Thanks for your kind suggestion. As suggested, we have added FTO citations in the revised manuscript, which were highlighted in yellow.

Reviewer 2:

In this paper, Li et al. demonstrate that decreased expression of FTO in aged liver causes the exacerbation of ferroptosis at HIRI. Although the involvement of ferroptosis and FTO in the aggravation of HIRI has been already reported by the other groups, the authors found that hepatic FTO expression decreased with age by LC-MS/MS analysis and that deficiency of FTO in aged liver could affect m6A modification of UTR of *Acs14* and *Tfrc*, two key molecules for ferroptosis accompanied with the upregulation of these gene products. These findings are novel and interesting, accounting for the reason why aged liver is more susceptible to HIRI and/or ferroptosis compared to young one. Conversely, overexpression or activation of FTO in hepatocytes are shown to ameliorate signs of ferroptosis and HIRI *in vivo* and *in vitro*. These experiments and presented data are generally well done and sound.

Comment to the authors:

1. In this paper, TUNEL assay are used for the detection of cell death in liver at IRI. However, it is not generally used for the detection of ferroptosis but apoptosis. Does this image mean the severity of liver injury or a total number of cell death including apoptosis and ferroptosis? On the other hand, I think that nuclear staining with PI or SYTOX Green etc. will distinguish ferroptosis from apoptosis clearly in the *in vitro* assay.

Response: Thank you for the kind question. Indeed, this is an essential point for our study. In this study, we performed TUNEL staining to reflect the injury of cells or tissues following our previous studies (*Biomaterials*. 2022 May;284:121486; *Adv Sci (Weinh)*. 2020 Aug 20;7(18):1903746; *Cell Death Dis*. 2020 Apr 20;11(4):256; *Aging (Albany NY)*. 2018 Aug 8;10(8):1902-1920). And DHE staining, C11 BODIPY staining, transmission electron microscopy (TEM) assays, and western blotting assays were herein mainly used to represent the ferroptosis in HIRI. Indeed, TUNEL staining is identified as a technique for detecting DNA fragmentation derived from cellular apoptotic death. Therefore, we apologize for making this confusion during manuscript preparation, as ferroptosis is the key point of our research. In addition, we would also like to express our gratitude to the reviewer for informing us of the relevant issue, as PI or SYTOX Green is commonly utilized for detecting cellular damage, which can enter the cells from the damaged cell membrane, staining the nucleus when the cells suffering from ferroptosis, pyroptosis, or other types of cell death. Calcin-AM can penetrate the membrane of living cells and is one of the most commonly used fluorescent probes for staining living cells. Calcin-AM/PI double staining can be used to display both living

and death cells. Therefore, we performed Calcein-AM/PI double staining to evaluate the cell injury in *in vitro* assays and replaced the related TUNEL staining in the revised manuscript.

2. The value of Y axis in many graphs are represented as “percent”. I think it is a mistake since percent means the value in one hundred. The terms “ratio” or “rate” would be better.

Response: Thanks for your kind suggestion. We apologize for the mistake in y-axis label. According to your suggestion, we have carefully modified the problematic Y-axis label and replaced “percent” with “ratio” or “rate” in the revised manuscript.

3. In Figure 3j, the levels of m6A methylation seemed not to be upregulated in old samples at a glance. Densitometric analysis of M6A normalized by MB with a software such as Image J may demonstrate a significant difference between old and young samples. Also in Figure 8d, such an analysis is recommended since this data are important to demonstrate that NMN actually regulate m6A demethylase activity in hepatocytes. Even though NMN ameliorated IRI, NAD⁺ could inhibit ferroptosis by affecting mitochondrial function directly.

Response: Thanks for your kind suggestion. This is an essential point for our study. As suggested, we used image J to scan the gray values of the dot blotting results in Fig. 3j-3k and Fig. 8d. Based on the gray values, we plotted corresponding bar charts and added them into revised Fig. 3j-3k and Fig. 8d (revised Fig. 7d), which further proved that the

levels of m6A methylation were upregulated in aged liver tissues and NMN could increase the deacetylase activity of FTO.

4. Regarding the processing of photo information, I wonder how the ratio is calculated. How many photos are used for the average that corresponds to $n=1$? Or is each dot derived from one photo? As the authors know, one photo with high magnification does not always represent the status of whole liver. In addition, I want to know how the authors calculate the percent of oxidized lipid in DODIPY C11 assay. The authors showed both oxidized lipids (green) and non-oxidized lipids (magenta) in the same view. Is the non-oxidized image information utilized for the calculation of ratio (e.g. used for the calculation of total lipid area) or shown as only reference? Otherwise, does the presented graph mean the ratio of green signals per a total view area? Please make it clear.

Response: Thanks for your kind suggestion. We apologize for the lack of a detail description of the method for analyzing C11 BODIPY assays. The representative images of C11 BODIPY staining and related quantitative bar charts were used to display the results of C11 BODIPY assays. In the bar charts, each point represented one sample and the related value for this sample was the average of the oxidized lipids ratios in five fields of view for this sample. The ratio of oxidized lipids = the number of cells with oxidized lipids/ the number of total cells, which is consistent with the calculation method used in previous studies (*Cell Death Differ.* 2022 Nov;29(11):2123-2136; *Nat Chem Biol.* 2023 Jan;19(1):28-37; *Nat Commun.* 2017 Mar 28;8:14844). Due to

oxidized lipids and non-oxidized lipids can exist in the same cell under some condition, the non-oxidized lipids (magenta) was not used to calculate the ratio of oxidized lipids, which was mainly used to reflect that the increase of oxidized lipids in the cell is usually accompanied by a decrease in non-oxidized lipids. The methods for calculating the ratio of oxidized lipids and drawing the corresponding quantitative bar chart had been added into the “Methods” section of revised manuscript, which were highlighted in yellow.

5. Figure 4n, S3e and Line 206: The authors mention “which was accompanied by the upregulation of ferroptosis-inhibiting molecules” by referring to the data of Figure 4n and S3e. However, the presented bands of WB seem to be comparable between Vector and FTO. Is there any other evidence that FTO is involved in the upregulation of these molecules via m6A demethylation?

Response: Thanks for your kind question. This is an essential point for our study. According to the reviewer’s suggestion, we rechecked the results of our original manuscript. Authentically, the presented bands of western blotting in Fig. 4n seemed to be comparable between vector and FTO, which may be due to the insufficient overexpression of FTO. We apologize for any oversight during the manuscript preparation. And we have repeated the western blotting assays and replaced the old bands with new ones in the revised Supplementary Fig. 4k. In addition, we can also see that compared with vector, FTO-overexpression can upregulate the levels of NRF2, FTH1, and GPX4 in the original Supplementary Fig. 3e. To further confirm the regulation of FTO on NRF2, FTH1, and GPX4, we also repeated the western blotting

assays and replaced the old bands with new ones in the revised Supplementary Fig. 5d.

Based on the MeRIP results for FTO in our study, *Nrf2*, *Fth1*, and *Gpx4* were not the potential downstream targets of FTO, while regulating FTO could significantly affect the expressions of NRF2, FTH1, and GPX4, speculating that these may be independent of the m6A demethylation manner. It is well known that NRF2, FTH1, and GPX4 are mainly involved in the redox process of glutathione; and ACSL4 and TFRC are mainly involved in Fe²⁺-dependent lipid peroxidation. Previous studies have shown that these two processes are closely communicated, and ferroptosis is triggered when the balance of them is disrupted (*Signal Transduct Target Ther.* 2021 Feb 3;6(1):49; *Nat Rev Cancer.* 2022 Jul;22(7):381-396.). Therefore, we presumed that the upregulation of ACSL4 and TFRC may affect the levels of GPX4, NRF2, and FTH1. To verify these, we additionally isolated the primary hepatocytes from the livers of aged mice and modulated their levels of ACSL4 or TFRC under FTO overexpression/silencing followed by collecting cell protein for western blotting assays to evaluate the affection of ACSL4 / TFRC on GPX4, NRF2, and FTH1. As shown in Attached Fig. 4A-4B, overexpressing or silencing ACSL4 could strongly regulate the levels of GPX4, NRF2, and FTH1, while robustly reversing the effects mediated by FTO. Modulation of TFRC can produce similar effects (Attached Fig. 4C-4D). These results may hint that the increase in lipid peroxidation will further enhance the consumption of components involved in the redox process of glutathione. Moreover, due to the limited stability maintenance capacity of senescent hepatocytes, the recovery rate of these components is insufficient, eventually inducing significant decreases in

the levels of these components, such as GPX4, NRF2, and FTH1.

Attached Figure. 4. The regulation of ACSL4 or TFRC on FTO-influenced GPX4, NRF2 and FTH1. (A-B) Western blotting assays showing the effects of overexpressing (A) or silencing (B) ACSL4 on the levels of GPX4, NRF2 and FTH1. (C-D) Western blotting assays showing the effects of overexpressing (C) or silencing (D) TFRC on the levels of GPX4, NRF2 and FTH1.

6. Figure 5a,b,d,f and j: Although the comparison between IRI+Fer-1 and FB23-2+Fer-1 represents NS, these graphs show a clear tendency of higher value in FB23-2+Fer-1 samples. If Fer-1 worked to inhibit the final production of lipid peroxides, and TFO-

mediated ACSL4 and TFRC worked at the upstream side from the accumulation of lipid peroxides, no difference between two values would be expected. In other words, there is a possibility that FTO may play a role in the other ferroptosis pathway as well as lipid peroxidation. How do the authors think about it?

Response: Thanks for your kind question. This is a very meaningful and valuable question for our study. Fer-1 is a potent and selective ferroptosis inhibitor, which can significantly inhibit the occurrence of ferroptosis, but has no obvious effect on other forms of cell death. It is accepted that the damage of hepatocytes may be attributed to multiple patterns of cell death during the process of IRI, other than ferroptosis. And increasing evidence has suggested that FTO could mitigate IRI via multiple approaches even in the same models. Ke et al. showed that FTO could inhibit the apoptosis and inflammation of cardiomyocytes via YAP1 to mitigate myocardial IRI (*Bioengineered. 2022 Mar;13(3):5443-5452*). Fei Sun et al. proved that FTO could also repress NLRP3-mediated pyroptosis to alleviate myocardial IRI (*Nat Commun. 2017 Mar 28;8:14844*). FTO has also been reported to alleviate cerebral IRI via different downstream targets (*J Physiol Biochem. 2023 Feb;79(1):133-146; Cell Signal. 2023 Sep;109:110751*). In our work, it is focus on the function of ferroptosis in aged HIRI and the regulation of FTO on ferroptosis. And we agree the reviewer's standpoint that FTO can also alleviate aged HIRI via other ways, apart from ferroptosis. Therefore, although the comparison between IRI+Fer-1 and FB23-2+Fer-1 represents NS, these graphs show a tendency of higher value in FB23-2+Fer-1 samples (revised Fig. 4c-4f), indicating that Fer-1 did not completely reverse the effects of FB23-2 and the alleviation of aged HIRI by FTO

may also mediate via other methods apart from ferroptosis, which is also deserves further investigation.

In addition, we can observe that Fer-1, as an antioxidant, is unable to completely clear intracellular reactive oxygen species under normal dosage. In this case, the elevation of ACSL4 and TFRC induced by FB23-2 may, to some extent, cause an increase in lipid peroxides, which may be related to the slightly higher values of FB23-2+Fer-1 group than Fer-1 group (revised Fig. 4h and 4k). Of course, we don't exclude that FTO may play a role in the other ferroptosis pathway as well as lipid peroxidation as you mentioned, which is worthy of further research in the future.

7. Figure 8a: Serum ALT seems to be suppressed almost completely by only shACSL4. However, FB23-2 worsen ALT under shACSL4 condition. There is no discussion about this result. Does this result mean that increased part of ACSL4 by FB23-2 cannot be suppressed by shACSL4? Otherwise, although shACSL4 could suppress ACSL4, is there other ferroptosis promoting molecule by FB23-2 involved in the increase of ALT? WB data about ACSL4 and TFRC in Figure 7d would make it clear which possibility is plausible.

Response: Thanks for your kind question. This is an essential point for our study. Herein, we have investigated the effect of AAV8-shACSL4 or AAV8-shTFRC on the increase of ACSL4 or TFRC mediated by FB23-2 via western blotting assays. As shown in revised Supplementary Fig. 8e-8f, the AAV-shRNAs almost abolished the effects of FB23-2 on ACSL4 and TFRC. However, the liver function (especially ALT) of AAV8-

shACSL4+FB23-3 group was slightly worse than that in AAV8-shACSL4 group (revised Fig. 6a), suggesting that FB23-2 or FTO may have other functions besides regulating ACSL4 and TFRC to affect liver function. During the process of IRI, the damage of hepatocytes may be attributed to multiple types of cell death, not just ferroptosis (*Redox Biol.* 2019 Jan;20:296-306; *Cell Death Differ.* 2021 May;28(5):1705-1719; *J Hepatol.* 2022 Apr;76(4):896-909). In addition, previous studies showed that FTO could inhibit the apoptosis and inflammation of cardiomyocytes via YAP1 to mitigate myocardial IRI (*Bioengineered.* 2022 Mar;13(3):5443-5452). Fei Sun et al. proved that FTO could also repress NLRP3-mediated pyroptosis to alleviate myocardial IRI (*FASEB J.* 2023 Jun;37(6):e22964). In addition, FTO has also been reported to alleviate cerebral IRI via different downstream targets (*J Physiol Biochem.* 2023 Feb;79(1):133-146; *Cell Signal.* 2023 Sep;109:110751). These studies indicate that FTO could mitigate IRI via multiple methods even in the same tissue or organ. In our work, we mainly focus on the function of ferroptosis in aged HIRI and the regulation of FTO on ferroptosis. Apart from ferroptosis, we believe that FTO can alleviate aged HIRI also via other ways, which is worth further investigation. In addition, we thank you for the suggestion. As suggested, we have discussed this in the “discussion” in our revised manuscript, which was highlighted in yellow.

8. If FTO is generally downregulated and thereby ACSL4 is upregulated in aged liver, it follows that the ratio of phospholipids including PUFA is higher in aged livers than

young ones. If there is such a report, please add it to references. In addition, what is the mechanism underlying the decline of FTO in liver with age?

Response: Thanks for your kind suggestion. Whether there are differences in the ratio of phospholipids including PUFA during liver aging is a very valuable and meaningful question. And this is also an essential point for our study. To figure this out, we performed targeted metabolomics study via Q300 Metabolomics chip in Metabo-Profile (Shanghai, China). Q300 Metabolomics chipTM technology focus ~300 targeted metabolites across all major functional metabolite classes such as amino acids and amines, organic acids, carbohydrates, fatty acids and lipids, nucleotides, sugars, vitamins, and co-factors, etc., measured by triple quadrupole mass spectrometer with an ultrahigh performance liquid chromatography.

We collected liver samples from 10 young mice (10 weeks) and 10 old mice (20 months) for targeted metabolomics analysis. The results of principal component analysis (PCA) showed that liver samples derived from young and old mice could basically be grouped independently, suggesting that there were indeed differences between the groups (Attached Fig. 3A). Subsequently, we analyzed the metabolic composition of liver tissues in different groups. As shown in Attached Fig. 3B, the proportion of fatty acids in liver samples of the elderly group increased significantly. We also displayed the differential expression of metabolites in young and old liver samples with heatmaps. As shown in Attached Fig. 3C, there were significant differences in the levels of multiple metabolites in young and old liver tissue. Compared with young liver tissues, lots of metabolites were low in old liver tissues, while the

expressions of multiple fatty acids were high, which were highlighted in a red box. Based on the differential metabolites, functional enrichment analysis was performed. As shown in Attached Fig. 3D, the pathways with the highest enrichment scores were mainly focus on the fatty acid-related signaling pathways. A total of 25 fatty acids showed significant differences between young and old liver tissues. Among them, 19 were highly expressed in old liver tissues, and the other 6 were low expressed in old liver tissues (Attached Fig. 3E). We further analyzed the types of differential metabolites and found that about 43% of fatty acids highly expressed in old liver tissues were poly-unsaturated fatty acids (Attached Fig. 3F). A detailed list of differential fatty acids has been shown in Attached Table 1. Taken together, the results of targeted metabolomics analysis suggested that poly-unsaturated fatty acids were significantly higher in old livers than that in young livers. Polyunsaturated fatty acids are the important substrates of ACSL4 (*Brain Behav Immun. 2023 Mar;109:331-343*), and our study also found that ACSL4 is highly expressed in old liver, further suggesting that old liver is more prone to ferroptosis. In addition to the above sequencing results, Julio J Ochoa et al. also found that PUFA in the liver of rats increased with aging during studying the function of Coenzyme Q in prolonging the lifespan of animals (*J Gerontol A Biol Sci Med Sci. 2007 Nov;62(11):1211-8*).

Attached Figure 3

Attached Figure 3. The Metabolomics profile of liver tissues between young and aged mice. (A) PCA showed both intragroup repeatability and intergroup variability. (B) Bar chart showing the composition of common metabolites in young and aged livers. (C) Heatmap showing differentially expressed metabolites in the different groups. (D) Enrichment analysis of differential metabolic pathways in young and aged livers. (E) Bar chart showing the statistic of differential fatty acids in young and aged livers. (F) Pie charts showing the detail composition of differential fatty acids.

Attached Table 1. Differential Fatty Acids between young and aged liver using Fold Change (FC) ≥ 1.1					
Class	HMDB	Metabolite	P-value	FC(OM/YM)	Types
Fatty Acids	HMDB0000847	Nonanoic acid	0.086449976	2.85	Medium-chain fatty acid
Fatty Acids	HMDB0001873	Isobutyric acid	0.000487129	2.092926491	Short chain fatty acid
Fatty Acids	HMDB0000207	Oleic acid	1.08E-05	1.972348687	Mono-unsaturated fatty acid
Fatty Acids	HMDB0003229	Palmitoleic acid	0.003886207	1.885591792	Mono-unsaturated fatty acid

Fatty Acids	HMDB0062248	Myristelaidic acid	1.08E-05	1.794871795	Mono-unsaturated fatty acid
Fatty Acids	HMDB0013622	Nonadeca_10Z_enoic acid	0.014689645	1.787224942	Mono-unsaturated fatty acid
Fatty Acids	HMDB0000673	Linoleic acid	1.08E-05	1.72604227	Poly-unsaturated fatty acid
Fatty Acids	HMDB0060038	10Z_Heptadecenoic acid	0.000725281	1.681465453	Mono-unsaturated fatty acid
Fatty Acids	HMDB0000529	5_Dodecenoic acid	0.017131827	1.581632653	Mono-unsaturated fatty acid
Fatty Acids	NA	Gamma_Linolenic acid	0.000487129	1.466145151	Poly-unsaturated fatty acid
Fatty Acids	HMDB0002183	Docosahexaenoic acid, DHA	0.014689645	1.43877303	Poly-unsaturated fatty acid
Fatty Acids	HMDB0001043	Arachidonic acid	0.018543376	1.409956754	Poly-unsaturated fatty acid
Fatty Acids	HMDB0001999	Eicosapentaenoic acid, EPA	0.075256013	1.340252238	Poly-unsaturated fatty acid
Fatty Acids	HMDB0002925	8,11,14_Eicosatrienoic acid	0.014689645	1.291791986	Poly-unsaturated fatty acid
Fatty Acids	HMDB0000826	Pentadecanoic acid	0.001050034	1.281845479	Long-chain fatty acid
Fatty Acids	HMDB0006528	Docosapentaenoic acid, DPA	0.005196042	1.252585083	Poly-unsaturated fatty acid
Fatty Acids	HMDB0002259	Heptadecanoic acid	0.023290235	1.249637725	Long-chain fatty acid
Fatty Acids	HMDB0034297	Ricinoleic acid	0.049195354	1.235294118	Mono-unsaturated fatty acid
Fatty Acids	HMDB0002226	Adrenic acid	0.063012839	1.197719053	Poly-unsaturated fatty acid
Fatty Acids	HMDB0031158	2_Methyl_4_pentenoic acid	0.039052775	0.684210526	Methyl-branched fatty acid
Fatty Acids	HMDB0000407	2_Hydroxy_3_methylbutyric acid	0.000328133	0.624209575	Hydroxy fatty acid
Fatty Acids	HMDB0000555	3_Methyladipic acid	0.031082168	0.469314079	Methyl-branched fatty acid
Fatty Acids	HMDB0000237	Propanoic acid	0.000324753	0.386490341	Short chain fatty acid
Fatty Acids	HMDB0000754	3_Hydroxyisovaleric acid	0.001003556	0.371841155	Short-chain hydroxy acid
Fatty Acids	HMDB0001901	Aminocaproic acid	0.000180635	0.122699387	Medium-chain fatty acid

Attached Table 1. Differential Fatty Acids between young and aged liver tissues

As to the mechanism underlying the decline of FTO in livers with age, we have made a preliminary investigation on this. Given that the alteration in protein's level is typically associated with variations in the rate of protein synthesis or degradation, we first detected the difference of FTO synthesis or degradation between young and aged livers. We isolated primary hepatocytes from young and aged mice, treated hepatocytes with MG132 or Cycloheximide (CHX) for indicated time, then collected cell proteins to examine the protein levels of FTO via western blotting assays. MG132, a potent proteasome inhibitor, can inhibit protein degradation via proteasome, which is commonly used to investigate the alterations in protein synthesis. CHX can strongly inhibit the protein synthesis, which is commonly used to detect the changes in protein degradation. As shown in Attached Fig. 5A-5B, the accumulation of FTO was with no significant difference between young and old hepatocytes, while the degradation rate of FTO was significantly faster in the old ones, suggesting that the decline of FTO in aged livers may attribute to the faster degradation. Victor Anggono et al. showed that the degradation of FTO was mainly mediated by the ubiquitin proteasome pathway (*J Mol Biol.* 2018 Feb 2;430(3):363-371). Therefore, we further detected the ubiquitination of FTO in young and old hepatocytes. As shown in Attached Fig. 5C, the ubiquitination level of FTO in old hepatocytes was significantly higher than that in young ones, which suggested that the decrease of FTO in aged livers may be due to the faster degradation via the ubiquitin proteasome pathway. And it is worth investigating the specific molecular mechanism of the accelerated degradation of FTO via the ubiquitin proteasome pathway in aged livers in our further study.

Attached Figure 5. The degradation of FTO in old hepatocytes is faster than that in young ones. (A) WB assays showing the synthesis of FTO in young and old hepatocytes. (B) WB assays showing the degradation of FTO in young and old hepatocytes. (C) Co-IP assays showing the ubiquitination levels of FTO in young and old hepatocytes.

Reviewers 3:

This manuscript discusses how the FTO regulated hepatic ischemia-reperfusion injury (HIRI) in aged liver by targeting ferroptosis inducers ACSL4 and TFRC. In line with higher susceptibility to HIRI in aged liver, they found aged liver were characterized by a higher intensity of ferroptosis than young livers, and inhibiting ferroptosis can significantly alleviates aged HIRI. What's more, they found that aged livers had lower FTO expression and higher m6A methylation levels. FTO-mediated demethylation was found to shorten the half-life of ACSL4 and TFRC, two important inducers for ferroptosis, to regulate ferroptosis in aged HIRI.

The author investigated how FTO influence aged HIRI by regulating ferroptosis. However, the regulation of ferroptosis by FTO(ref 1,2) has been studied, and the link

between HIRI and ferroptosis (ref3) has also been established previously. Therefore, it is not surprising that FTO affects HIRI through ferroptosis. A more interesting question is that, why FTO was down-regulated in aged livers. What's more, the authors show that FTO will be down-regulated after IR both in young and old livers. Therefore, exploring the regulatory mechanisms of FTO down-regulation becomes even more intriguing.

Another major concern is about the validity of their MeRIP-seq analysis. It appears that the analysis lacks replicates and an input control (Figure 6e), and there is also a notable absence of enrichment scores (such as P-value or fold change) and rankings for motifs (Figure 6C).

Overall, the aforementioned concerns limit the impact and validity of the manuscript.

Major point:

1. In the manuscript, the authors did not adequately utilize HPLC to explain why the aging liver is more susceptible to ischemia-reperfusion injury (Figure 3). I would like the authors to employ HPLC for additional analyses, such as enrichment analysis, to elucidate the differential responses observed during IRI in young and aged livers. This would provide a more comprehensive understanding of the mechanisms underlying the increased susceptibility of the aging liver to IRI.

Response: Thanks for your kind suggestion. According to your suggestion, we performed functional enrichment analysis based on the differential proteins among the groups of HPLC. The related results of functional enrichment analysis were integrated

in the revised Supplementary Fig. 2a-2d. These results showed that the differential proteins between old-IRI and old-sham were mainly involved in multiple processes of metabolism, including “Biosynthesis of unsaturated fatty acids”. In addition, “ferroptosis” was also included in the enriched signaling pathways (Supplementary Fig. 2b). Compared with old groups, the differential proteins between young-IRI and young-sham were mainly participated in the processes of inflammation, such as “Neutrophil extracellular trap formation”. And similar to the old groups, ferroptosis was included in the enriched signaling pathways (Supplementary Fig. 2b). Then, we analyzed the differential proteins between old-sham and young sham and found that the enriched signaling pathways were mainly associated with metabolism, such as “Biosynthesis of unsaturated fatty acids” (Supplementary Fig. 2c). Moreover, the enriched signaling pathways of “old-IRI vs young IRI” were also mainly related to metabolism, including “linoleic acid metabolism” (Supplementary Fig. 2d). Based on the functional enrichment analysis of the above four groups, we found that ferroptosis was indeed involved in HIRI. And compared with young groups, the metabolic changes were more active in the old groups, especially the metabolic pathways associated with unsaturated fatty acids, such as “Biosynthesis of unsaturated fatty acids” and “linoleic acid metabolism”. The increase of unsaturated fatty acids is one of the key causes of ferroptosis, the above enrichment analysis results further suggested that aged livers is susceptible to ferroptosis.

2. The authors focused on the differential expressed protein in comparison groups

(Figure 3E). However, it is essential for them to clearly specify whether this analysis is conducted specifically on up-regulated proteins or down-regulated proteins.

Response: Thank you for the kind question. This is an essential point for our study. In this study, the Venn map in Fig. 3E covered all differential expressed proteins in comparison groups. We did not perform the analysis specially on up-regulated proteins or down-regulated proteins, as separate analysis may exist risk for missing some proteins that do not show the same trend between different comparison groups, such as the proteins up-regulated in “young-IRI vs young-sham” while down-regulated in “old-IRI vs old-sham”. The abilities of young and aged livers to cope with IRI are significantly different (*Aging Cell. 2020 Aug;19(8):e13186.*), and under this condition, the differential proteins with inconsistent trends between the two comparison groups may make sense.

3. The western blot in Figure 3f indicates that FTO was down-regulated after IR in both young and old livers. However, the HPLC analysis in Figure 3d demonstrates that FTO was only down-regulated in old livers. These two results appear to be somewhat contradictory. Therefore, it is crucial for the authors to clearly present and discuss the HPLC data regarding FTO expression to reconcile the observed differences.

Response: Thank you for the kind question. This is an essential point for our study. We rechecked the results from the original manuscript and apologize for the low quality of control bands in Fig. 3f, which may affect our judgment of the results. For these, we have repeated the western blotting assays, which showed that the levels of FTO were

not significantly changed after IR in young livers (both in human and mouse). In addition, the results in primary hepatocytes and THLE2 cells also proved that the down-regulation of FTO during IR were mainly occurred in aged liver cells (Fig. 3g). We have replaced the bands with new ones in the revised manuscript.

4. The author presented two presentative motifs in Figure 6C. However, it appears that the motif-1 does not correspond to canonical m6A motif. Furthermore, it is important to include enrichment scores, such as p-values and fold changes, for the identified motifs in the MeRIP-seq analysis. Providing these additional metrics would enhance the interpretation and significance of the identified motifs.

Response: Thanks for your kind suggestion. This is an essential point for our study. Several evidence demonstrated that RRACH (R=A/G; H=A/U/C) is the canonical m6A motif in mammals, fungi, and plants (*Nature. 2012 Apr 29;485(7397):201-6; Nat Chem Biol. 2011 Oct 16;7(12):885-7*). Therefore, the sequences of motif-1 and motif-2 in our study are basically conformed to the classical motif sequences of m6A. In addition, we would like to express our gratitude to the reviewer for informing us of the relevant concern. As suggested, we carefully added the p-values of motifs in the revised manuscript (revised Fig. 5C).

5. Although the author performs function enrichment on differential m6A peaks (Figure 6d), it is not clearly specified whether this analysis is conducted specifically on up-regulated peaks, down-regulated peaks.

Response: Thanks for your kind suggestion. We apologize for the oversight during the conduction of function enrichment analysis on differential m6A peaks. According to the reviewer's suggestion, we re-conducted the function enrichment analysis specifically on up-regulated peaks or down-regulated peaks and the related results were showed in Attached Fig. 6A-6B. Considering that when FTO is overexpressed, the m6A levels of its downstream targets are decreased, we mainly focused our attention on the signaling pathways enriched by down-regulated m6A peaks (Attached Fig. 6A). And we have replaced Fig. 6d (revised Fig. 5d) with Attached Fig. 6A. The signaling pathways enriched by down-regulated m6A peaks included ferroptosis and multiple metabolism-related signaling pathways, which further suggested that FTO could participate in the regulation of ferroptosis (Attached Fig. 6A). We have added a more detail description of the enrichment analysis for Fig. 6d (revised Fig. 5d) in revised manuscript, which was highlighted in yellow.

A

KEGG enrichment analysis on down-regulated m6A peaks

B

KEGG enrichment analysis on up-regulated m6A peaks

Attached Figure 6. KEGG enrichment analysis on differential m6A peaks. (A) The signaling pathways enriched by down-regulated m6A peaks. (B) The signaling pathways enriched by up-regulated m6A peaks.

6. Given that MeRIP-seq signal is positively correlated with gene expression, it is essential to include replicates samples and input control to ensure the reliability of the finding. If there are replicates and input control, there are recommended to presented in the IGV (Figure 6E).

Response: Thanks for your kind suggestion. This is an essential point for our study. We have performed MeRIP-seq for replicate samples and related input control. We apologize for did not displaying them in the IGV during manuscript preparation, which may affect the reliability of the findings. According to the reviewer's suggestion, we added the replicate samples and related input controls into IGV in the revised manuscript (Attach Fig. 7). And we selected representative one FTO and one vector to display the m6A modification in IGV, which were shown in the revised Fig. 5E.

Attached Figure 7. Integrated genome browser views show the m6A modifications of *Acs14* (A) and *Tfrce* (B) in the control and FTO groups. Each group consisted of three replicate samples.

7. In addition to MeRIP-seq, it is important to obtain additional evidence, such as FTO CLIP-seq, to confirm the direct targeting of ACSL4 and TFRC by FTO.

Response: Thanks for your kind suggestion. And this is an essential point for our study.

Authentically, to fully validate that ACSL4 and TFRC are the downstream targets of FTO is a very important point in this study. In our original manuscript, MeRIP-qPCR was performed to verify the results of MeRIP-seq (Fig. 6f). And we would like to express our gratitude to the reviewer for informing us of the relevant concern.

According to the reviewer's suggestion, we additionally performed FTO CLIP-seq, which showed that FTO could enrich *Acs14* and *Tfrc* in the regions where the m6A modification levels were significantly decreased under FTO-overexpressing, which further suggested that FTO could bind *Acs14* and *Tfrc* to decreased the m6A modification of both. The relevant results were added in the revised Fig. 5g in our revised manuscript.

8. The authors observed that FTO inhibits the stability of ACSL4 and TFRC mRNAs by demethylation. To strengthen this conclusion, it is necessary for the authors to present supplementary evidence that confirms the participation of m6A reader proteins in the regulation of ACSL4 and TFRC.

Response: Thanks for your kind suggestion. And this is an essential point for our study. Figuring out the m6A reader proteins, which may regulate the stability of *Acs14* and *Tfrc* mRNAs in our study, will make the study more convincing and complete. Previous studies have identified that YTHDF1, YTHDF2, YTHDF3, EIF3A, IGF2BPs, YTHDC1, and YTHDC2 are the most dominant m6A readers in mammalian cells. Among them, YTHDF3 commonly serves as a hub for fine-tuning the RNA accessibility of YTHDF1 and YTHDF2 (*Cell Res. 2018 Jun;28(6):616-624; Cell. 2015 Jun 4;161(6):1388-99.*); nuclear m6A reader YTHDC1 plays an important role in mRNA splicing (*Mol Cell. 2016 Feb 18;61(4):507-519*); IGBPs consists of three proteins, IGBP1, IGBP2 and IGBP3, which usually function in the form of complex and the studies of IGBPs usually take IGBP1 as the entry point (*Nat Cell Biol. 2018*

Mar;20(3):285-295). Based on these, we selected YTHDF1, YTHDF2, EIF3A, IGF2BP1, and YTHDC2 for further investigation. We first detected the effects of different readers on the protein levels of ACSL4 and TFRC. As shown in revised Supplementary Fig. 7e, YTHDC2 could increase the level of ACSL4 and YTHDF2 could increase the level of TFRC, which suggests that YTHDC2 may be the m6A reader of *Acs14* and YTHDF2 may be the m6A reader of *Tfrc*. To validate this hypothesis, we performed RNA Immunoprecipitation (RIP) assays with YTHDC2 and YTHDF2. As shown in revised Supplementary Fig. 7f, YTHDC2 exhibited significant enrichment of *Acs14* mRNA, while YTHDF2 preferentially enriched *Tfrc* mRNA. Previous studies have shown that YTHDF2 and YTHDC2 could enhance mRNA stability (*Cancer Discov. 2021 Feb;11(2):480-499; FASEB J. 2023 Mar;37(3):e22803*). In this study, we observed that FTO reduced the half-life of *Acs14* and *Tfrc* mRNAs, which suggested that the erasure of m6A modification by FTO on *Acs14* and *Tfrc* may affect the regulation of YTHDC2 on ACSL4 and YTHDF2 on TFRC. Then, we examined the influence of overexpressing or silencing FTO on the regulation of ACSL4 or TFRC mediated by YTHDC2 or YTHDF2, respectively. As shown in revised Fig. 5m, overexpressing FTO almost abolished the increase of ACSL4 mediated by YTHDC2, while silencing FTO could enhance the regulation of YTHDC2 on ACSL4. Similar to the effect of FTO on YTHDC2, altering the levels of FTO can also significantly influence YTHDF2-mediated regulation on TFRC (revised Fig. 5n). In addition, FTO also significantly influenced the mRNA enrichment of *Acs14* by YTHDC2 (revised Fig. 5o) and *Tfrc* by YTHDF2 (revised Fig. 5p). All these results indicate that the removal

of posttranscriptional m6A modification on *Acsl4* and *Tfrc* by FTO affected the regulation of YTHDC2 on ACSL4 and YTHDF2 on TFRC, shortened the half-life of *Acsl4* and *Tfrc* mRNAs, which ultimately reduced the expression of ACSL4 and TFRC.

Minor points:

1. The spelling of "ischaemia" or "ischemia" should be consistent throughout the text

Response: Thanks for your kind suggestion. And we apologize for the spelling mistake during the manuscript preparation. As suggested, we've changed all "ischemia" to "ischaemia" in the revised manuscript.

2. The intensity of the β -actin signal appears inconsistent across different groups (Figure 3f, bottom)

Response: Thanks for your kind suggestion. We apologize for any oversight during the manuscript preparation. As suggested, we have repeated the western blotting assays for Fig. 3f and replaced low quality bands with new ones.

3. The author needs to improve the annotation of the Figure, for example, Figure x should be annotated on the page

Response: Thanks for your kind suggestion. We apologize for the oversight during the annotation of all Figures in the original manuscript. As suggested, we have carefully refined the annotation of all Figures in the revised manuscript.

4. The author should pay attention to the writing standards for protein and mRNA labeling.

Response: Thanks for your kind suggestion. We apologize for any oversight during the manuscript preparation. As suggested, we have rechecked the labels for protein / mRNA and corrected the wrong labels in the revised manuscript.

REVIEWER COMMENTS

Reviewer #1 (Remarks to the Author):

The authors' efforts in addressing the previous questions are appreciated. On the other hand, some questions related to the central conclusions of the study require further clarifications:

Question #3 remains unresolved, since (polyunsaturated) fatty acids are substrates of ACSL4, not products. The direct (ACSL4-dependent) substrates for lipid peroxidation and ferroptosis are esterified lipids, including PUFA-phospholipids, not free fatty acids. The approach for addressing this question should be lipidomics, not polar metabolomics. Without this result, it remains unclear whether the changes in ACSL4 expression downstream of FTO expression alterations has a contribution in driving ferroptosis in the indicated settings.

Related to Question #1, it remains not completely resolved that whether the TUNEL-positive cells are the same as the oxidized-lipid-positive cells. Without evidence along this line, it is difficult to conclude that the lipid peroxidation-positive cells are indeed undergoing cell death (via ferroptosis) during IRI or other stress employed in this work, which is a key conclusion in this study. Could it be that the TUNEL-positive cells are a different group of cells from those exhibiting lipid peroxidation?

A technical question related to Question #1: in imaging results where BODIPY-C11 staining was employed, why are the non-oxidized lipids distributed so distinctly (unevenly)? Would the unevenness of the basal distribution of the dye in the tissue introduce biases in the validity of the oxidation signal?

In addition, the writing of the manuscript should also be carefully scrutinized since many claims are inaccurate, inappropriate or misleading. The authors should seek for professional assistance for revising the narratives. Some examples are listed:

Line #31, "obviously" seems an over-claiming

In abstract, line #32, spell out FTO

Line 34, ACSL4 and TFRC, two key inducers of ferroptosis, change "inducers" to "positive contributors" or "participants"

Line 65, HUWEI should be HUWE1

Line 122, typo in “pattens”, do the authors intend to mean “pathways”?

Line 127-128, fig 1e, the changes in the expression of these upstream regulatory components of the ferroptosis pathway do not indicate that “the change of ferroptosis was relatively more significant and the degree of ferroptosis in aged livers was more serious”.

Line 177-179, do the authors mean LC-MS proteomics by HPLC?

Line 185-186, “ferroptosis was indeed involved in both young and aged HIRI”, the protein expression analysis can not be used to infer functional involvement of a cell death pathway

Reviewer #2 (Remarks to the Author):

The authors have sufficiently addressed my concerns. The paper can be accepted.

Reviewer #3 (Remarks to the Author):

The authors have adequately addressed all my comments in the revised version of the manuscript. Therefore, I have no further comments.

Reviewer #1:

The authors' efforts in addressing the previous questions are appreciated. On the other hand, some questions related to the central conclusions of the study require further clarifications:

Question #3 remains unresolved, since (polyunsaturated) fatty acids are substrates of ACSL4, not products. The direct (ACSL4-dependent) substrates for lipid peroxidation and ferroptosis are esterified lipids, including PUFA-phospholipids, not free fatty acids. The approach for addressing this question should be lipidomics, not polar metabolomics. Without this result, it remains unclear whether the changes in ACSL4 expression downstream of FTO expression alterations has a contribution in driving ferroptosis in the indicated settings.

Response : Thanks for your kind suggestion. And this is an essential point for our study.

According to your suggestion, we performed Relative Quantitative Lipidomics with liver tissues obtained from young (2 months, n=8) and aged mice (20 months, n=8) in APTBIO (Shanghai, China).

The results of principal component analysis (PCA), Partial Least Squares Discrimination Analysis (PLS-DA) and Orthogonal PLS-DA (OPLS-DA) showed that liver samples derived from young and old mice could basically be grouped independently, suggesting that there were indeed differences in the lipidomics between these two groups (Attached Fig. 1A). As shown in Attached Fig. 1B, a total of 42 lipid classes (including 2486 lipid species) were identified in young and aged liver tissues. Then, we displayed the differential expression of lipid species in young and old liver samples with heatmaps. As shown in Attached Fig. 1C, there were significant differences in the levels of multiple lipid species in young and old liver tissues. Compared with young liver tissues, lots of lipid species were increased in that of old. Among them, the top 14 lipid classes with the most significant differences between young and old liver tissues were showed in bubble charts (Attached Fig. 1D), which further suggested that multiple lipids were higher expressed in old liver tissues.

As the reviewer mentioned, we should pay more attention to the expression profile

of esterified lipids, which are the direct (ACSL4-dependent) substrates for lipid peroxidation and ferroptosis, in our work. Previous studies (Nat Rev Clin Oncol,2021; Redox Biol,2023; Biochim Biophys Acta Mol Cell Biol Lipids,2019; Cell,2024; Signal Transduct Target Ther,2022) have shown that membrane-associated glycerophospholipids, such as phosphatidylethanolamine (PE), phosphatidylserine (PS), phosphatidylcholine (PC), et al., are the key phospholipids that induce ferroptosis in cells. ACSL4 participates in the biosynthesis and remodeling of them, activates polyunsaturated fatty acids and affects their transmembrane properties, eventually inducing ferroptosis under certain circumstance. Given that phosphatidylglycerol (PG), phosphatidylinositol (PI), phosphatidic acid (PA) and sphingomyelin (SM) are also critical membrane-associated phospholipids, we herein compared the lipid unsaturation of PE, PC, PS, PG, PI, PA and SM between young and old liver tissues. As shown in Attached Fig. 2, the lipid unsaturation of these phospholipids were increased in old liver tissues. In addition, we also evaluated the lipid unsaturation of fatty acids (FA), and the results also suggested that there was a slight upward trend in the unsaturation of FA in old liver tissues compared to the young, with no significance, which may be attributed to the low number of FA detected in this work (n=5). Taken together, all these results indicate that the esterified lipids, which are the direct (ACSL4-dependent) substrates for lipid peroxidation and ferroptosis are highly expressed in aged liver tissues. Under this condition, the increase of ACSL4 can significantly induce ferroptosis.

Attached Figure 1. The lipidomics profile of young and aged liver tissue. (A) PCA, PLS-DA, and OPLS-DA showed both intragroup repeatability and intergroup variability. **(B)** Bar chart showing the overview of lipid classes and related lipid species' numbers detected in young and old liver tissues. **(C)** Heatmap showing differentially expressed lipid species in the indicated groups. **(D)** Bubble charts showing the top 14 lipid classes with the most significant differences in young and aged livers.

Attached Figure 2. The differential analysis of critical esterified lipids between young and aged liver tissue. Bar charts showing the lipid unsaturation analysis of indicated esterified lipids, including phosphatidylethanolamine (PE), phosphatidylserine (PS), phosphatidylcholine (PC), phosphatidylglycerol (PG), phosphatidylinositol (PI), phosphatidic acid (PA), sphingomyelin (SM) and Fatty acid (FA). *P < 0.05, **P < 0.01, ***P < 0.001, ****P < 0.0001.

Related to Question #1, it remains not completely resolved that whether the TUNEL-positive cells are the same as the oxidized-lipid-positive cells. Without evidence along this line, it is difficult to conclude that the lipid peroxidation-positive cells are indeed undergoing cell death (via ferroptosis) during IRI or other stress employed in this work, which is a key conclusion in this study. Could it be that the TUNEL-positive cells are a different group of cells from those exhibiting lipid peroxidation?

Response : Thanks for your kind suggestion. C11 BODIPY staining is one of the most common methods to detect ferroptosis and has been reported to be used in multiple

studies(Cancer Res,2022; J Exp Clin Cancer Res,2022; Cell Chem Biol,2019). In the past, several studies have co-stained TUNEL and C11 BODIPY to detect the correlation between TUNEL-positive cells and oxidized-lipid-positive cells (Bone Res,2018). And we appreciate the reviewer's question mentioned above. This is an essential point for our study. As suggested, to figure out whether the TUNEL-positive cells are the same as the oxidized-lipid-positive cells in this study, we herein additionally used a confocal microscopy to detect the localization of TUNEL- and C11 BODIPY-staining in the liver tissues derived from the aged HIRI mice. As shown in Attached Fig.3, the results presented four distribution patterns of TUNEL and C11 BODIPY staining, including: (1) TUNEL-positive cells were the same as the oxidized-lipid-positive cells, such as Case 1 and Case 2; (2) The region of oxidized-lipid-positive cells showed no presence of TUNEL-staining, such as Case 3; (3) The oxidized-lipid-positive cells and TUNEL-positive cells were distributed in the same area but not co-located with each other, such as Case 4; (4) The area of TUNEL-positive cells showed no presence of C11 BODIPY-staining, such as Case 5. And the pattern 1 of the staining distribution was predominance in our study, indicating that most of the TUNEL-positive cells were the same as the oxidized-lipid-positive cells. Apart from C11 BODIPY staining, we also performed Reactive Oxygen Species (ROS) detection, Transmission electron microscopy (TEM) assays and Western Blotting assays to detect ferroptosis in our work.

Indeed, not all TUNEL-positive cells were suffered from lipid peroxidation, such as Case 4-5. Given that TUNEL staining is a classical method to indicate apoptosis, which is also involved in liver IRI, and apoptosis and ferroptosis are distinct patterns of cell death, the TUNEL-positive cells are not the same as the oxidized-lipid-positive cells is also reasonable.

Attached Figure 3. The results of TUNEL and C11 BODIPY co-staining assays in liver tissues derived from aged HIRI.

A technical question related to Question #1: in imaging results where BODIPY-C11 staining was employed, why are the non-oxidized lipids distributed so distinctly (unevenly)? Would the unevenness of the basal distribution of the dye in the tissue introduce biases in the validity of the oxidation signal?

Response : Thanks for your kind reminding. C11 BODIPY 581/591 is a BODIPY borofluoroprene derivative with good light stability and low fluorescence artifacts,

which is commonly used to detect ferroptosis. C11 BODIPY 581/591 is emitted at 591 nm (reduced prototype, non-oxidized lipids, green), or redshifted to 510 nm (oxidized type, oxidized lipids, magenta). The distribution and intensity of non-oxidized lipids are closely related to the total lipids level and the peroxidation of lipids in tissues or cells under specific conditions. In our work, C11 BODIPY staining was mainly used to detect the ferroptosis in aged liver IRI. Not all intracellular lipid content in liver tissues is uniform, and the degree of lipid peroxidation in cells is also different when subjected to IRI stress (Hepatology,2010; Nat Commun,2021). In addition, the exposure time chosen when taking pictures is fixed, making the signal not significance where there are less non-oxidized lipids. All these may cause the distribution of non-oxidized lipids to appear uneven. And the previous studies (Sci Adv,2023; J Transl Med,2023; Biomark Res,2024) have also showed the similar results.

In addition, we herein used the representative images of C11 BODIPY staining and bar charts of oxidized lipid rates to display the results of C11 BODIPY assays. The calculation method of the ratio of oxidative lipids was performed according to the previous studies (Nat Chem Biol,2023; Cell Death Differ,2022; Nat Commun,2017): the ratio of oxidized lipids = the number of cells with oxidized lipids/ the number of total cells. Due to oxidized lipids and non-oxidized lipids can exist in the same cell under some condition, the non-oxidized lipids (magenta) was not used to calculated the ratio of oxidized lipids, which was mainly used to reflect that the increase of oxidized lipids in the cell is usually accompanied by a decrease in non-oxidized lipids. Therefore, the distribution and intensity of non-oxidized lipids in tissues or cells don't have much influence on the interpretation of the results of C11 BODIPY staining assays.

In addition, the writing of the manuscript should also be carefully scrutinized since many claims are inaccurate, inappropriate or misleading. The authors should seek for professional assistance for revising the narratives. Some examples are listed:

Response: Thanks for your kind suggestion. And we apologized for any oversight during the manuscript preparation. As suggested, we had carefully revised the issues you mentioned below and we also sent the manuscript to an expert in the related field

for revising the narratives. All revisions were highlighted with yellow in the revised manuscript.

Line #31, “obviously” seems an over-claiming

Response: Thanks for your kind suggestion. As suggested, we have amended it in the revised manuscript and highlighted in yellow.

In abstract, line #32, spell out FTO

Response: Thank you for the kind reminding. And we apologized for the oversight during the manuscript preparation. As suggested, we have revised it, which was highlighted with yellow in the revised manuscript.

Line 34, ACSL4 and TFRC, two key inducers of ferroptosis, change “inducers” to “positive contributors” or “participants”

Response: Thanks for your kind suggestion. This is an essential point for our study. We have revised it according to your suggestion, which was highlighted with yellow in the revised manuscript.

Line 65, HUWEI should be HUWE1.

Response: Thanks for your kind reminding. We apologized for the oversight during the manuscript preparation. We have corrected it and marked it as yellow in the revised manuscript.

Line 122, typo in “pattens”, do the authors intend to mean “pathways”?

Response: Thanks for your kind reminding. As suggested, we have revised it, which was highlighted with yellow in the revised manuscript.

Line 127-128, fig 1e, the changes in the expression of these upstream regulatory components of the ferroptosis pathway do not indicate that “the change of ferroptosis was relatively more significant and the degree of ferroptosis in aged livers was more

serious”.

Response: Thank you for the kind suggestion. And we apologized for any inappropriate description during the manuscript preparation. As suggested, we modified this and highlighted in yellow in our revised manuscript.

Line 177-179, do the authors mean LC-MS proteomics by HPLC?

Response: Thanks for your kind reminding. And we apologized for the oversight during the preparation of the manuscript. We have corrected it and marked it as yellow in the revised manuscript.

Line 185-186, “ferroptosis was indeed involved in both young and aged HIRI”, the protein expression analysis can not be used to infer functional involvement of a cell death pathway

Response: Thanks for your kind suggestion. And we apologized for any inappropriate description during the manuscript preparation. As suggested, we have modified it, which were highlighted in yellow in our revised manuscript.

Anandhan, A., Dodson, M., Shakya, A., Chen, J., Liu, P., Wei, Y., Tan, H., Wang, Q., Jiang, Z., Yang, K., Garcia, J. G., Chambers, S. K., Chapman, E., Ooi, A., Yang-Hartwich, Y., Stockwell, B. R., & Zhang, D. D. (2023). NRF2 controls iron homeostasis and ferroptosis through HERC2 and VAMP8. *Sci Adv*, 9(5), eade9585. <https://doi.org/10.1126/sciadv.ade9585>

Atiya, H. I., Frisbie, L., Goldfeld, E., Orellana, T., Donnellan, N., Modugno, F., Calderon, M., Watkins, S., Zhang, R., Elishaev, E., Soong, T. R., Vlad, A., & Coffman, L. (2022). Endometriosis-Associated Mesenchymal Stem Cells Support Ovarian Clear Cell Carcinoma through Iron Regulation. *Cancer Res*, 82(24), 4680–4693. <https://doi.org/10.1158/0008-5472.Can-22-1294>

Barayeu, U., Schilling, D., Eid, M., Xavier da Silva, T. N., Schlicker, L., Mitreska, N., Zapp, C., Gräter, F., Miller, A. K., Kappl, R., Schulze, A., Friedmann Angeli, J. P., & Dick, T. P. (2023). Hydropersulfides inhibit lipid peroxidation and ferroptosis by scavenging radicals. *Nat Chem Biol*, 19(1), 28–37.

- <https://doi.org/10.1038/s41589-022-01145-w>
- Byun, J. K., Lee, S., Kang, G. W., Lee, Y. R., Park, S. Y., Song, I. S., Yun, J. W., Lee, J., Choi, Y. K., & Park, K. G. (2022). Macropinocytosis is an alternative pathway of cysteine acquisition and mitigates sorafenib-induced ferroptosis in hepatocellular carcinoma. *J Exp Clin Cancer Res*, 41(1), 98. <https://doi.org/10.1186/s13046-022-02296-3>
- Chen, X., Kang, R., Kroemer, G., & Tang, D. (2021). Broadening horizons: the role of ferroptosis in cancer. *Nat Rev Clin Oncol*, 18(5), 280–296. <https://doi.org/10.1038/s41571-020-00462-0>
- Greenough, M. A., Lane, D. J. R., Balez, R., Anastacio, H. T. D., Zeng, Z., Ganio, K., McDevitt, C. A., Acevedo, K., Belaidi, A. A., Koistinaho, J., Ooi, L., Ayton, S., & Bush, A. I. (2022). Selective ferroptosis vulnerability due to familial Alzheimer's disease presenilin mutations. *Cell Death Differ*, 29(11), 2123–2136. <https://doi.org/10.1038/s41418-022-01003-1>
- Kannan, M., Sil, S., Oladapo, A., Thangaraj, A., Periyasamy, P., & Buch, S. (2023). HIV-1 Tat-mediated microglial ferroptosis involves the miR-204-ACSL4 signaling axis. *Redox Biol*, 62, 102689. <https://doi.org/10.1016/j.redox.2023.102689>
- Kuwata, H., Nakatani, E., Shimbara-Matsubayashi, S., Ishikawa, F., Shibamura, M., Sasaki, Y., Yoda, E., Nakatani, Y., & Hara, S. (2019). Long-chain acyl-CoA synthetase 4 participates in the formation of highly unsaturated fatty acid-containing phospholipids in murine macrophages. *Biochim Biophys Acta Mol Cell Biol Lipids*, 1864(11), 1606–1618. <https://doi.org/10.1016/j.bbalip.2019.07.013>
- Lee, S. W., Rho, J. H., Lee, S. Y., Chung, W. T., Oh, Y. J., Kim, J. H., Yoo, S. H., Kwon, W. Y., Bae, J. Y., Seo, S. Y., Sun, H., Kim, H. Y., & Yoo, Y. H. (2018). Dietary fat-associated osteoarthritic chondrocytes gain resistance to lipotoxicity through PKCK2/STAMP2/FSP27. *Bone Res*, 6, 20. <https://doi.org/10.1038/s41413-018-0020-0>
- Li, T., Owsley, E., Matozel, M., Hsu, P., Novak, C. M., & Chiang, J. Y. (2010). Transgenic expression of cholesterol 7 α -hydroxylase in the liver prevents high-fat diet-induced obesity and insulin resistance in mice. *Hepatology*, 52(2), 678–690. <https://doi.org/10.1002/hep.23721>
- Liu, D. S., Duong, C. P., Haupt, S., Montgomery, K. G., House, C. M., Azar, W. J., Pearson, H. B., Fisher, O. M., Read, M., Guerra, G. R., Haupt, Y., Cullinane, C., Wiman, K. G., Abrahmsen, L., Phillips, W. A., & Clemons, N. J. (2017). Inhibiting the system x(C)(-)/glutathione axis selectively targets cancers with mutant-p53 accumulation. *Nat Commun*, 8, 14844. <https://doi.org/10.1038/ncomms14844>
- Magtanong, L., Ko, P. J., To, M., Cao, J. Y., Forcina, G. C., Tarangelo, A., Ward, C. C., Cho, K., Patti, G. J., Nomura, D. K., Olzmann, J. A., & Dixon, S. J. (2019). Exogenous Monounsaturated Fatty Acids Promote a Ferroptosis-Resistant Cell State. *Cell Chem Biol*, 26(3), 420–432.e429. <https://doi.org/10.1016/j.chembiol.2018.11.016>
- Qiu, B., Zandkarimi, F., Bezjian, C. T., Reznik, E., Soni, R. K., Gu, W., Jiang, X., &

- Stockwell, B. R. (2024). Phospholipids with two polyunsaturated fatty acyl tails promote ferroptosis. *Cell*, 187(5), 1177–1190.e1118. <https://doi.org/10.1016/j.cell.2024.01.030>
- Rizvi, F., Everton, E., Smith, A. R., Liu, H., Osota, E., Beattie, M., Tam, Y., Pardi, N., Weissman, D., & Gouon-Evans, V. (2021). Murine liver repair via transient activation of regenerative pathways in hepatocytes using lipid nanoparticle-complexed nucleoside-modified mRNA. *Nat Commun*, 12(1), 613. <https://doi.org/10.1038/s41467-021-20903-3>
- Tai, P., Chen, X., Jia, G., Chen, G., Gong, L., Cheng, Y., Li, Z., Wang, H., Chen, A., Zhang, G., Zhu, Y., Xiao, M., Wang, Z., Liu, Y., Shan, D., He, D., Li, M., Zhan, T., Khan, A., . . . Cao, K. (2023). WGX50 mitigates doxorubicin-induced cardiotoxicity through inhibition of mitochondrial ROS and ferroptosis. *J Transl Med*, 21(1), 823. <https://doi.org/10.1186/s12967-023-04715-1>
- Tan, X., Kong, D., Tao, Z., Cheng, F., Zhang, B., Wang, Z., Mei, Q., Chen, C., & Wu, K. (2024). Simultaneous inhibition of FAK and ROS1 synergistically repressed triple-negative breast cancer by upregulating p53 signalling. *Biomark Res*, 12(1), 13. <https://doi.org/10.1186/s40364-024-00558-0>
- Tuo, Q. Z., Liu, Y., Xiang, Z., Yan, H. F., Zou, T., Shu, Y., Ding, X. L., Zou, J. J., Xu, S., Tang, F., Gong, Y. Q., Li, X. L., Guo, Y. J., Zheng, Z. Y., Deng, A. P., Yang, Z. Z., Li, W. J., Zhang, S. T., Ayton, S., . . . Lei, P. (2022). Thrombin induces ACSL4-dependent ferroptosis during cerebral ischemia/reperfusion. *Signal Transduct Target Ther*, 7(1), 59. <https://doi.org/10.1038/s41392-022-00917-z>

REVIEWERS' COMMENTS:

Reviewer #1 (Remarks to the Author):

The experiments the authors performed during this revision have addressed this referee's questions. The results (at least the lipidomics) are critical to the conclusions of this study and should be presented in the main manuscript.

Several parts of the writing remain inaccurate or difficult to read, the authors may want to work on this issue further to increase the readability of the manuscript. Some examples are listed here with suggestions for modifications:

Title: "via upregulating of ACSL4 and TFRC", removing "of" or "via upregulation of"?

Line 67, "Therefore, ferroptosis has recently become a hot research topic in a variety of liver diseases, including HIRI."-> "Ferroptosis has been extensively implicated in various forms of liver diseases, including HIRI"? References should be inserted as well.

Line 68, "Naoya Yamada proved for the first time that ferroptosis contributes to ". -> "Yamada et al demonstrated a contribution by ferroptosis to/in ..."

Line 70-71, "Then, some studies on the mechanism of ferroptosis regulation during HIRI were subsequently carried out. " An unnecessary sentence without specific information. Deletion is suggested.

Line 94, "(Acsl4) and Tfrc, which are two critical mediators of ferroptosis", "...which are two general contributors to ferroptosis sensitivity"

Line 142, "transmission electron microscopy (TEM) assays is also the classical method to detect ferroptosis. ", "...TEM is frequently used to detect the mitochondrial morphological aberrancy associated with ferroptosis"

Line 144, " When cells undergo ferroptosis, we can observe that the " Do the authors intend to describe their result here (should cite the data Fig. 1k) or to introduce a background knowledge?

Line 138-167, this paragraph is one and half page and difficult to read, maybe considering cutting into 2-3 paragraphs.

Line 176, "Moreover, results from the in vitro ", "... results from in vitro characterizations.."

Reviewer #1 (Remarks to the Author):

The experiments the authors performed during this revision have addressed this referee's questions. The results (at least the lipidomics) are critical to the conclusions of this study and should be presented in the main manuscript.

Response: Thanks for your kind suggestion. And this is indeed an essential point for our study. As suggested, We have placed the description of lipidomics results in the revised manuscript within Line 204-221 and the graphical representation in Supplementary Figure 4 and Supplementary Figure 5.

Several parts of the writing remain inaccurate or difficult to read, the authors may want to work on this issue further to increase the readability of the manuscript. Some examples are listed here with suggestions for modifications:

Title: “via upregulating of ACSL4 and TFRC”, removing “of” or “via upregulation of”?

Response: Thank you for your valuable suggestion. And we apologize for the oversight during the manuscript preparation. As suggested, we have revised the title with a better formation.

Line 67, “Therefore, ferroptosis has recently become a hot research topic in a variety of liver diseases, including HIRI.”-> “Ferroptosis has been extensively implicated in various forms of liver diseases, including HIRI”? References should be inserted as well.

Response: Thanks for your kind reminding. We apologize for the oversight during the preparation of the manuscript and we have carefully inserted the relative references according to your suggestion.

Line 68, “Naoya Yamada proved for the first time that ferroptosis contributes to ”. -> “Yamada et al demonstrated a contribution by ferroptosis to/in ...”

Response: Thanks for your kind suggestion. We apologize for the oversight during the manuscript preparation. As suggested, we have modified the sentence and highlighted in yellow in our revised manuscript.

Line 70-71, “Then, some studies on the mechanism of ferroptosis regulation during HIRI were subsequently carried out. ” An unnecessary sentence without specific information. Deletion is suggested.

Response: Thanks for your kind reminding. And we apologize for the oversight during the preparation of the manuscript. We have deleted it in our revised manuscript according to your suggestion.

Line 94,“ (Acsl4) and Tfrc, which are two critical mediators of ferroptosis”, “...,which are two general contributors to ferroptosis sensitivity”

Response: Thank you for the kind suggestion. And we apologize for the oversight during the preparation of the manuscript. As suggested, we have modified this and highlighted in yellow in our revised manuscript.

Line 142, “transmission electron microscopy (TEM) assays is also the classical method to detect ferroptosis. ”, “...TEM is frequently used to detect the mitochondrial morphological aberrancy associated with ferroptosis”

Response: Thanks for your kind reminding. And we apologize for the oversight during the preparation of the manuscript. We have corrected it and marked as yellow in the revised manuscript.

Line 144,” When cells undergo ferroptosis, we can observe that the ” Do the authors intend to describe their result here (should cite the data Fig. 1k) or to introduce a background knowledge?

Response: Thanks for your kind reminding. The sentence “When cells undergo ferroptosis, we can observe that the mitochondrial become smaller and the mitochondrial cristae decrease or disappear with TEM” is an introduce of

background knowledge and we have inserted the related reference. We apologize for the oversight during the preparation of the manuscript and thanks again for your kind reminding.

Line 138-167, this paragraph is one and half page and difficult to read, maybe considering cutting into 2-3 paragraphs.

Response: Thank you for the kind reminding. And we apologize for the oversight during the preparation of the manuscript. As suggested, We have separated this paragraph into three subsections based on your suggestion.

Line 176, “Moreover, results from the in vitro ”, “... results from in vitro characterizations..”

Response: Thanks for your kind suggestion. And we apologize for the oversight during the preparation of the manuscript. We modified this and highlighted in yellow in our revised manuscript.

We deeply apologize for the language structure problem of this article. Based on your valuable comments on our manuscript above, we sent the manuscript to AJE (American Journal Experts) for language polishing, and the Editing Certificate is attached at the bottom of this letter. Thank you again for your guidance on the syntactic structure of our article.

Editing Certificate

This document certifies that the manuscript

FTO deficiency in older livers exacerbates ferroptosis during ischaemia/reperfusion injury by upregulating ACSL4 and TFRC

prepared by the authors

Rong Li, Xijing Yan, Cuicui Xiao, Jiebin Zhang, Jinliang Liang, Jianye Cai, Xin Sui, Jiaqi Xiao, Haitian Chen, Yasong Liu, Chenhao Jiang, Guo Lv, Guihua Chen, Jia Yao, Yingcai Zhang, Jun Zheng, Yang Yang

was edited for proper English language, grammar, punctuation, spelling, and overall style by one or more of the highly qualified native English speaking editors at AJE.

This certificate was issued on **April 19, 2024** and may be verified on the AJE website using the verification code **7EA3-6709-47E6-2850-B85P**.

Neither the research content nor the authors' intentions were altered in any way during the editing process. Documents receiving this certification should be English-ready for publication; however, the author has the ability to accept or reject our suggestions and changes. To verify the final AJE edited version, please visit our verification page at aje.com/certificate. If you have any questions or concerns about this edited document, please contact AJE at support@aje.com.

AJE provides a range of editing, translation, and manuscript services for researchers and publishers around the world. For more information about our company, services, and partner discounts, please visit aje.com.